# Characterizing the evolutionary dynamics of cancer proliferation in single-cell clones with SPRINTER

Olivia Lucas [1,2,3,4,14], Sophia Ward [2,3,5,14], Rija Zaidi [1,2], Abigail Bunkum [1,2,6], Alexander M. Frankell [2,3], David A. Moore [2,3,7], Mark S. Hill [3], Wing Kin Liu[2,6], Daniele Marinelli [6,8,9], Emilia L. Lim[2,3], Sonya Hessey[1,2,4,6], Cristina Naceur-Lombardelli [2], Andrew Rowan[3], Sukhveer Kaur Purewal-Mann[10], Haoran Zhai [2,3], Michelle Dietzen [2,3,8], Boyue Ding[11], Gary Royle[11], Samuel Aparicio [12,13], TRACERx Consortium*, PEACE Consortium*, Nicholas McGranahan [2,8], Mariam Jamal-Hanjani [2,4,6], Nnennaya Kanu [2,15] ✉, Charles Swanton [2,3,4,15] ✉ & Simone Zaccaria [1,2,15] ✉

Proliferation is a key hallmark of cancer, but whether it differs between evolutionarily distinct clones co-existing within a tumor is unknown. We introduce the Single-cell Proliferation Rate Inference in Non-homogeneous Tumors through Evolutionary Routes (SPRINTER) algorithm that uses single-cell whole-genome DNA sequencing data to enable accurate identification and clone assignment of S- and G2-phase cells, as assessed by generating accurate ground truth data. Applied to a newly generated longitudinal, primary-metastasis-matched dataset of 14,994 non-small cell lung cancer cells, SPRINTER revealed widespread clone proliferation heterogeneity, orthogonally supported by Ki-67 staining, nuclei imaging and clinical imaging. We further demonstrated that high-proliferation clones have increased metastatic seeding potential, increased circulating tumor DNA shedding and clone-specific altered replication timing in proliferation- or metastasis-related genes associated with expression changes. Applied to previously generated datasets of 61,914 breast and ovarian cancer cells, SPRINTER revealed increased single-cell rates of different genomic variants and enrichment of proliferation-related gene amplifications in high-proliferation clones.

High proliferation is one of the key hallmarks of cancer[1] and is linked to worse clinical outcomes across a range of tumor types[2–10]. Thus far, proliferation has been estimated by measuring the fraction of S-phase cells using pathological or experimental techniques on bulk tumor samples[2–5], such as Ki-67 staining, or using bulk and single-cell RNA sequencing[6–8,11–13]. However, most tumors have been shown to be heterogeneous compositions of genetically distinct subpopulations of cancer cells, or clones, with different evolutionary histories and roles[14–20].

Because proliferation may vary between distinct clones within the same tumor, the joint inference of clone-specific proliferation rates and the reconstruction of their evolutionary dynamics may allow the identification of clones that develop more aggressive phenotypes[14,21–26] (for example, metastatic potential), providing mechanistic insight into the link between proliferation and prognosis. Recent studies have shown that accurate identification of clones and reconstruction of their evolution requires whole-genome DNA sequencing, as it provides a sufficiently

high number of mutations and genomic alterations for robust evolutionary analyses[15–19,27]. Therefore, the joint measurement of clone-specific proliferation rates and related evolutionary dynamics has thus far been unfeasible because proliferation and tumor clonal evolution could not easily be measured from the same data for the same cells.

Recent single-cell whole-genome DNA sequencing (scDNA-seq) technologies based on tagmentation without genome preamplification[17,19,20], such as direct library preparation+ (DLP+)[17,19], and similar techniques[18,28] have enabled the accurate genomic and evolutionary characterization of distinct tumor clones[17–20] while also providing a signal to identify cell cycle states[13,17,28]. On the one hand, scDNA-seq data enables the inference of single-cell copy-number alterations (CNAs)[13,17–20,29], which are frequent genomic alterations in cancer resulting from amplifications or deletions of large genomic regions[15,30]. Tumor clones can thus be inferred by grouping cells that share the same CNAs[13,17–20] and their evolution can be reconstructed using corresponding mutations[13,17–20]. On the other hand, scDNA-seq data can be used to identify S-phase cells because replication induces fluctuations in the sequencing read counts observed across the whole genome[13,17,28]. In fact, replication is an asynchronous process in which different genomic regions replicate their DNA at different times during S phase, and early-replicating regions thus yield higher read counts than late-replicating regions.

In principle, these joint scDNA-seq measurements should allow the estimation of clone proliferation by analysis of S fractions in distinct tumor clones. However, in practice, this task remains unfeasible due to the lack of a formal method to assign S-phase cells to their corresponding clones, which is a challenging problem because replication-induced fluctuations prevent accurate CNA identification in S-phase cells[13,17–20,29]. Moreover, high-sensitivity identification of S-phase cells is required for accurate S fraction estimates of the small clones often found in single-cell studies[17–20], but two key limitations restrict the power of previous methods[13,17,28]. First, these methods rely on standard algorithms for copy-number analysis (for example, guanine–cytosine (GC) content correction or copy-number segmentation) that ignore sequencing fluctuations induced by replication. Second, they assume that the sequenced cells belong to a homogeneous population and thus aggregate all cells together, identifying S-phase cells as those with some sequencing signal that deviates from the rest[13,17,28]. While this assumption may be true in cell lines (used in most previous studies[13,17,28]), this is not the case in cancer tissues that are often heterogeneous mixtures of normal and different cancer cell clones[14,15,18,21], such that each clone may need to be treated differently for S-phase identification.

In this study, we introduce Single-cell Proliferation Rate Inference in Non-homogeneous Tumors through Evolutionary Routes (SPRINTER), an algorithm that uses tumor scDNA-seq data to enable accurate identification and clone assignment of S- and G2-phase cells, thus providing a proxy to estimate clone-specific proliferation rates. We evaluated SPRINTER's accuracy by generating a scDNA-seq dataset of 8,844 cells from diploid and tetraploid cell lines sorted with 5-ethynyl-2-deoxyuridine (EdU) into different cell cycle phases[31], providing a more accurate ground truth dataset than previous approaches.

While the link between cancer proliferation and prognosis has been clearly shown[2–10], SPRINTER allows us to investigate if distinct clones co-existing within the same tumor have different proliferation rates, particularly clones with different evolutionary roles, such as metastatic seeding clones comprising the subset of cancer cells responsible for metastasis. To explore this, we generated a longitudinal, primary-metastasis-matched dataset of 14,994 single non-small cell lung cancer (NSCLC) cells, applied SPRINTER and performed detailed phylogenetic analysis to characterize the evolutionary dynamics of genetic and non-genetic features, such as proliferation and altered replication timing (ART), of distinct clones. We additionally analyzed circulating tumor DNA (ctDNA), for which a link with proliferation has only been revealed in previous bulk-based studies[32–34] for distinct tumors in different patients. Furthermore, we illustrated SPRINTER's broad applicability on previous scDNA-seq datasets[19] of 61,914 cells from 7 triple-negative breast cancer (TNBC) and 15 high-grade serous ovarian cancer (HGSC) tumors.

## Results
### The SPRINTER algorithm
The SPRINTER algorithm uses scDNA-seq data to identify S- and G2-phase cells and assign them to distinct tumor clones identified using inferred single-cell CNAs. SPRINTER achieves this goal by leveraging prior information on genomic regions that are expected to have early or late replication timing, which is known to be conserved across a high fraction of the genome in different cell types[35–37] and cancer cells[38–43] (~50% at minimum; Supplementary Fig. 1). Because the replication timing of some genomic regions can still vary in the analyzed cells, SPRINTER uses statistical approaches that do not fully rely on this prior information but rather account for the presence of potential changes or errors. As such, SPRINTER introduces two key contributions to overcome previous limitations. First, SPRINTER introduces a probabilistic method to enable the accurate clone assignment of S-phase cells. CNAs cannot be directly inferred for S-phase cells because both replication and CNAs induce read count fluctuations in scDNA-seq data (Extended Data Figs. 1–3). Therefore, SPRINTER corrects replication-induced fluctuations using the distribution of early- or late-replicating regions across the genome to calculate the probability that any S-phase cell belongs to each clone identified using non-S-phase cells (Extended Data Fig. 4). Second, SPRINTER introduces a replication-aware framework for the accurate identification of S-phase cells. Particularly, SPRINTER extends previous methods that rely on algorithms designed for CNA analysis of non-S-phase cells[13,17–20,29] to account for expected replication-induced fluctuations and introduces a statistical permutation test based on these fluctuations for the high-sensitivity identification of S-phase cells.

SPRINTER is composed of six steps (Fig. 1) based on a partitioning of the reference genome into bins (50 kb by default). First, it identifies early- and late-replicating bins using experimentally derived replication scores from normal and cancer cells[38,44,45] (Supplementary Figs. 1–4) and calculates read depth ratios (RDRs) to capture read count variations as per standard CNA identification[13,17–20,29]. During this step, SPRINTER accounts for varying total read counts for cells in different phases and incorporates a replication-aware GC-content bias correction (Supplementary Figs. 5 and 6). Second, it infers high-confidence CNA-induced segments in the genome of each cell while accounting for replication-induced RDR fluctuations (leveraging the fact that CNAs are substantially larger than regions with the same replication timing[17–19,30]; Supplementary Fig. 2). Third, it identifies S-phase cells using a statistical permutation test based on the higher and lower RDRs expected for early and late bins within copy-number segments in these cells, respectively (Extended Data Figs. 1–4 and Supplementary Fig. 7). Fourth, it identifies clones by inferring and clustering CNAs in G0/G1/G2-phase cells by extending previous approaches[17–19]. Fifth, it assigns S-phase cells to maximum-a-posteriori probability clones and infers related CNAs by subtracting replication-induced fluctuations from RDRs (Extended Data Fig. 4). Finally, it identifies G2-phase cells per clone based on expected higher total read counts[17] (Supplementary Fig. 8). The details of each of SPRINTER's steps are reported in Methods.

### SPRINTER exhibits high accuracy and sensitivity
To evaluate SPRINTER's performance, we generated a ground truth scDNA-seq dataset of 8,844 diploid and tetraploid cancer cells with known cell cycle phases from the HCT116 colorectal cancer cell line[46]. While previous datasets have been generated using standard fluorescence-activated cell sorting (FACS)[17,28], these approaches are error-prone and mostly enriched for mid-S-phase cells and are thus not suitable for the comprehensive assessment of S-phase identification.

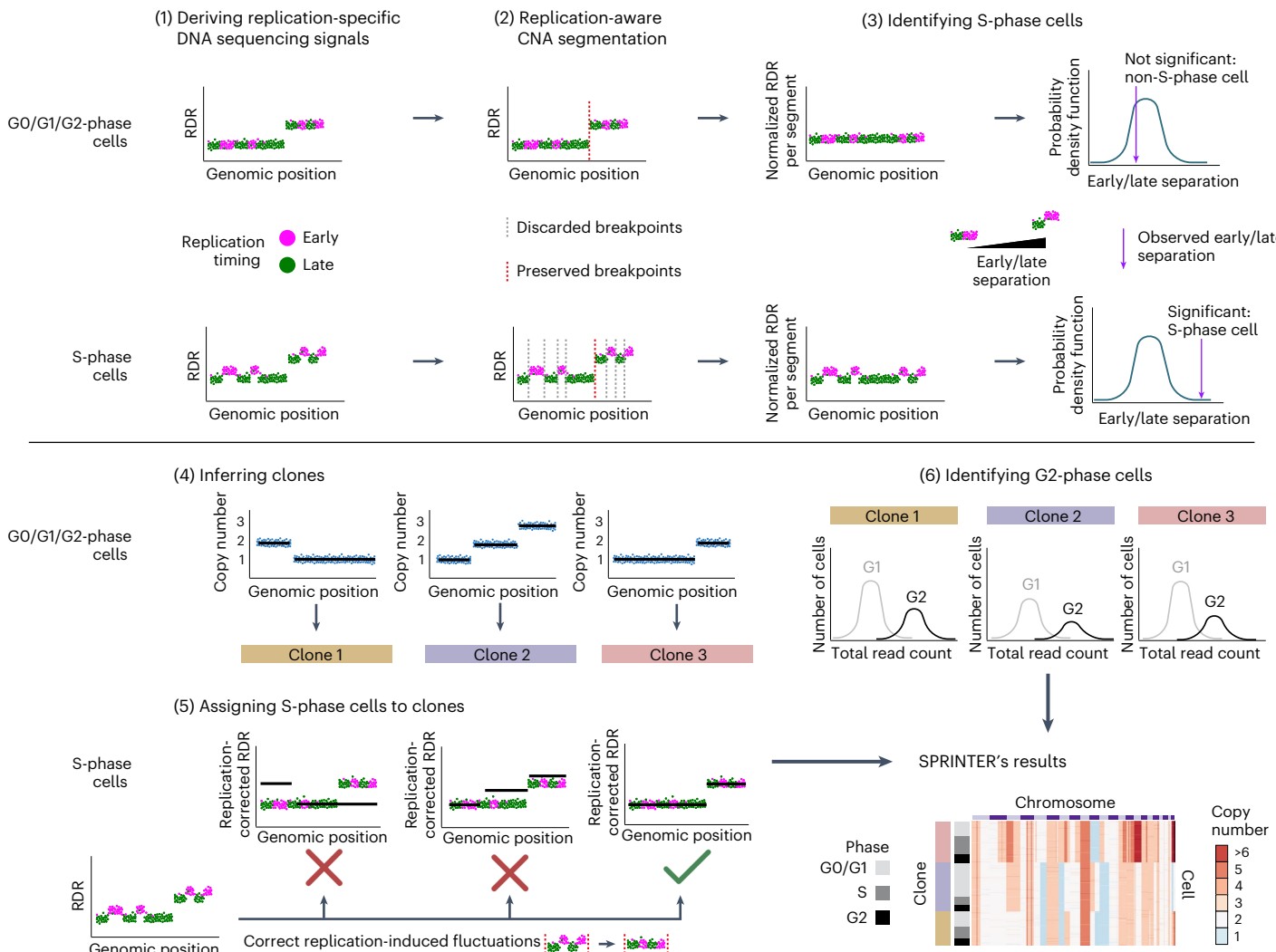

**Fig. 1 | The SPRINTER algorithm.** There are six main steps in SPRINTER. (1) The first step calculates the RDR and replication timing (early and late in magenta and green, respectively) of each genomic bin. (2) The second step infers segments of neighboring bins likely to be affected by the same CNAs by identifying candidate breakpoints independently in early or late bins and preserving only those breakpoints supported by both (dashed red lines preserved versus dashed gray lines discarded). (3) The third step identifies S-phase cells by performing a statistical permutation test of replication timing on RDRs normalized per segment (to remove the effect of CNAs) to assess the presence of significant differences between early (higher values) and late (lower values) bins expected for S-phase cells (bottom row) in contrast to G0/G1/G2-phase cells (top row). (4) The fourth step infers clones by identifying cell-specific CNAs (black lines) for all G0/G1/G2-phase cells and grouping cells with the same complement of CNAs (colored bars). (5) The fifth step assigns each S-phase cell to the maximum-a-posteriori clone (green check mark)—RDRs are corrected for replication fluctuations, and clone assignment is chosen to maximize the posterior probability across all possible assignments (best fit of black lines). (6) The sixth step identifies G2-phase cells per clone by deconvolving the distribution of total read counts yielded by either G0/G1-phase (light gray with lower values) or G2-phase (black with higher values) cells. SPRINTER's results—each cell (row) with inferred CNAs (colors) across bins (columns) is assigned to a clone, providing estimates of S (left dark gray bars) and G2 (black bars) fractions. The figure is created with BioRender.com.

To overcome these limitations, we applied a FACS approach incorporating EdU, as demonstrated in recent studies[31], and sequenced cells separated into five different cell cycle phases using DLP+ (Supplementary Figs. 9–11 and Methods). The availability of tetraploid cells also improved upon previous datasets, as the increased rate of CNAs in genome-doubled cells may complicate related analyses[30,47].

In the identification of S-phase cells, we found that SPRINTER outperformed two previously established methods, the cell cycle classifier (CCC)[17] and the median absolute deviation of pairwise differences (MAPD) method[28], as well as a version of the latter incorporating replication timing information (rtMAPD), in both the diploid and tetraploid datasets, with improvements of 10–90% in mid- and late-S-phase identification while maintaining high precision (Fig. 2a and Supplementary Fig. 12). SPRINTER's improved accuracy was further confirmed on a previous phase-sorted dataset[28] of 5,970 lymphoblastoid cells generated with a different scDNA-seq technology (Supplementary Figs. 13 and 14). In contrast, methods like MAPD that aggregate all cells during S-phase identification failed to deal with an additional dataset comprising cells of mixed ploidy, confirming the importance of SPRINTER's cell-specific test in analyzing these heterogeneous but realistic cases (Supplementary Fig. 15). Notably, SPRINTER's accuracy remained robust for a fraction of replication timing errors higher than the maximum expected in both normal and cancer cells (Supplementary Fig. 16) and for the use of different input replication scores (Supplementary Fig. 17). Moreover, SPRINTER accurately identified G2-phase cells (>80% precision and recall; Supplementary Fig. 18) and provided the best prediction of actively replicating cells (in S and G2 phase; Fig. 2b). Further details are given in Supplementary Note 1.

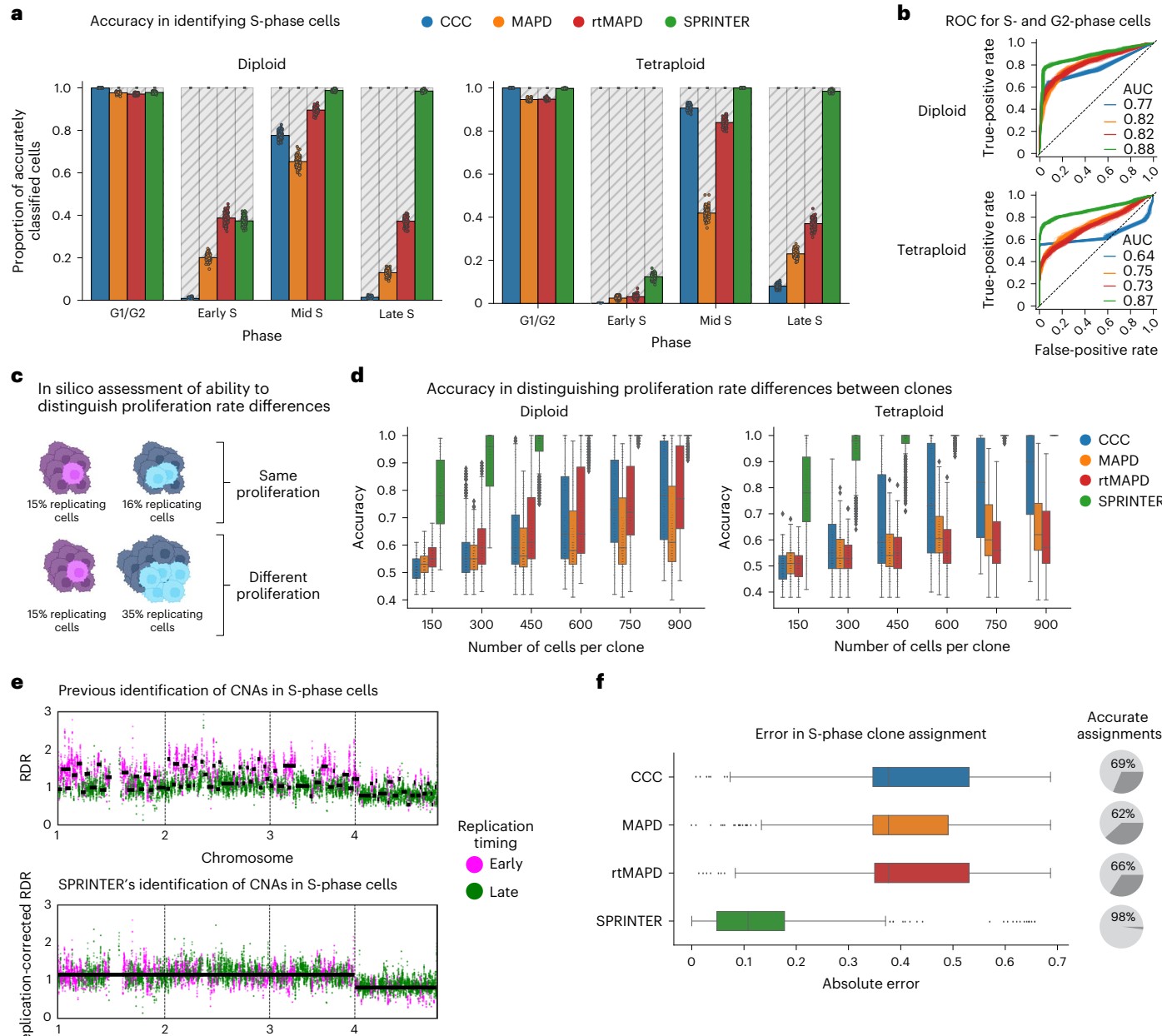

**Fig. 2 | SPRINTER improves S-phase identification and enables accurate clone assignment of S-phase cells. a**, The proportion of correctly identified G1/G2- and S-phase cells (*y* axis) was computed for CCC (blue), MAPD (orange), rtMAPD (MAPD extended with replication timing, red) and SPRINTER (green) across cell cycle phases (*x* axis) for 100 cell subpopulations (dots), each formed by sampling 500 cells from the diploid (left) or tetraploid (right) ground truth datasets. **b**, ROC curves (false-positive rates versus true-positive rates) measure the performance in distinguishing G1-phase cells from actively replicating cells using the classification scores computed by existing methods (blue, orange and red) or combining SPRINTER's S- and G2-phase *P* values (using the minimum, green) by bootstrapping 300 diploid (top) or tetraploid (bottom) cells for 100 repeats (each curve). **c**, A binomial process was used to generate cell subpopulation pairs with the same (top) or different (bottom) true underlying fractions of replicating cells (that is, proliferation). The figure is created with BioRender.com. **d**, The proliferation accuracy was computed for all methods (colors) considering 600

pairs of clones generated as described in **c** by sampling varying numbers of diploid (left) and tetraploid (right) cells per clone (*x* axis) with varying S and G2 fractions (20–30% ± 30–50%) for 50 repeats (dots). **e**, Top, RDRs across 50 kb bins (*x* axis) for an S-phase cell are affected by replication-induced fluctuations (early- and late-replicating bins in magenta and green, respectively) preventing accurate CNA identification (scattered black lines for expected CNAs). Bottom, instead, SPRINTER's replication-corrected RDRs are similar to CNA expectations (black lines). **f**, The absolute error rate (*x* axis) between true and expected fractions of S-phase cells assigned to a clone was calculated per cell using all methods (colors) in 30 populations of 300 tetraploid cells each, altogether comprising 389 clones. The proportion of clones for which the assigned true S fraction was compatible with the expected S fraction was computed using a binomial test (pie charts). In **d** and **f**, box plots show the median and IQR with whiskers denoting values within 1.5 times the IQR from the first and third quartiles. AUC, area under the curve; ROC, receiver operating characteristic; IQR, interquartile range.

We further demonstrated that the increased number of replicating cells inferred by SPRINTER was necessary to accurately distinguish proliferation rate differences between clones of sizes similar to those identified in previous studies[17–20] (Fig. 2c,d and Supplementary Note 2).

Even more notably, we found that SPRINTER's new features are required for the accurate clone assignment of S-phase cells, outperforming correlation-based heuristics proposed in previous studies[13] (98% versus 62–69% accuracy), as measured using the clones in the tetraploid

dataset expected to have equal proliferation rates[46] (Fig. 2e,f, Supplementary Figs. 19 and 20 and Supplementary Note 3). Finally, a spike-in experiment of CNAs demonstrated that SPRINTER's clone assignments also enabled the accurate inference of most >3 Mb CNAs for both S- and non-S-phase cells (Supplementary Fig. 21).

## Primary and metastatic tumor proliferation heterogeneity

To assess whether genomically distinct clones with different proliferation rates are present in tumors, we applied DLP+ (refs. 17,19) to sequence 14,994 cells from ten tumor samples from patient CRUKP9145 with metastatic NSCLC enrolled in the TRAcking non-small cell lung Cancer Evolution through therapy (Rx) (TRACERx) study[14,21] and Posthumous Evaluation of Advanced Cancer Environment (PEACE) autopsy study[32] (Supplementary Figs. 22 and 23 and Supplementary Note 4). We sequenced 9,532 cells from five distinct samples from the primary tumor (regions 2, 3, 4, 5 and 8; Supplementary Fig. 24) and 5,462 cells from five samples obtained from anatomically distinct metastases (left adrenal, right adrenal, left frontal lobe, right occipital lobe and liver; Supplementary Fig. 25), demonstrating the applicability of DLP+ to fresh frozen human metastatic tissues obtained at autopsy. Using SPRINTER, we identified the presence of 52 distinct tumor clones with S- and G2-phase cells (Extended Data Fig. 5 and Supplementary Figs. 26 and 27), high rates of CNAs and enough cells for accurate clone identification (Supplementary Fig. 28).

SPRINTER revealed widespread heterogeneity in the proliferation rates of clones between and within tumor samples (Fig. 3a). In fact, SPRINTER identified clones with substantially different proliferation rates both between different primary tumor samples and between different metastases (Fig. 3a and Supplementary Fig. 29). Moreover, in nearly all samples, SPRINTER identified clones with significantly higher or lower S fractions compared to other clones within the same sample (Fig. 3a), with similar patterns supported by related G2 fractions (Supplementary Figs. 30 and 31). Notably, SPRINTER's clone-specific estimates were required to identify these differences as they would have been missed by previous bulk-based estimates (for example, Ki-67 analysis). For example, primary regions 2, 3, 4 and 8 had indistinguishable bulk-based S fraction estimates (22% compatible with all samples; Fig. 3a) despite the presence of several differentially proliferative clones. Conversely, the bulk-based estimate in the right occipital lobe was half that of the right adrenal (13% versus 31%) despite the presence of clones with similar S fractions.

We orthogonally validated SPRINTER's results using Ki-67 analysis, DLP+ nuclear imaging and clinical imaging (see details in Supplementary Notes 5 and 6). First, for samples with sufficiently high-quality Ki-67 staining, proliferation estimates from Ki-67 and SPRINTER were overall consistent, and Ki-67 analysis corroborated the heterogeneous proliferation rates revealed by SPRINTER within samples (Fig. 3b and Supplementary Fig. 32). Second, SPRINTER's phase predictions were significantly associated with nuclear diameters measured with DLP+ nozzle-based imaging[17] both per cell and per clone (Fig. 3c and Supplementary Fig. 33), in keeping with previous expectations[17] and with the ground truth datasets (Supplementary Fig. 34). Finally, SPRINTER's estimated average S fractions ranked metastases in the same order as tumor growth rates measured using longitudinal computed tomography and magnetic resonance imaging (Extended Data Fig. 6), further confirming that increased S fractions relate to increased proliferation and disease burden rather than changes in the length of cell cycle phases. Furthermore, while the numerous clones identified by SPRINTER's single-cell analysis demonstrated its increased resolution over previous bulk analysis of the same tumor[14], we observed highly consistent somatic single-nucleotide variants (SNVs) and CNAs (Fig. 3d, Supplementary Figs. 35 and 36 and Supplementary Note 4).

## The evolution and ART of clones with different proliferation

In addition to the identification of clones, scDNA-seq data enable the accurate reconstruction of their evolutionary history[17,18,20] and related metastatic migration patterns[48]. We reconstructed the tumor phylogeny based on both SNVs and CNAs (Fig. 4a,b and Supplementary Figs. 37–41) in multiple steps by combining pseudobulk approaches[17,18] and extending existing phylogenetic methods[49–51] (Methods). We further used the MACHINA algorithm[48] to reconstruct metastatic migration patterns and identify seeding clones, which are the ancestral clones comprising the disseminating cancer cells responsible for seeding metastases. As such, we identified three main metastatic clades, each of which was defined by a distinct seeding clone in the primary tumor (Fig. 4c)—the first defined by the clone seeding the liver metastasis, the second by the clone seeding the right occipital lobe metastasis and the third by the clone seeding the other metastases. Remarkably, only one of these clades, the third, contained most of the clones with the highest proliferation.

We investigated the presence of genetic alterations in key cancer genes associated with these clades. Despite the high proliferation of the third clade, no driver mutation was identified unique to this clade and shared by most of its clones, and all the identified driver mutations, for example, in TP53, NF1 and CDK12 genes, were shared with other clades (Supplementary Fig. 38). Similarly, no particular CNA specific to this clade was identified (Supplementary Fig. 41). In addition to genetic alterations, non-genetic alterations have a key role in cancer progression[40–43] and may be present in different metastatic clades. To investigate this, we demonstrated that SPRINTER's clone assignments can be used to identify clone-specific ART (that is, changes in the replication timing of a genomic region in the tumor compared to the default reference replication timing classifications derived from normal cell lines; Methods).

Overall, the fraction of the genome with ART was <10% on average across clones (Extended Data Fig. 7), which matched previous measurements[38,40–43] and did not affect SPRINTER's results

---

**Fig. 3 | SPRINTER identifies tumor clone proliferation heterogeneity in patient CRUKP9145 with NSCLC. a**, The distributions of SPRINTER's inferred S fractions (bottom, *y* axis) for each NSCLC clone (*x* axis) with varying cell numbers (top, *y* axis) in primary (top) and metastatic (bottom) samples were calculated by bootstrapping (300 repeats; dashed lines represent sample-level averages). Clone S fractions were compared per sample using a two-sided chi-square test, combined using the minimum and a Benjamini–Hochberg correction was applied (family-wise error rate = 0.1; red asterisks indicate significant *P* values). Sample-level S fraction 95% CIs (between axes) were computed by bootstrapping cells per sample. \**P* < 0.1, \*\**P* < 0.05 and \*\*\**P* < 0.005. **b**, Ki-67 staining from one representative slide in primary and metastatic samples, indicating areas with high and low Ki-67 (boxes) that were consistent with SPRINTER clone S fractions (red asterisk). **c**, Top, nuclear diameter (*x* axis, micrometers, normalized by sample mean) was measured by DLP+ nozzle-based imaging for 14,569 cells with successfully recorded images inferred to be in G1, S or G2 phase by SPRINTER (*y* axis), with each pair of distributions compared using a one-sided Mann–Whitney *U* test (*P* values on right). Bottom, the nuclear diameter per clone (*x* axis) was calculated using the minimum diameter across the cells in each clone (each dot) that were assigned to different cell cycle phases by SPRINTER (*y* axis). Across cell cycle phases, clones are linked by lines, such that the line width is proportional to clone size and the line color indicates whether the nuclear diameter per clone has increased as expected (red) or not (blue). Nuclear diameters in different cell cycle phases were compared per clone using a one-sided Wilcoxon signed-rank test (*P* values on right). Right, example microscopy images of nuclei in each phase. **d**, For five primary tumor samples in this study (colored circles on photo) and three additional samples (gray circles), each bulk clone identified in previous analysis (hexagons comprising clones with different inner shapes of size proportional to cell proportion) was assigned to the most similar SPRINTER clone using SNVs (colors, with legend marker size proportional to SPRINTER's inferred S fraction). In **a** and **c**, box plots show the median and the IQR with whiskers denoting values within 1.5 times the IQR from the first and third quartiles. CI, confidence interval.

(Supplementary Fig. 42). We found ART events affecting genes known to impact tumor proliferation or metastatic potential that were shared or unique to different metastatic clades (Fig. 4d and Extended Data Fig. 8); for example, ART of *PDL1* and *PIK3CA* was shared by all clones, ART of *CDK12* was only present in the right phylogenetic branch comprising the second and third clades and ART of *KRAS* was mostly exclusive to the most proliferative third clade. Because ART is associated with differential gene expression (higher/lower expression for genes affected by late-to-early/early-to-late ART compared to normal tissue without ART, respectively)[38,39], we showed that these ART events were supported by related expression changes. Specifically, we compared previous bulk RNA sequencing data[52] from matched primary tumor regions to primary regions from different clades, to cancer and normal tissue samples from 347 other TRACERx patients or to a premortem relapse sample from the

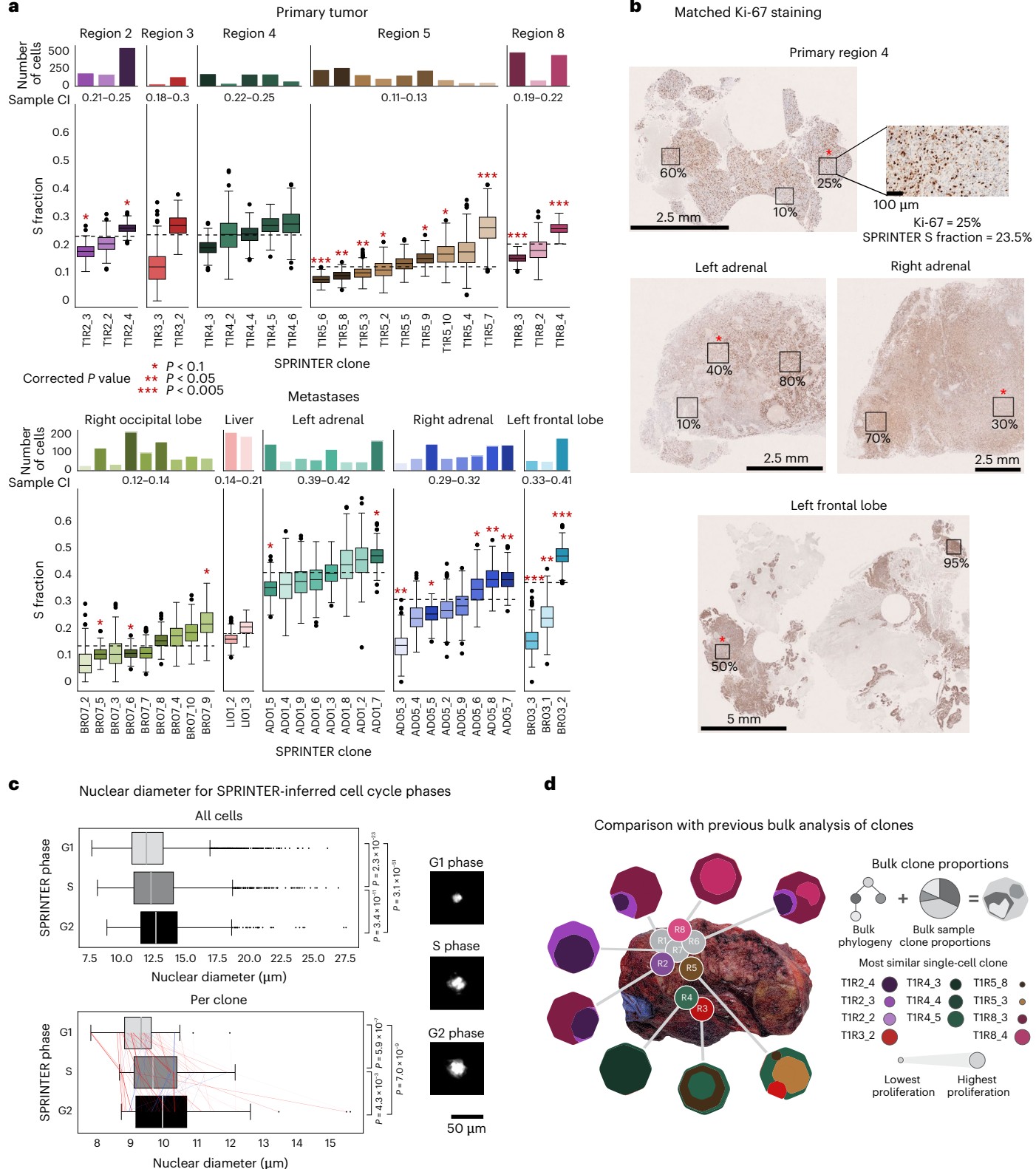

**a** Primary tumor

**b** Matched Ki-67 staining

Corrected *P* value
\* *P* < 0.1
\*\* *P* < 0.05
\*\*\* *P* < 0.005

**c** Nuclear diameter for SPRINTER-inferred cell cycle phases

**d** Comparison with previous bulk analysis of clones

left adrenal metastasis in the third clade (Fig. 4d and Supplementary Fig. 43). For example, the comparisons with different clades and the premortem relapse sample confirmed a significant increase in *KRAS* expression in the third clade, particularly in the left adrenal metastasis that contained the highest proliferation clones, most of which displayed late-to-early ART of *KRAS* (Fig. 4d). Moreover, reduced *PDL1* expression correlating to early-to-late ART in almost all clones was further confirmed by the clinal immunohistochemistry report (PDL1 0%; Methods).

### Seeding potential and ctDNA shedding of proliferative clones

In addition to primary tumor clones, clones in metastases can also disseminate and seed further metastases[21]. In fact, we found that the third metastatic clade, comprising the most proliferative clones identified by SPRINTER (Fig. 4b), was the only clade containing additional seeding clones—MACHINA identified two seeding clones in the left adrenal metastasis that disseminated and further seeded two other metastases in the right adrenal and left frontal lobe (Fig. 4c). This occurrence of metastasis-to-metastasis seeding was also supported by clinical imaging, which indicated that the right adrenal metastasis likely arose after the left adrenal metastasis (Extended Data Fig. 6). Prompted by this observation, we investigated whether high-proliferation clones are the specific clones that are more likely to seed metastases. While the proliferation of seeding clones cannot generally be measured because these clones are often extinct at the time of sampling[48], we calculated a seeding genetic distance between each SPRINTER clone and the closest seeding clone using the evolutionarily reconstructed SNVs and CNAs as a proxy. We found significant negative correlations between the seeding genetic distance (where a smaller distance indicates a higher similarity to seeding clones) and S fraction for primary tumor and metastatic clones using both SNV- and CNA-based distances (Fig. 4e and Supplementary Fig. 44). These results suggest that clones that are more genomically similar to seeding clones exhibit higher proliferation rates, thus indicating an association between high clone proliferation and the metastatic seeding potential of individual clones.

Finally, because highly proliferative tumors in different patients have been shown to shed more ctDNA into the bloodstream[33,34], we leveraged SPRINTER's results to investigate whether the same association holds for different clones within the same tumor. Based on previous ctDNA measurements for the same patient[33], we found that clones belonging to the third metastatic clade harbored SNVs with high ctDNA frequency (that is, cancer cell fraction) across multiple time points, consistent with the high-proliferation rates inferred by SPRINTER (Supplementary Fig. 45a). Because ctDNA frequency is not only influenced by proliferation but also by clone volume (that is, number of cells), we calculated a ctDNA shedding index at surgery for each clone as the difference between ctDNA and primary tumor frequencies of SNVs, with the latter estimated based on either this single-cell analysis or previous bulk analyses[14] (Methods). In all cases,

the ctDNA shedding index was significantly and strongly correlated with SPRINTER's estimated S fractions (Fig. 4f and Supplementary Fig. 45), indicating differential ctDNA shedding between clones, with more proliferative clones shedding more ctDNA.

### Dynamics of genomic variants in proliferative clones

Finally, we demonstrated applicability to different datasets and cancer types by applying SPRINTER to two previous datasets[19], including 42,009 cells from 7 TNBC tumors and 19,905 cells from 15 HGSC tumors. SPRINTER identified 280 tumor clones with CNAs highly consistent with those previously inferred for non-S-phase cells (Supplementary Fig. 46). Moreover, SPRINTER identified the presence of clones with varying S fractions in most tumors (Fig. 5a), supported by similar patterns of G2 fractions (Extended Data Fig. 9). Overall, there was no relationship between the number of cells in a clone and its S fraction, nor between the number of clones in a tumor and the presence of differentially proliferative clones, indicating that SPRINTER's results are not biased by varying clone sizes or numbers (Supplementary Fig. 47).

Leveraging these large datasets, we investigated whether there is a relationship between clone proliferation and the rates of different genomic variants in individual cells by integrating SPRINTER with scDNA-seq measurements of clone-specific variants[17–19]. When calculating single-cell rates of clone-specific SNVs, structural variants (SVs) and CNAs for each cell individually (Methods), we found in both datasets that cells belonging to high-proliferation clones (higher than the cancer-type median) displayed significantly higher rates of all types of variants compared to cells belonging to low-proliferation clones (Fig. 5b–d). Because most of these SNVs have been shown[19] to be generated by mutational processes that act during cell divisions[53,54] and most SVs and CNAs might also be generated during cell divisions[55,56], these results are compatible with the expectation that clones with higher proliferation underwent more cell divisions.

We next investigated whether specific driver mutations or CNAs in known cancer genes were enriched in high-proliferation clones in the TNBC and HGSC datasets (Methods). We found several oncogene amplifications that were significantly associated with increased clone proliferation (for example, *CDK4* and *EGFR*; Fig. 5e), further supported by a gene set enrichment analysis revealing an enrichment in relevant pathways related to the cell cycle and proliferation (for example, PI3K/AKT/mTOR signaling and KRAS signaling upregulation; Fig. 5f). Moreover, a smaller number of driver mutations and deletions in tumor suppressor genes (for example, *KEAP1* and *SMAD4*) were also significantly associated with high clone proliferation (Supplementary Fig. 48), matching results in previous cell line small interfering RNA experiments[57,58].

Finally, SPRINTER can elucidate changes in the relative length of different cell cycle phases that might occur in cancer[59], given that they

---

**Fig. 4 | SPRINTER reveals a link between clone proliferation and metastatic seeding, and clone-specific ART present in distinct metastatic clades.**
**a**, Tumor phylogeny was reconstructed for SPRINTER's single-cell clones (tree leaves) from patient CRUKP9145 (colored by sample, with clones uniquely shaded). Seeding clones (dark gray) and ancestral clones (white with border colored according to inferred anatomical site) were inferred, with some clones harboring ctDNA-tracked SNVs (Roman numerals). **b**, Phylogeny from **a** with clones colored by SPRINTER's S fractions. **c**, Across samples (anatomical location indicated as circles on body map), metastatic migrations (arrows) were inferred, and metastatic clades (blue, green and pink with corresponding clones indicated in tree) were defined based on primary tumor seeding clones. The figure is created with BioRender.com. **d**, In the two main phylogenetic branches containing different metastatic clades (top row), SPRINTER inferred ART (colored rectangles) for each clone (second row) for genes (left) known to impact proliferation or metastatic potential, with reference replication timing derived from normal cells shown (left column). ART is supported by related gene

expression changes measured using bulk RNA sequencing (right heatmap), with late-to-early and early-to-late ART associated with increased and decreased gene expression, respectively (*P* values derived using a two-sided Wald test with a Benjamini–Hochberg correction with family-wise error rate = 0.05). \**P* < 0.1, \*\**P* < 0.05 and \*\*\**P* < 0.01. **e**, For each SPRINTER clone (dot) in the primary tumor (dark blue) or metastases (orange), the seeding genetic distance (*x* axis) computed with respect to the closest seeding clone based on either SNVs (left) or CNAs (right) was compared to SPRINTER's S fraction (*y* axis) using two-sided Pearson correlation tests (correlation coefficients and *P* values reported), and the 95% CI was calculated for linear regressions (shaded areas). **f**, For each ctDNA-tracked clone (dot), a ctDNA shedding index (*x* axis) was calculated using the frequency of SNVs for either (left) SPRINTER single-cell clones or (right) previous bulk clones and compared to the maximum S fraction inferred from descendant SPRINTER clones (*y* axis). In each case, a two-sided Spearman correlation test was performed (with correlation coefficients and *P* values reported), and the 95% CI was calculated for linear regressions (shaded areas).

are expected to induce changes in the relative ratio of G2 and S fractions (G2/S ratio; Methods). For example, in the TNBC dataset, SPRINTER tumor clones with previously identified[19] homologous recombination deficiency (HRD) displayed a significantly higher G2/S ratio than other clones ($P = 0.008$; Extended Data Fig. 10). This result is consistent with a prolonged G2 phase relative to S phase in HRD clones, as reported in previous studies[59,60].

## Discussion

Despite several evolutionary studies on cancer[14–21], the evolutionary dynamics of cancer phenotypes remain poorly explored, partially due to the lack of methodologies that allow the joint and accurate characterization of cancer genotypes and phenotypes. Recent scDNA-seq technologies enable a step in this direction by jointly allowing accurate genomic characterization of distinct tumor clones and measurement of

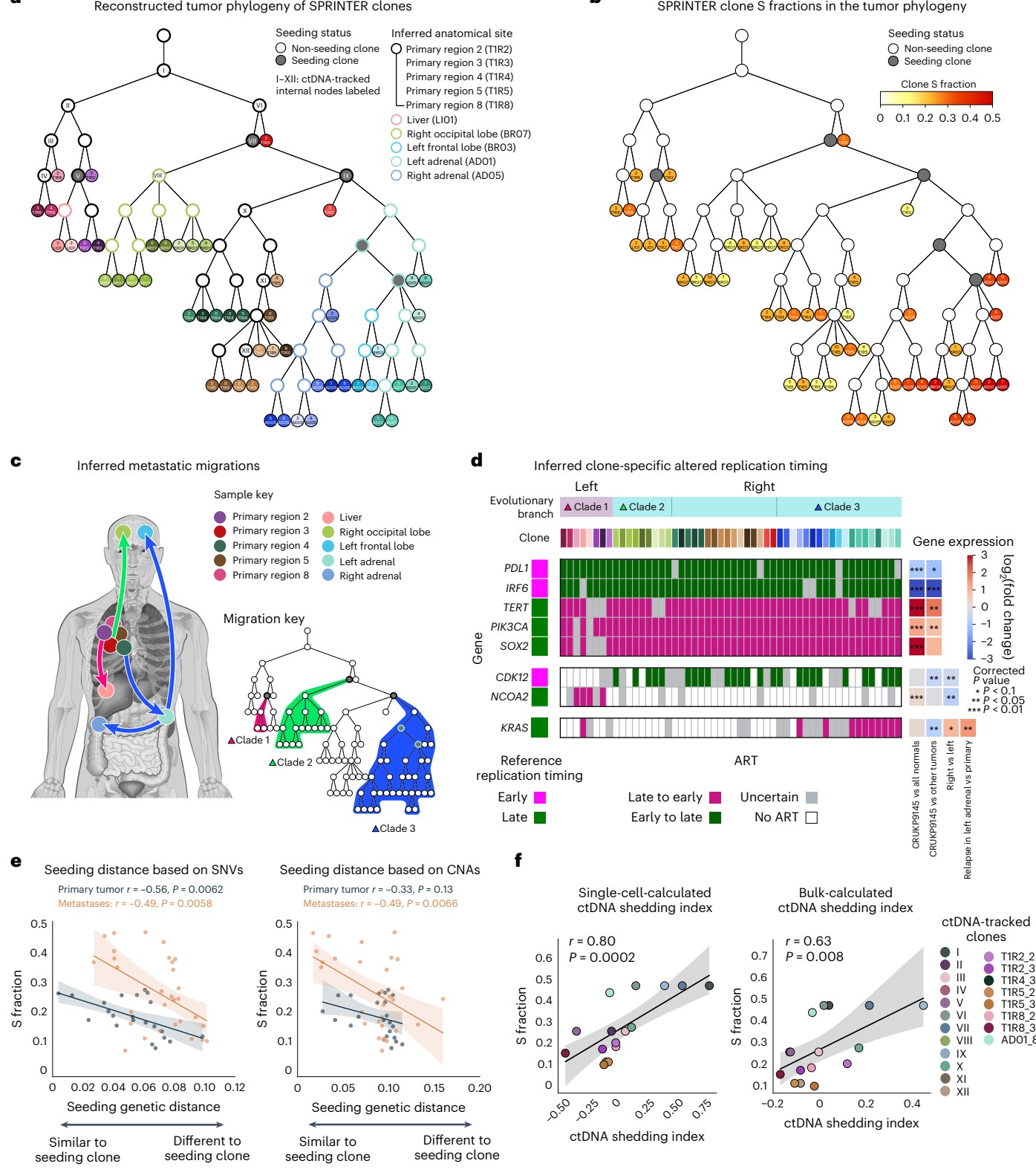

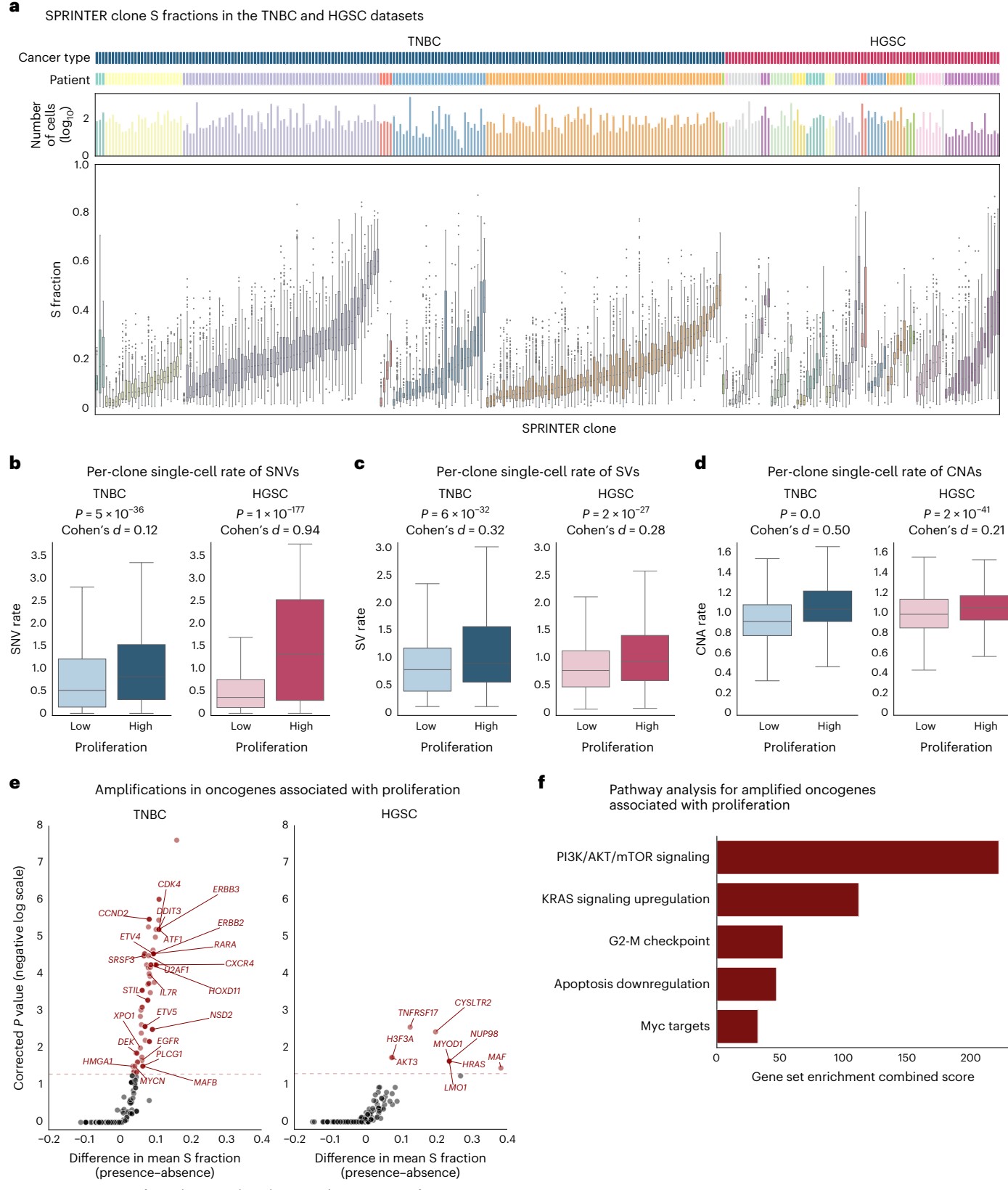

**a** SPRINTER clone S fractions in the TNBC and HGSC datasets

**b** Per-clone single-cell rate of SNVs

TNBC
$P = 5 \times 10^{-36}$
Cohen's $d = 0.12$

HGSC
$P = 1 \times 10^{-177}$
Cohen's $d = 0.94$

**c** Per-clone single-cell rate of SVs

TNBC
$P = 6 \times 10^{-32}$
Cohen's $d = 0.32$

HGSC
$P = 2 \times 10^{-27}$
Cohen's $d = 0.28$

**d** Per-clone single-cell rate of CNAs

TNBC
$P = 0.0$
Cohen's $d = 0.50$

HGSC
$P = 2 \times 10^{-41}$
Cohen's $d = 0.21$

**e** Amplifications in oncogenes associated with proliferation

● Significant (corrected $P$ value < 0.05)   ● Not significant

**f** Pathway analysis for amplified oncogenes associated with proliferation

replicating cells[17], providing a potential proxy for proliferation[4,6–8,11–13]. To realize this potential, we introduced the SPRINTER algorithm, a formal method to enable accurate identification and, especially, clone assignment of S- and G2-phase cells. We demonstrated SPRINTER's utility and

accuracy on ground truth datasets and validated results with multiple orthogonal analyses on primary and metastatic tumor samples.

Using a newly generated single-cell, longitudinal, primary-metastasis-matched NSCLC dataset, SPRINTER's results combined

**Fig. 5 | SPRINTER reveals increased single-cell rates of clone-specific genomic variants and enrichment for specific oncogene amplifications in TNBC and HGSC high-proliferation clones. a**, In 7 TNBC and 15 HGSC tumors (dark blue and dark pink in the first row with distinct tumors colored differently in the second row), the distribution of the S fraction (bottom, *y* axis) of each SPRINTER clone (*x* axis) with varying cell numbers (top, *y* axis in $\log_{10}$ scale) was calculated by bootstrapping (with 300 repeats) using the S-phase cells identified and assigned to clones by SPRINTER. **b**–**d**, Single-cell rates of clone-specific genomic variants were measured in individual cells (*y* axis, for 23,383 TNBC and 10,235 HGSC cells, excluding cells classified as outliers, tumors with single clones and cells without measured variants) for SNVs (**b**), SVs (**c**) and CNAs (**d**) in high- and low-proliferation clones (separated by the median of inferred S fractions, *x* axis) in the TNBC (left) and HGSC (right) datasets, with *P* values as measured by a one-sided Mann–Whitney *U* test and Cohen's *d* effect sizes shown. **e**, For

each known oncogene (dots, obtained from the COSMIC Cancer Gene Census excluding tumor suppressor genes), a one-sided Mann–Whitney *U* test was used to identify amplifications present in clones with significantly higher S fractions than other clones, with *P* values multiple hypothesis-corrected using the Benjamini–Hochberg method with family-wise error rate = 0.05 (*y* axis, negative log scale) and the related differences between the average S fractions (*x* axis) shown for each test. Genes passing the test (red, with the minimum corrected threshold indicated with the dotted line) are enriched in clones with increased proliferation, with genes relevant to cancer proliferation annotated. **f**, Cancer-relevant pathways (*y* axis) enriched for genes with amplifications significantly associated with high clone proliferation from **e** were identified using a gene set enrichment analysis (combined scores on *x* axis). In **a**–**d**, box plots show the median and the IQR with whiskers denoting values within 1.5 times the IQR from the first and third quartiles, respectively.

---

with metastatic evolutionary analysis suggest that high-proliferation clones within an individual tumor have increased metastatic seeding potential, that is, comprise the specific cancer cells more likely to metastasize. While consistent with the known link between proliferation and outcomes for distinct tumors in different patients[2–8], these clone-specific results were not necessarily expected based on previous studies suggesting that disseminating cells undergo epithelial–mesenchymal transition, associated with a more invasive but less proliferative phenotype[61–64]. Our results are consistent with high-proliferation clones undergoing epithelial–mesenchymal transition but then plastically returning to a proliferative state in a target organ. Furthermore, our results are consistent with the recent TRACERx[14,21] observation that metastatic seeding clones, despite being present in only some primary tumor regions, are highly expanded in those regions, which could be explained by the increased proliferation illustrated by SPRINTER. Because SPRINTER revealed that high-proliferation clones also shed more ctDNA, these results motivate the development of scalable precision-medicine approaches[25,26] (for example, liquid biopsies[32–34] or inexpensive methylation assays[65]) to predict the metastatic potential of different clones. Nonetheless, SPRINTER's results warrant careful ctDNA interpretation given that its prevalence does not only relate to clone volume but also clone proliferation. While these results were derived from an individual case, the cancer-agnostic and technology-independent nature of SPRINTER demonstrated here makes it applicable to the increasing number of different scDNA-seq datasets[13,17–20,28], allowing generalization of these findings.

SPRINTER's results lay the foundation for investigating the cellular and evolutionary mechanisms underlying cancer proliferation and progression in human tumors. Here we found that high-proliferation clones in TNBC and HGSC tumors have increased rates of multiple genomic variants (SNVs, SVs and CNAs), which might provide an evolutionary advantage. In fact, high-proliferation clones were associated with specific genetic alterations enriched in proliferation-related gene pathways, illustrating a possible mechanism driving clone proliferation. Beyond genetic mechanisms, SPRINTER's results also enable ART investigation in tumor clones. Given the established link between ART and both gene expression changes and epigenetic modifications[38,39], SPRINTER thus provides a way to investigate non-genetic evolutionary mechanisms driving cancer progression. For instance, in the NSCLC case, we did not identify genetic drivers unique to the most proliferative and disseminating metastatic clade, but we did identify a unique late-to-early ART event in *KRAS* associated with increased expression. To further these opportunities, SPRINTER's results can be leveraged to improve ART identification in individual cells, for which methods are being developed[28,66].

While SPRINTER establishes a general framework to enable the evolutionary and clone-specific analysis of S- and G2-phase cells in human tumors, there are opportunities for further improvement. For instance, G2 fraction estimates are expected to be less robust than S fraction estimates because G2-phase identification only relies on a

single signal (that is, total read counts). Incorporation of additional signals (for example, nuclear imaging) could thus improve G2-phase identification, further enhancing the analysis of the relative length of cell cycle phases in human tumors that we started to demonstrate here. Moreover, while SPRINTER provides high sensitivity for mid- and late-S-phase identification, early-S-phase identification remains limited. We expect that the generated ground truth datasets will support the development of related algorithmic improvements. Finally, we note that low scDNA-seq coverage prevents the comprehensive characterization of SNVs only present in individual cells, which would require deeper sequencing experiments.

In conclusion, SPRINTER enables the characterization of the evolutionary dynamics of proliferation and non-genetic alterations such as ART in distinct clones co-existing within a tumor. This provides the substrate for the next generation of cancer research studies that can jointly investigate the genetic and non-genetic mechanisms underlying clinically relevant cancer phenotypes, like metastatic potential.

## Online content

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

[1]Computational Cancer Genomics Research Group, University College London Cancer Institute, London, UK. [2]Cancer Research UK Lung Cancer Centre of Excellence, University College London Cancer Institute, London, UK. [3]Cancer Evolution and Genome Instability Laboratory, The Francis Crick Institute, London, UK. [4]University College London Hospitals, London, UK. [5]Genomics Science Technology Platform, The Francis Crick Institute, London, UK. [6]Cancer Metastasis Laboratory, University College London Cancer Institute, London, UK. [7]Department of Cellular Pathology, University College London Hospitals, London, UK. [8]Cancer Genome Evolution Research Group, University College London Cancer Institute, London, UK. [9]Department of Experimental Medicine, Sapienza University of Rome, Rome, Italy. [10]Flow Cytometry Science Technology Platform, The Francis Crick Institute, London, UK. [11]Department of Medical Physics and Biomedical Engineering, University College London, London, UK. [12]Department of Molecular Oncology, British Columbia Cancer Research Centre, Vancouver, British Columbia, Canada. [13]Department of Pathology and Laboratory Medicine, University of British Columbia, Vancouver, British Columbia, Canada. [14]These authors contributed equally: Olivia Lucas, Sophia Ward. [15]These authors jointly supervised this work: Nnennaya Kanu, Charles Swanton, Simone Zaccaria. *Lists of authors and their affiliations appear at the end of the paper. ✉e-mail: n.kanu@ucl.ac.uk; charles.swanton@crick.ac.uk; s.zaccaria@ucl.ac.uk

## TRACERx Consortium

Charles Swanton[2,3,4,15], Mariam Jamal-Hanjani[2,4,6], Nicholas McGranahan[2,8], Simone Zaccaria[1,2,15], Nnennaya Kanu[2,15], Olivia Lucas[1,2,3,4,14], Sophia Ward[2,3,5,14], Rija Zaidi[1,2], Abigail Bunkum[1,2,6], Alexander M. Frankell[2,3], David A. Moore[2,3,7], Wing Kin Liu[2,6], Emilia L. Lim[2,3], Sonya Hessey[1,2,4,6], Cristina Naceur-Lombardelli[2], Andrew Rowan[3] & Gary Royle[11]

Full lists of members and their affiliations appear in the Supplementary Information.

## PEACE Consortium

Charles Swanton[2,3,4,15], Mariam Jamal-Hanjani[2,4,6], Nicholas McGranahan[2,8], Simone Zaccaria[1,2,15], Nnennaya Kanu[2,15], Olivia Lucas[1,2,3,4,14], Sophia Ward[2,3,5,14], Rija Zaidi[1,2], Abigail Bunkum[1,2,6], Alexander M. Frankell[2,3], David A. Moore[2,3,7], Wing Kin Liu[2,6], Emilia L. Lim[2,3], Sonya Hessey[1,2,4,6], Cristina Naceur-Lombardelli[2] & Andrew Rowan[3]

Full lists of members and their affiliations appear in the Supplementary Information.

## Methods

### Clinical data, ethics and consent

The five primary tumor samples and five anatomically distinct metastatic samples analyzed in this study have been obtained from patient CRUKP9145 with NSCLC (Supplementary Note 7) from the TRACERx study[14,21] (https://clinicaltrials.gov/ct2/show/NCT01888601, approved by an independent Research Ethics Committee, 13/LO/1546) and the PEACE autopsy study[32] (https://clinicaltrials.gov/ct2/show/NCT03004755, approved by an independent Research Ethics Committee, 13/LO/0972). Ethical approvals for the research performed in this study and informed written consent were obtained as part of TRACERx and PEACE studies. Moreover, pathological assessment of Ki-67 and patient clinical imaging were obtained and processed according to the standards of these studies (Supplementary Notes 8 and 9). The other patient data analyzed in this study are publicly available with ethics and consent reported in the corresponding studies[19].

### The SPRINTER algorithm

We introduce the SPRINTER algorithm to identify and assign S- and G2-phase cells to distinct tumor clones using scDNA-seq data. Specifically, SPRINTER enables this goal in the following six main steps: (1) the computation of replication-specific DNA sequencing signals, (2) the replication-aware segmentation of the genome into likely CNA segments, (3) the high-sensitivity inference of S-phase cells, (4) the inference of distinct tumor clones as subpopulations of cells with different complements of CNAs, (5) the assignment of S-phase cells to the corresponding clone and (6) the inference of G2-phase cells for each clone. To do this, SPRINTER uses the following two inputs: the count of sequencing reads aligned to different genomic regions (or read counts) for each sequenced cell and replication scores (that is, a measure of the replication timing[38,44]) for each genomic region. We detail how SPRINTER sequentially performs each of these steps in the following sections.

**Deriving replication-specific DNA sequencing signals.** The first challenge addressed by SPRINTER is the computation of DNA sequencing signals that incorporate replication-specific information for the identification and clone assignment of S-phase cells. For each cell, given a partition of the reference genome into $m$ bins (50 kb by default), SPRINTER calculates two signals for each bin. The first signal is the replication timing, which is a classification of the bin as either early-replicating, late-replicating or unknown. In particular, SPRINTER computes the average replication scores per bin across a subset of available[38,44] Repli-Seq[45] datasets generated for different normal and cancer cell lines, and it identifies early/late bins as those that confidently belong to the two main modes of the replication score distribution (Supplementary Note 10). The remaining bins are classified as unknown and are only used in the CNA analysis of G0/G1/G2-phase cells. In fact, preserving >50% of bins with conserved early or late replication timing is sufficient for S-phase identification and clone assignment because CNAs are large (mostly >2 Mb (refs. 16–19,30); Supplementary Fig. 2), while replication fluctuations are substantially shorter (Supplementary Fig. 2) and occur across the whole genome (Extended Data Figs. 1–3).

The second signal calculated for each bin is the RDR, which is a signal used in standard copy-number analysis to identify CNAs[18]. Because different bins replicate their DNA at different times, the genome of S-phase cells is characterized by the alternation of replicated regions with higher read counts and unreplicated regions with lower read counts, inducing read count fluctuations across the whole genome[17,28]. To capture these fluctuations, SPRINTER calculates RDRs similarly to previous scDNA-seq methods[17–19] by aggregating read counts in windows of neighboring bins and applying standard normalizations for alignment bias (Supplementary Note 11).

Additionally, SPRINTER improves RDR calculation in two ways. First, during the identification of S-phase cells, SPRINTER only aggregates bins with the same replication timing and chooses the window size for each cell independently given a fixed value of average read counts per bin. This cell-specific choice is important as it accounts for the fact that cells in different cell cycle states yield different total read counts, and it hence allows SPRINTER to calculate RDRs with the same expected variance across cells, in contrast to previous studies[13,17–19,28] (Supplementary Note 11). Second, SPRINTER introduces a replication-aware method to correct RDRs for GC sequencing bias[29,67]. While previous methods correct GC bias in RDRs by fitting a function that models the relationship between RDRs and GC content (for example, using local regressions[17,19,29,67]), these approaches also lead to the erroneous correction of RDR fluctuations induced by replication in S-phase cells (Supplementary Fig. 5). This is because early-replicating genomic regions are GC enriched and late-replicating regions are GC depleted, and hence replication-induced fluctuations are identified as GC bias and erroneously corrected, discarding the main signal used to identify S-phase cells. To preserve replication fluctuations, SPRINTER leverages two key observations. First, groups of bins with the same replication timing (early or late) are less affected by replication fluctuations as they replicate at more similar times. Thus, SPRINTER infers GC biases in early and late bins separately using a quantile linear regression. Second, bins with higher GC content tend to replicate earlier than bins with lower GC content, and they produce increased RDRs during S phase (Supplementary Fig. 6). Thus, SPRINTER identifies the inferred regressions that are still affected by GC bias as those with an inferred slope substantially higher than other cells and corrects them, assuming that cells sequenced together have similar GC bias. Further details are given in Supplementary Note 12.

**Replication-aware copy-number segmentation.** The second challenge addressed by SPRINTER is the copy-number segmentation of the genome of each cell into groups of consecutive bins affected by the same CNAs. Inferring CNA segments is an essential task for the accurate identification of S-phase cells because CNAs induce similar RDR fluctuations (higher/lower RDRs for higher/lower copy numbers, respectively) to those induced by replication in S-phase cells (higher/lower RDRs for replicated/unreplicated bins, respectively; Extended Data Figs. 1–4). Therefore, identifying RDR fluctuations that are induced by CNAs with high confidence is essential for S-phase identification because fluctuations induced by replication are only present in S-phase cells. Previous methods to identify S-phase cells rely on standard algorithms for single-cell copy-number segmentation and adopt approaches that either ignore CNA fluctuations[17] or use CNA information that can be obtained by collapsing all cells together[28]. However, the former approach is not suited to the analysis of cancer cells with high rates of CNAs as in most solid tumors[15,17–20,30], and the latter is affected by intratumor heterogeneity and cell-unique CNAs, which are also frequent in most solid tumors[17–20].

SPRINTER overcomes the challenges of CNA segmentation in S-phase cells by introducing a replication-aware segmentation algorithm that leverages the expected differences between early- and late-replicating bins to only identify segments that are likely induced by CNAs. Specifically, SPRINTER separates early- and late-replicating bins into two groups and identifies candidate breakpoints for CNA segments in each group independently, such that most replication-induced fluctuations in RDRs between early and late bins that occur in S-phase cells are not erroneously inferred as CNA breakpoints. In each group, SPRINTER identifies breakpoints by using a hidden Markov model (HMM), similar to standard copy-number methods[17,19]. Because CNAs tend to affect large genomic segments[16–19,30] (that is, ~42 Mb on average with >99.9% of CNA segments >2 Mb in size, as measured in previous single-cell studies[19]) in contrast to the short length of consecutive regions of bins with the same replication timing (that is, <1 Mb on average with a median of 250 kb; Supplementary Figs. 1–3), CNAs are expected to induce segments containing both early- and

late-replicating bins. As such, SPRINTER obtains a cell-specific copy-number segmentation by combining all the identified breakpoints and preserving only related segments that include both early and late bins, corresponding to likely CNA segments. Instead, segments that only include either early or late bins are discarded because they are likely due to replication (that is, due to differences in RDRs between replicated and unreplicated bins with the same replication timing). Note that rare CNAs that exclusively overlap large domains of early/late regions can be correctly recovered in later SPRINTER steps—in G0/G1/G2-phase cells, all CNAs can be accurately inferred because all RDR fluctuations can be related to CNAs in these cells, while in S-phase cells, these rare CNAs can be later corrected using the CNAs inferred for the G0/G1/G2-phase cells assigned to the same clone. Further details are given in Supplementary Note 13.

**Identifying S-phase cells.** The third challenge addressed by SPRINTER is the identification of S-phase cells. Existing methods calculate a statistic per cell by combing multiple sequencing signals to identify replication-induced fluctuations and separate G0/G1/G2- and S-phase cells by using a single threshold after aggregating these statistics across all cells[13,17,28]. However, this approach has two main limitations that reduce its sensitivity. First, replication fluctuations are difficult to identify in cells during the early and late stages of S phase because only a limited fraction of bins is replicated or unreplicated during these stages, respectively (Extended Data Fig. 3 and Supplementary Figs. 11–13). Second, RDR fluctuations differ for cells with different ploidies because copy-number changes generally result in smaller fluctuations in the expected RDRs for cells with higher ploidies[18]. Therefore, approaches that rely on aggregating sequencing signals across all sequenced cells are not suited to accurately identify S-phase cells when sequencing mixtures of cells with different ploidies, for example, mixtures of diploid normal cells and aneuploid cancer cells that are often found in tumor samples.

To improve the sensitivity of previous approaches, SPRINTER leverages the expected RDR fluctuations between early- and late-replicating bins to introduce a statistical permutation test that can be applied to each cell independently. This test is based on the replication timing profile (RTP) of each cell, which is calculated by normalizing the RDRs of all bins within the same copy-number segment (inferred in the previous SPRINTER step) by their median to correct for the effect of CNAs (Extended Data Fig. 4 and Supplementary Fig. 7). Because the resulting RTP values only depend on the replication state of the corresponding bins with higher and lower values indicating replicated and unreplicated bins, respectively, varying RTP values across the genome are a hallmark of S-phase cells. Although the replication state of a bin is unknown, early bins are expected to replicate before late bins, and, hence, every S-phase cell is expected to have a subset of early-replicating bins with higher RTP values than late-replicating bins across the genome, or a subset of late-replicating bins with lower RTP values than early-replicating bins (Extended Data Fig. 3 and Supplementary Figs. 11–13).

As such, SPRINTER performs two permutation tests of replication timing classifications (by default $10^5$ permutations) to test the presence of such a subset containing a significantly high number of bins. Specifically, this is achieved by introducing a new summary statistic that captures the fraction of early or late bins with substantially higher or lower RTP values, respectively, than bins with different replication timing. Note that this statistic is expected to be robust to the presence of alterations or errors in replication timing classifications because it requires only a subset of bins, not all early or late bins, to display the expected difference in RTPs. Because replication fluctuations are expected to occur along the entire genome during the S phase, SPRINTER performs the test on each chromosome independently, and the resulting values of each statistic are combined using the harmonic mean; this approach helps overcome noise and errors that can be localized to

certain genomic regions. Finally, the two $P$ values obtained for each test are combined using the minimum, and a multiple-hypothesis correction is applied to all cells using the Holm–Šidák method to identify S-phase cells. In contrast to previous approaches that aggregate all sequenced cells together, SPRINTER's method is applied to each cell independently, providing a significance assessment for each cell individually and making the method suitable to heterogeneous tumor samples characterized by cells with different ploidies and CNA rates. Further details are given in Supplementary Note 14.

**Inferring distinct clones.** The fourth challenge addressed by SPRINTER is the inference of clones. Like previous single-cell studies[17–20], SPRINTER identifies CNAs in single cells, and, based on these, it infers clones as subpopulations of cells that share the same complement of CNAs. Because CNAs cannot be directly and easily inferred from the replication-influenced RDRs of S-phase cells, SPRINTER improves the inference of clones by using only the inferred G0/G1/G2-phase cells, under the realistic assumption that every clone contains corresponding G0/G1/G2-phase cells. Specifically, SPRINTER identifies CNAs in G0/G1/G2-phase cells by inferring the underlying copy numbers using an HMM that also incorporates the parameters inferred in the previous segmentation (Supplementary Note 15). Moreover, SPRINTER improves the inference of clones in two additional ways. First, SPRINTER introduces an auto-tuning clustering procedure to infer clones while automatically adapting to different rates of CNAs and errors in the inferred CNAs that can be present in distinct tumor samples, in contrast to previous clustering approaches with fixed parameters[18] (Supplementary Note 16). Second, SPRINTER introduces a hypothesis-testing approach to identify and correct artefactual clones derived from errors in the inferred ploidy of each cell (that is, mean copy number), which are frequent errors as shown in previous studies[18,19]. Specifically, SPRINTER tests if any clone inferred with different ploidy, that is, a ploidy different to most other tumor cells, can be equally explained by the ploidy and CNAs of other clones, and, if so, the clone is discarded and the corresponding cells are assigned to other clones while correcting their ploidy (Supplementary Note 17).

**Assigning S-phase cells to distinct clones.** The fifth challenge addressed by SPRINTER is the assignment of S-phase cells to the corresponding clone, as well as the inference of CNAs for these cells. While S-phase cells are expected to have the same set of CNAs as the G0/G1/G2-phase cells present in the same clone, CNAs cannot be directly inferred from their observed RDRs because RDRs are affected by both CNAs and replication fluctuations in S-phase cells, as described above. Furthermore, different S-phase cells can be affected by substantially different RDR fluctuations induced by replication. For example, RDR fluctuations are frequent across the entire genome in cells that are in mid-S phase, and these cells display the largest separation between the RDRs of early and late bins (Extended Data Fig. 3). Conversely, cells that are in early- or late-S phase might only display focal RDR fluctuations, which can be mistakenly identified as potential CNAs (Extended Data Fig. 3 and Supplementary Figs. 11–13). Consequently, every S-phase cell must be treated differently for CNA analysis.

To enable the accurate assignment of S-phase cells to clones, SPRINTER introduces a Bayesian, maximum-a-posteriori probability method with two steps applied to each S-phase cell independently. First, SPRINTER corrects replication-induced fluctuations by normalizing the RDRs of groups of bins within the same segment that have been previously inferred to have the same underlying replication state (that is, not separated by breakpoints inferred in SPRINTER's second step) around the median RDR of the segment (Extended Data Fig. 4 and Supplementary Fig. 7). Next, SPRINTER obtains the likelihood that each cell belongs to every clone by calculating the probability that the replication-corrected RDRs are generated by the copy numbers of the clone. Based on this, it thus assigns the cell to the clone that maximizes

the posterior probability, calculated using the likelihood and a prior probability that depends on the clone's size measured from the corresponding number of G0/G1/G2-phase cells. Further details are given in Supplementary Note 18.

In addition to clone assignments, SPRINTER also infers the CNAs of the identified S-phase cells using the same HMM algorithm described in SPRINTER's previous step but using the replication-corrected RDRs and additionally fixing the ploidy to be the same as the assigned clone. Moreover, SPRINTER uses the assigned clone to correct small, rare CNAs that exclusively occur in genomic regions with only early or late replication timing and other small CNAs in S-phase cells (Supplementary Note 19), allowing SPRINTER to accurately recover most CNAs in both S- and non-S-phase cells (Supplementary Fig. 21).

**Identifying G2-phase cells in distinct clones.** The sixth and last challenge addressed by SPRINTER is the identification of G2-phase cells in each inferred clone. Although G2-phase cells cannot be distinguished from G0/G1-phase cells solely based on RDRs (Supplementary Note 15), G2-phase cells are expected to yield higher total read counts than G1-phase cells due to increased DNA content, especially for tagmentation-based technologies such as DLP+ (ref. 17) (Supplementary Fig. 8). Based on this, SPRINTER introduces an importance sampling method to estimate the fraction $\mu$ of G2-phase cells in each clone by deconvolving the distributions of total read counts generated by either G0/G1- or G2-phase cells using a negative binomial mixture model. Additionally, the method integrates information from the identified S-phase cells—because G2-phase cells are also expected to yield higher read counts than S-phase cells on average (Supplementary Fig. 8), we constrain the inference of $\mu$ such that the resulting G2-phase cells have an expected read count higher than the expected read count of S-phase cells. As such, the probability of each cell being in G0/G1 or G2 phase is computed by using the likelihoods of the fitted model and a uniform prior, and G2-phase cells are defined as those with a probability below a certain threshold of being in G0/G1 phase (<0.3 by default). Further details are given in Supplementary Note 20.

**scDNA-seq**
We performed scDNA-seq on all cells from the HCT116 ground truth dataset and the NSCLC case using the DLP+ protocol as previously described[17,19]. Given that only snap-frozen patient tissue was available for this study, all HCT116 single cells and patient tissue samples underwent single nuclei isolation before DLP+ library preparation and sequencing. The details of the protocol are described in Supplementary Note 21.

**Ground truth dataset of cell cycle-sorted cells**
We generated a ground truth scDNA-seq dataset of 4,410 diploid and 4,434 tetraploid cells in known cell cycle phases sequenced using the DLP+ protocol. To avoid cross-contamination between cell cycle phases, known to be a common occurrence when using standard FACS techniques[28], we used an improved approach based on previous studies[31], which used two independent signals during FACS. The first is EdU, which is incorporated into actively replicating DNA and has been shown to accurately and comprehensively capture S-phase cells[31], and the second is DNA Hoechst 33342 dye, which is used to measure DNA content (Supplementary Fig. 9). To apply this approach, we chose the colorectal cancer cell line HCT116 as it provided an isogenic system that had already been analyzed in previous longitudinal studies[46] and enabled the generation of both diploid and tetraploid ground truth datasets[46]. Related details are given in Supplementary Note 22.

**Bioinformatics analysis of single-cell data**
The generated datasets were aligned to the human reference genome hg19 and processed using standard scDNA-seq pipelines, obtaining a single-cell pseudobulk BAM file for each sample, for which all cells have been sequenced together (see details in Supplementary Note 23).

SPRINTER was applied independently to each pseudobulk BAM file generated for the ground truth datasets and the NSCLC samples using default parameters (Supplementary Note 24). Moreover, SPRINTER was applied to the previous TNBC and HGSC datasets using the available read counts[19]. On the ground truth datasets, the previous methods for inferring S-phase cells, CCC and MAPD, were applied using and extending the available implementations[17,28] (Supplementary Note 24).

**Phylogenetic and metastatic seeding analysis**
We reconstructed the tumor phylogeny for the clones inferred by SPRINTER in the NSCLC dataset using both SNVs and CNAs. In particular, SNVs and related driver mutations were identified using a pseudobulk approach[17–19] and standard tools (Supplementary Note 25). While existing methods can reconstruct tumor phylogenies from single-cell SNVs[50], these methods cannot be directly applied to SPRINTER's clones due to the presence of subclonal SNVs, that is, SNVs that are only present in a subset of the cells within the same clone. Moreover, while methods to reconstruct tumor phylogenies from clone-specific CNAs[51] also exist, these methods do not integrate both SNVs and CNAs in the reconstruction of tumor phylogenies. Therefore, we devised a three-step approach to overcome these challenges by integrating and extending existing methods—(1) the presence of SNVs in each clone was inferred using pseudobulk approaches[17,18] per clone and probabilistic models of SNV cellular frequency[49], (2) SNV evolution was reconstructed using the HUNTRESS algorithm[50] and (3) CNA evolution was reconstructed using the MEDICC2 algorithm[51], fixing the same topology as the SNV phylogeny reconstructed in the previous step (Supplementary Notes 26–28). Based on this phylogeny, the MACHINA algorithm[48] was applied to infer metastatic migration patterns and identify seeding clones, which were also used to calculate the seeding genetic distances based on both SNVs and CNAs (Supplementary Note 29).

**Identifying clone-specific ART**
SPRINTER's results were leveraged to identify clone-specific ART for the tumor clones inferred in the NSCLC dataset with respect to the reference replication timing classifications obtained from normal cells, included as an additional feature in the SPRINTER algorithm. Specifically, SPRINTER analyzes each clone independently and, based on previous replication timing approaches[28,38], uses high and low average RTP values per clone (calculated as described above but using the segments induced by the inferred CNAs) to identify early- or late-replicating bins, respectively, similar to SPRINTER's first step. As such, ART is identified in genomic regions inferred with early or late replication timing, but which were classified as the opposite from the reference replication timing classifications obtained from normal cells only (see details in Supplementary Note 30).

To support the inferred ART classifications, two analyses were performed integrating matched bulk RNA sequencing data previously generated for regions of the same primary tumor[52]. This is because late-to-early and early-to-late ART are known to generally be associated with increased and decreased gene expression compared to normal tissue without ART, respectively[38,39]. First, a gene set variation analysis[68] was performed using GSEApy[69] with the inferred replication timing classifications, revealing enrichment scores that support the inferred ART. To show that these results are specific to this patient, we also showed that arbitrary scores are obtained from this analysis when using gene expression data from 915 tumor samples from 347 other TRACERx patients (Supplementary Fig. 43). Second, for a subset of ART specifically affecting genes known to be involved in cancer proliferation or metastatic potential, we performed a differential gene expression analysis using the same method as in previous TRACERx studies[52] based on DESeq2 (ref. 70). We compared the gene expression measured in the samples with a related ART event to the expression measured in different sets of other samples not expected to have the same ART event. Related details are given in Supplementary Note 31.

## Analysis of ctDNA

Four blood samples were collected, and ctDNA was processed in previous studies[32,33] for patient CRUKP9145. Tracked SNVs were matched to SPRINTER's identified single-cell clones using the reconstructed phylogeny, and, for each clone with tracked SNVs, a ctDNA shedding index at the primary tumor time point was calculated by either (1) subtracting the frequency of the SNVs (that is, cancer cell fractions) as measured by bulk or single-cell sequencing in the primary tumor from the frequency of the same SNVs measured in ctDNA samples by the ECLIPSE algorithm[33] or (2) subtracting the clone proportion (that is, the proportion of cells uniquely assigned to the clone) as measured in either bulk or single-cell sequencing in the primary tumor from the measured ctDNA clone proportion (measured by subtracting the SNV frequencies of different clones according to the ancestral relationships defined by the reconstructed phylogeny, as described in previous studies[14,48,49]). Further details are given in Supplementary Note 32.

## Rates of clone-specific genomic variants in individual cells

In the TNBC and HGSC datasets, the single-cell rates of clone-specific SNVs, SVs and CNAs in individual cells were calculated using the variants identified in previous studies[19] by normalizing the number of variants per cell by the total number of clonal (that is, present in all cells in the clone) or clone-unique variants for SNVs or SVs/CNAs, respectively. Moreover, all clones in either the TNBC or HGSC datasets have been partitioned into two groups of high or low proliferation based on the median of the inferred S fractions. Further details are given in Supplementary Note 33.

## Genomic alterations enriched in high-proliferation clones

In the TNBC and HGSC datasets, a hypothesis-testing approach has been used to identify amplifications of known oncogenes, deletions of known tumor suppressor genes and driver mutations enriched in high-proliferation clones. Specifically, for each of these identified events, a one-sided Mann–Whitney $U$ test has been performed comparing SPRINTER's inferred S fractions for clones without the event to the S fraction for clones harboring the event, and enriched events have been selected after applying a multiple-hypothesis correction using the Benjamini–Hochberg method. Finally, a gene set enrichment analysis[71] has been performed for the selected amplifications with GSEApy[69]. Related details are given in Supplementary Note 34.

## Investigating changes in the relative length of cell cycle phases

SPRINTER's estimated S and G2 fractions can provide information about changes in the relative length of different cell cycle phases that might occur in cancer[59]. While increased or decreased S fractions are generally expected to yield increased or decreased G2 fractions, respectively (because the presence of more/less S-phase cells generally determines if more/less cells enter G2 phase), an increase or decrease in the G2 fraction without a corresponding variation in the S fraction could indicate a change in G2 phase length relative to the length of S phase. We quantified these changes using the fraction of G2-phase cells over the fraction of S-phase cells (G2/S ratio), with a higher G2/S ratio consistent with a possible prolonged G2 phase relative to the length of the S phase.

## Statistics and reproducibility

All statistical analyses and tests in Results were performed in Python (v3.10.13) using Scipy[72] (v1.11.4) and are described in the corresponding sections or figure legends. The target number of cells sequenced per sample was chosen based on previous studies[18]. The number of samples has been chosen based on previous bulk analyses of the same tumor[14] and tissue availability, but no statistical methods were used to predetermine the number of samples.

## Reporting summary

Further information on research design is available in the Nature Portfolio Reporting Summary linked to this article.

## Data availability

Raw scDNA-seq data generated in this study from the ground truth datasets have been deposited at the National Center for Biotechnology Information (NCBI) Sequence Read Archive (SRA) under accession code PRJNA1158752. Raw scDNA-seq data generated in this study from the patient enrolled in the TRACERx and PEACE studies have been deposited at the European Genome–Phenome Archive (EGA) under accession code EGAD00001015411. Access is controlled by the TRACERx and PEACE data access committees, who assess whether the proposed research is allowed given patient consent and ethical approvals, as well as the scientific purpose. Details on how to apply for access are available on EGA. The processed data for the figures and analyses performed in this study are available in Zenodo at https://doi.org/10.5281/zenodo.13754278 (ref. 73). The processed data and related genomic variants from the previous TNBC and HGSC datasets are available in Zenodo at https://doi.org/10.5281/zenodo.6998936 (ref. 74) and https://doi.org/10.5281/zenodo.7718917 (ref. 75) as part of previous studies[19]. Raw scDNA-seq data generated in a previous study[28] from phase-sorted lymphoblastoid cells are available in SRA under accession code PRJNA770772.

## Code availability

SPRINTER is available on GitHub at https://github.com/zaccaria-lab/sprinter, and it is distributed through Bioconda[76]. A reproducible capsule of SPRINTER with data from this study is available on CodeOcean https://doi.org/10.24433/CO.4888914.v1.

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

## Acknowledgements

This study was funded by a Cancer Research UK (CRUK) Career Development Fellowship (RCCCDF-Nov21\100005 to S.Z.) and was further supported by the Rosetrees Trust (M917), a CRUK City of London Centre Award (C7893/A26233) and the Breast Cancer Research Foundation (BCRF; BCRF-22-157). This work uses data

provided by patients and collected by the National Health Service (NHS) as part of their care and support. The TRACERx study (ClinicalTrials.gov identifier: NCT01888601) is sponsored by University College London (UCL/12/0279). TRACERx is funded by CRUK (C11496/A17786) and coordinated through the CRUK and UCL Cancer Trials Centre, which has a core grant from CRUK (C444/A15953). The PEACE study (ClinicalTrials.gov Identifier: NCT03004755) is sponsored by University College London (UCL/13/0165). PEACE is funded by CRUK (C416/A21999) and coordinated through the CRUK and UCL Cancer Trials Centre. We gratefully acknowledge the patients and relatives who participated in the TRACERx study and the PEACE national autopsy program. We also thank the members of the TRACERx and PEACE consortia for participating in these studies, especially all site personnel, investigators and funders who supported the generation of the data within these studies. We acknowledge the support of staff at Scientific Computing, Genomics, Advanced Light Microscopy, Flow Cytometry and Cell Services Science Technology Platforms at the Francis Crick Institute. We also thank C. Charoy and M. Renshaw in the Advanced Light Microscopy Facility at the Francis Crick Institute for their support, expertise and implementation of the DLP+ nanochip imaging workflow. We thank J. Wang and J. Brimhall (at BC Cancer Research Centre) and E. Trinh (at the Genome Sciences Centre of the BC Cancer Research Centre) for their support in setting up the DLP+ sequencing protocol. We thank M. J. Williams, A. McPherson and S.P. Shah (at the Memorial Sloan Kettering Cancer Center) for their support in accessing the processed data of previous breast and ovarian datasets. We thank A. Huebner and L. Patruno (at the University College London Cancer Institute) for their support with the deposit of raw scDNA-seq data. We acknowledge that various figure schematics (Figs. 1, 2c and 4c, Extended Data Fig. 5 and Supplementary Figs. 9a, 24a, 25a, 26a and 27a) in this study were fully or in part created with BioRender.com. O.L. is supported by a Wellcome Trust Clinical Research Fellowship (225491/Z/22/Z). R.Z. is supported by a Medical Research Council Studentship Award (MR/N013867/1). A.B. is supported by a CRUK UCL Centre Non-Clinical Training Award (CANTAC721\100022). A.M.F. is supported by Stand Up To Cancer (SU2C-AACR-DT23-17). D.A.M. is supported by the CRUK Lung Cancer Centre of Excellence (C11496/A30025). M.S.H. is supported by CRUK (TRACERx—C11496/A17786). E.L.L. receives funding from the Novo Nordisk Foundation (ID 16584). S.H. is supported by a CRUK City of London Clinical Research Training Fellowship and receives funds from the Rosetrees Trust. H.Z. is supported by the China Scholarship Council for a 4-year PhD study and the BCRF. M.D. is supported by CRUK (C11496/A26311) and the Lung Cancer Centre of Excellence (30025). N.M. is a Sir Henry Dale Fellow, jointly funded by the Wellcome Trust and the Royal Society (211179/Z/18/Z), and receives funding from CRUK, the Rosetrees Trust and National Institute for Health and Care Research (NIHR) Biomedical Research Centre (BRC) at University College London Hospitals and the CRUK University College London Experimental Cancer Medicine Centre. M.J.-H. is a CRUK Career Establishment Awardee and has received funding from CRUK, the IASLC International Lung Cancer Foundation, the Lung Cancer Research Foundation, the Rosetrees Trust, the UK and Ireland Neuroendocrine Tumour Society (UKI NETs) and the NIHR University College London Hospitals Biomedical Research Centre. N.K. is supported by CRUK, the BCRF and the Rosetrees Trust. C.S. is a Royal Society Napier Research Professor (RSRP\R\210001) and is supported by the Francis Crick Institute, which receives its core funding from CRUK (CC2041), the UK Medical Research Council (CC2041) and the Wellcome Trust (CC2041). C.S. is funded by CRUK (TRACERx—C11496/A17786), PEACE (C416/A21999), CRUK Cancer Immunotherapy Catalyst Network, CRUK Lung Cancer Centre of Excellence (C11496/A30025), the Rosetrees Trust, Butterfield and Stoneygate Trusts, the Novo Nordisk Foundation (ID 16584), a Royal Society Professorship Enhancement Award (RP/EA/180007), the NIHR University College

London Hospitals Biomedical Research Centre, the CRUK–University College London Centre, the Experimental Cancer Medicine Centre, the BCRF and The Mark Foundation for Cancer Research Aspire Award (21-029-ASP). C.S. is in receipt of an ERC Advanced Grant (PROTEUS) from the European Research Council under the European Union's Horizon 2020 research and innovation program (835297). S.Z. is a CRUK Career Development Fellow (RCCCDF-Nov21\100005) and is further supported by the Rosetrees Trust (M917).

## Author contributions

O.L. developed and performed SPRINTER's analyses, coordinated computational, bioinformatics and clinical analyses, conducted benchmarking on ground truth datasets, performed data quality control and performed statistical analyses. S.W. coordinated sequencing analyses, extracted and isolated nuclei, established and performed scDNA-seq with the DLP+ protocol and performed microscopy imaging. R.Z. developed and performed phylogenetic analysis, performed mutational analysis and contributed to developing the SPRINTER algorithm. A.B. developed and performed metastatic seeding analysis, performed genetic distance analysis, and performed phylogenetic and evolutionary visualization. A.M.F. performed ctDNA analysis. D.A.M. performed pathological analysis and review. M.S.H. conducted bioinformatics processing of raw sequencing data. W.K.L., B.D. and G.R. performed patient imaging analysis. D.M. performed driver mutational analysis. E.L.L. assisted with bioinformatics processing of raw sequencing data and benchmarking on ground truth datasets. S.H. collated and reviewed clinical data and annotation. C.N.-L. processed autopsy tumor samples and performed related quality control. A.R. processed primary tumor samples and performed related quality control. S.P. performed flow cell sorting for ground truth datasets and related quality control. H.Z. and M.D. generated and processed replication timing scores. S.A. assisted with establishing the DLP+ protocol. N.M. assisted with bioinformatics and statistical analyses and provided feedback on the paper. M.J.-H. provided oversight of clinical analyses and annotations, and assisted with metastatic evolutionary and seeding analysis. N.K. coordinated and performed ground truth data generation, led and conducted the cell culturing and provided oversight of replication timing, cell cycle phase and gene analyses. C.S., M.J.-H. and N.M. coordinated and provided oversight of the clinical, bioinformatics and experimental analyses related to the TRACERx and PEACE studies. S.Z. provided oversight of computational, bioinformatics, evolutionary and statistical analyses. S.Z. and O.L. conceived, developed and implemented the SPRINTER algorithm. S.Z., C.S. and N.K. jointly designed and supervised the study. S.W., R.Z., A.B. and M.J.-H. contributed to writing the paper. O.L., S.Z., C.S. and N.K. wrote the paper.

## Competing interests

A.M.F. is a co-inventor on a patent application to determine methods and systems for tumor monitoring (PCT/EP2022/077987). D.A.M. reports speaker fees from AstraZeneca and Takeda; consultancy fees from AstraZeneca, Thermo Fisher, Takeda, Amgen, Janssen, MIM Software, Bristol Myers Squibb and Eli Lilly and has received educational support from Takeda and Amgen. S.A. is a founder and shareholder of GenomeTherapeutics and scientific advisor to Sangamo Therapeutics, the Institute of Cancer Research, London, and the New York Genome Center, NY. N.M. has stock options in and has consulted for Achilles Therapeutics; holds a European patent in determining HLA LOH (PCT/GB2018/052004) and is a co-inventor to a patent to identifying responders to cancer treatment (PCT/GB2018/051912). M.J.-H. has received funding from CRUK, the National Institutes of Health (NIH) National Cancer Institute, International Association for the Study of Lung Cancer (IASLC) Foundation, Lung Cancer Research Foundation, Rosetrees Trust, UKI NETs and NIHR;

has consulted for Astex Pharmaceutical and Achilles Therapeutics; is a member of the Achilles Therapeutics Scientific Advisory Board and Steering Committee and has received speaker honoraria from Pfizer, Astex Pharmaceuticals, Oslo Cancer Cluster, Bristol Myers Squibb and Genentech. M.J.-H. is listed as a co-inventor on a European patent application relating to methods to detect lung cancer (PCT/US2017/028013); this patent has been licensed to commercial entities, and, under terms of employment, M.J.-H. is due a share of any revenue generated from such license(s) and is also listed as a co-inventor on the GB priority patent application (GB2400424.4) with title—Treatment and Prevention of Lung Cancer. C.S. acknowledges grants from AstraZeneca, Boehringer-Ingelheim, Bristol Myers Squibb, Pfizer, Roche-Ventana, Invitae (previously Archer Dx—collaboration in minimal residual disease sequencing technologies), Ono Pharmaceutical and Personalis. He is the chief investigator for the AZ MeRmaiD 1 and 2 clinical trials and is the Steering Committee Chair. He is also the co-chief investigator of the NHS Galleri trial funded by GRAIL and a paid member of GRAIL's Scientific Advisory Board (SAB). He receives consultant fees from Achilles Therapeutics (also a SAB member), Bicycle Therapeutics (also a SAB member), Genentech, Medicxi, the China Innovation Centre of Roche (CICoR), formerly Roche Innovation Centre—Shanghai, Metabomed (until July 2022), Relay Therapeutics SAB member, Saga Diagnostics SAB member and the Sarah Cannon Research Institute. C.S. has received honoraria from Amgen, AstraZeneca, Bristol Myers Squibb, GlaxoSmithKline, Illumina, MSD, Novartis, Pfizer and Roche-Ventana. C.S. has previously held stock options in Apogen Biotechnologies and GRAIL; currently has stock options in Epic Bioscience, Bicycle Therapeutics and Relay Therapeutics; and has stock options and is cofounder of Achilles Therapeutics. C.S. declares a patent application for methods to lung cancer (PCT/US2017/028013), targeting neoantigens (PCT/EP2016/059401), identifying patent response to immune checkpoint blockade (PCT/EP2016/071471), methods for lung cancer detection (US20190106751A1), identifying patients who respond to cancer treatment (PCT/GB2018/051912), determining HLA LOH (PCT/GB2018/052004), predicting survival rates of patients with cancer (PCT/GB2020/050221), methods and systems for tumor monitoring (PCT/EP2022/077987). C.S. is an inventor on a European patent application (PCT/GB2017/053289) relating to assay technology to detect tumor recurrence. This patent has been licensed to a commercial entity, and, under their terms of employment, C.S. is due a revenue share of any revenue generated from such license(s). The other authors declare no competing interests.

## Additional information

**Extended data** is available for this paper at https://doi.org/10.1038/s41588-024-01989-z.

**Correspondence and requests for materials** should be addressed to Nnennaya Kanu, Charles Swanton or Simone Zaccaria.

**a** Diploid ground truth: **G1/2 phase cells**

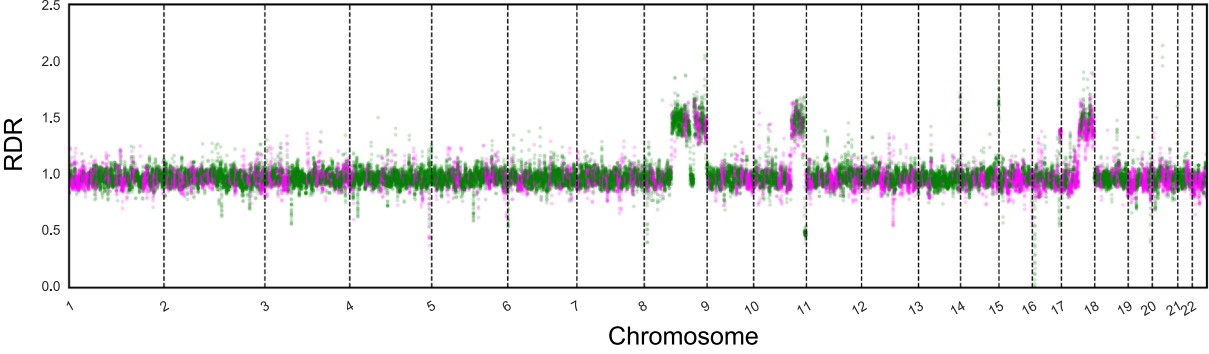

**b** Diploid ground truth: **mid S phase cells**

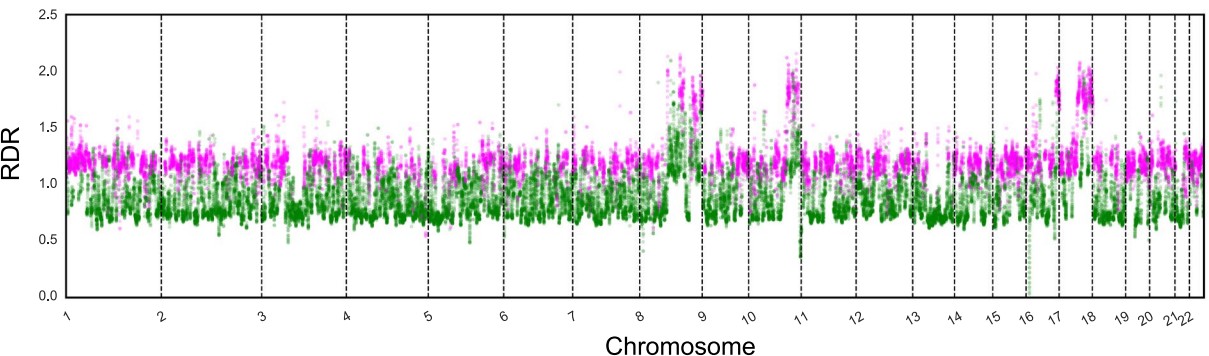

**c** Tetraploid ground truth: **G1/2 phase cells**

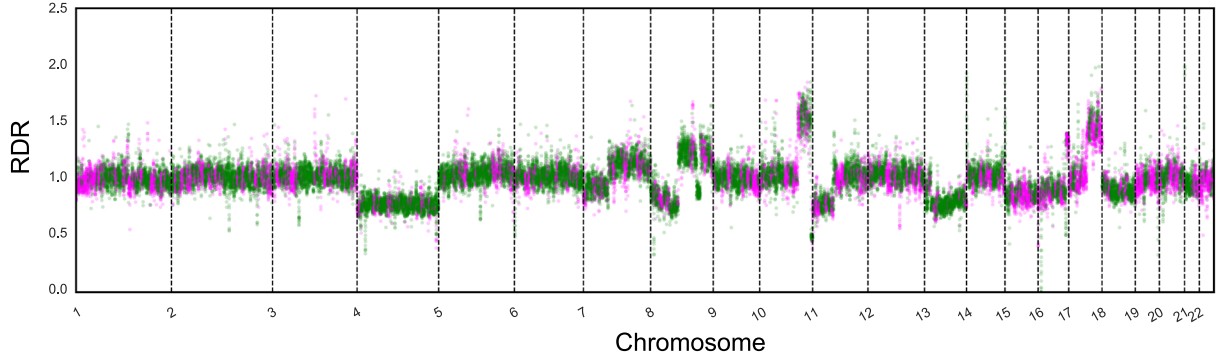

**d** Tetraploid ground truth: **mid S phase cells**

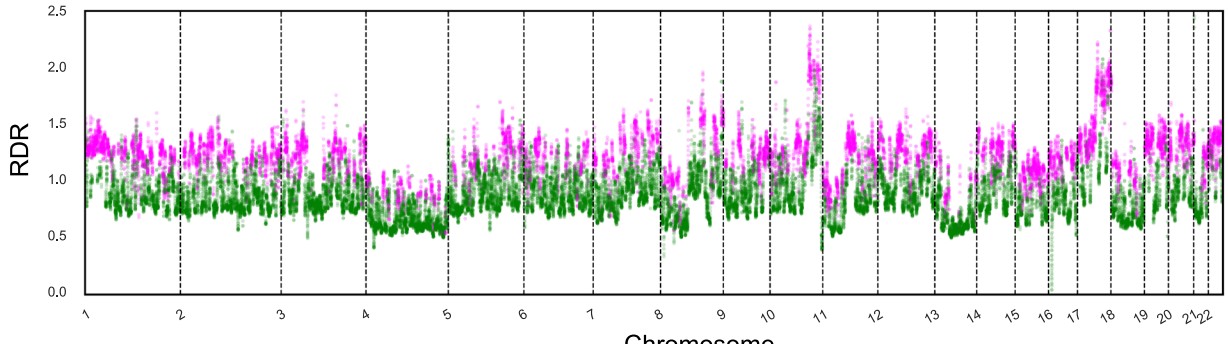

**Extended Data Fig. 1 | S-phase cells display a clear difference in read depth ratios (RDRs) between early and late genomic regions in contrast to G1/G2-phase cells.** Average RDRs (*y* axis) were measured by SPRINTER in 50 kb genomic bins with early (magenta) or late (green) replication timing across autosomes in the genome (*x* axis) in either the diploid (**a** and **b**) or tetraploid (**c** and **d**) ground truth datasets and across either G1/G2- (**a** and **c**) or mid-S-phase (**b** and **d**) cells (500 cells in each group).

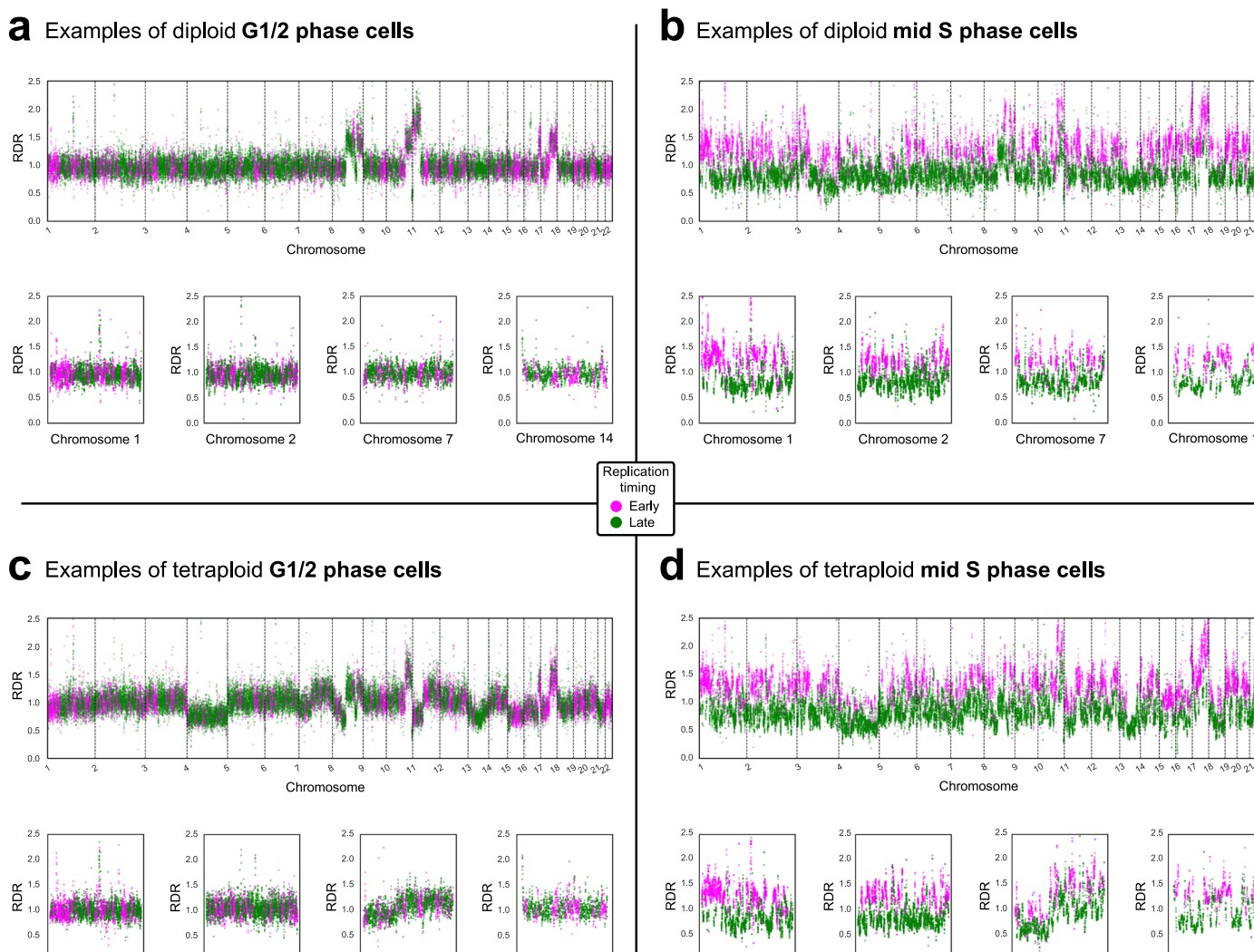

**Extended Data Fig. 2 | Early and late genomic regions are distributed across the genome and within chromosomes, displaying clear differences in read depth ratios (RDRs) between early and late genomic regions in S-phase cells in contrast to G1/G2-phase cells.** RDRs (*y* axis) were measured by SPRINTER in 50 kb genomic bins with early (magenta) or late (green) replication timing across autosomes in the genome (*x* axis in top) or in example chromosomes (bottom) for different examples of individual cells that belong to either the diploid (**a** and **b**) or tetraploid (**c** and **d**) ground truth datasets and are either in the G1/G2 (**a** and **c**) or S (**b** and **d**) phase of the cell cycle.

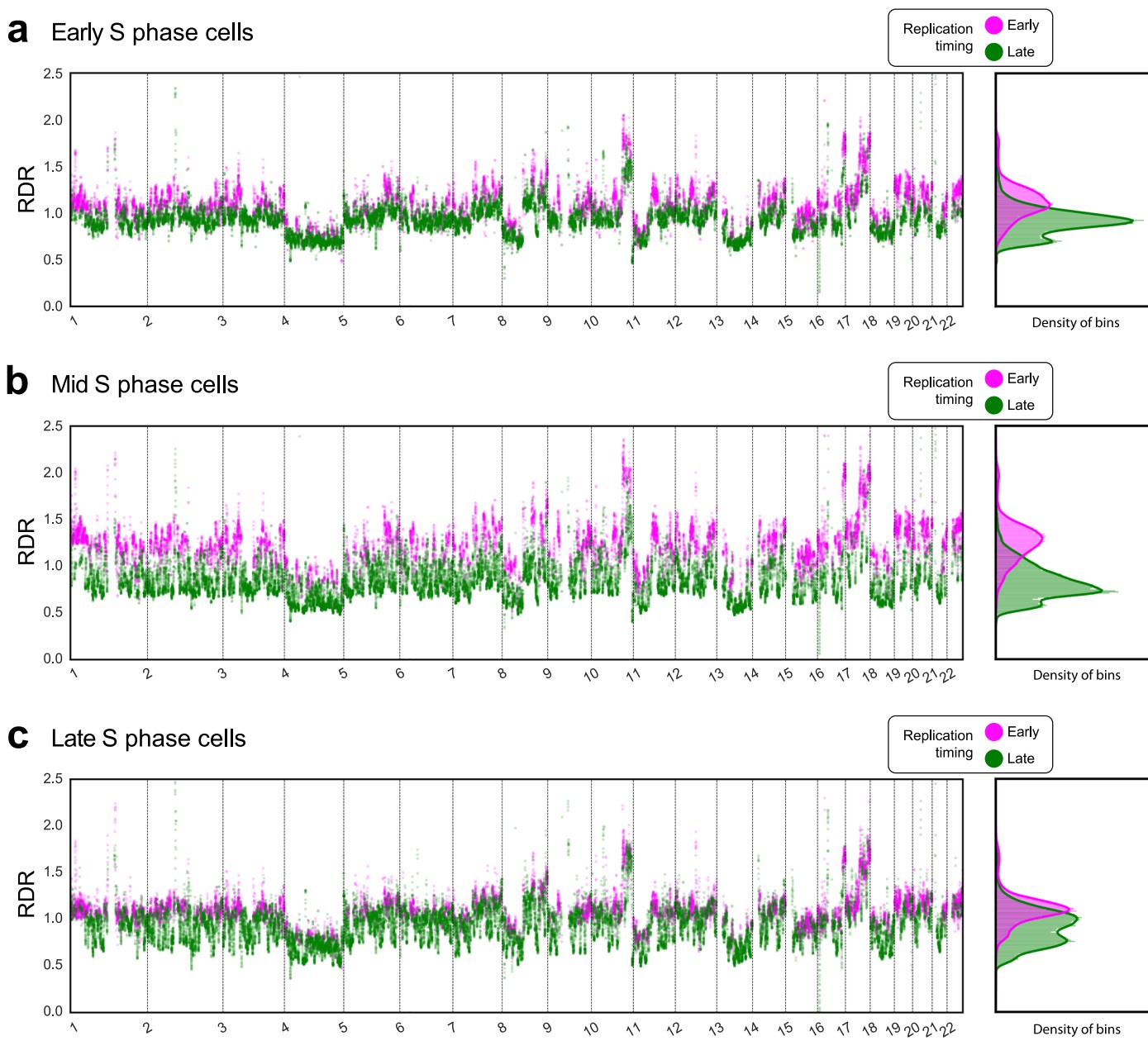

**Extended Data Fig. 3 | Cells at different stages of S phase display different replication-induced fluctuations of RDR.** Average RDRs (*y* axis) were measured by SPRINTER in 50 kb genomic bins with either early (magenta) or late (green) replication timing across autosomes in the genome (*x* axis) for (**a**) 180 early-S-phase cells, (**b**) 916 mid-S-phase cells and (**c**) 901 late-S-phase cells in the generated tetraploid ground truth dataset that were identified as S phase by SPRINTER. As expected, cells at different stages of S phase exhibit clearly different replication fluctuations in RDRs: in early-S phase only early-replicating bins shift to higher values of RDR, in mid-S phase all the early bins have completed replication and have distinctly higher values of RDR than late bins, and in late-S phase, late bins also start replicating and some of these bins increase their values of RDR.

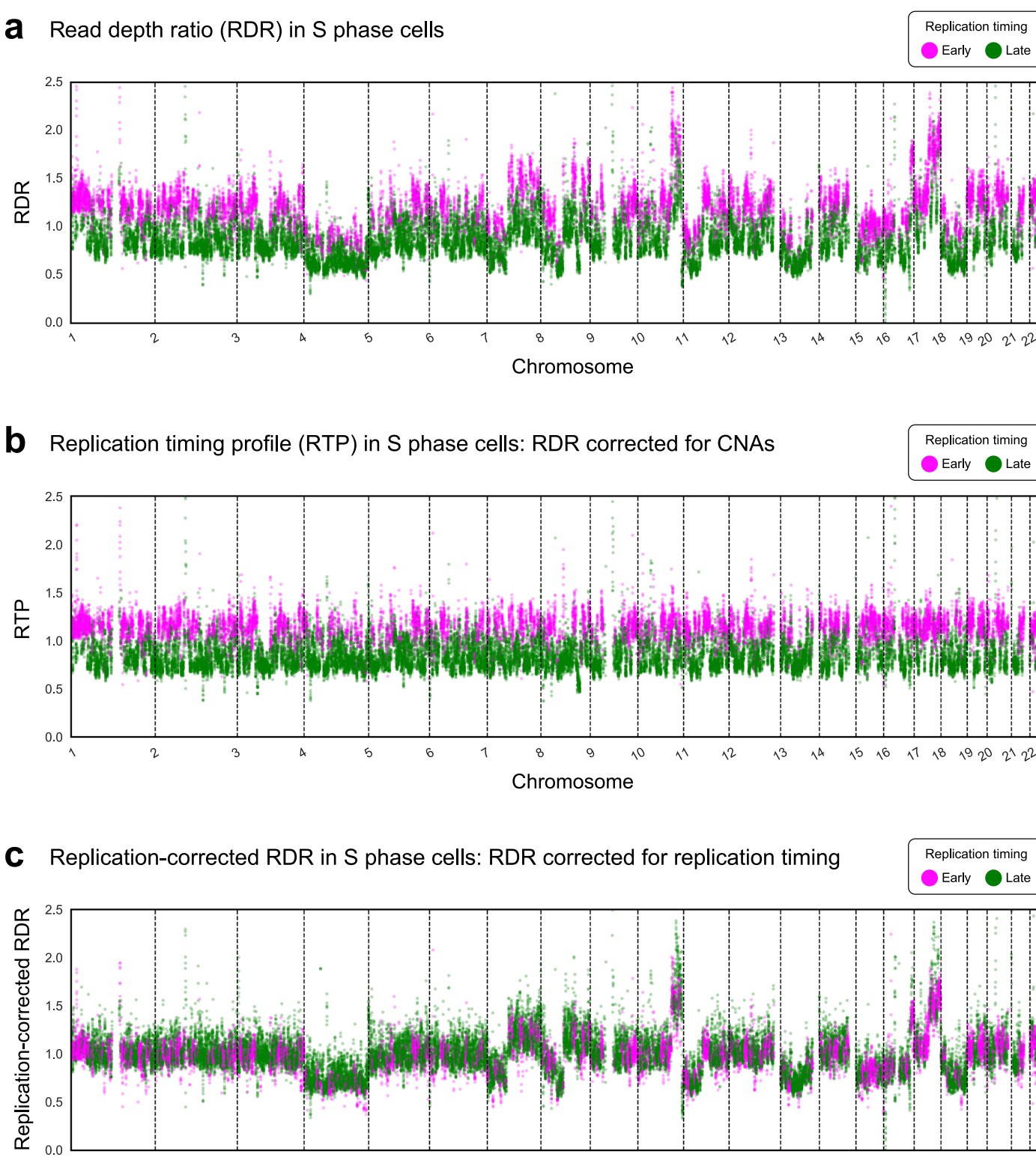

**Extended Data Fig. 4 | SPRINTER's replication-aware framework enables the differentiation of RDR fluctuations due to either replication or CNAs. a**, Average RDRs (*y* axis) were measured by SPRINTER in 50 kb genomic bins with either early (magenta) or late (green) replication timing across autosomes in the genome (*x* axis) for 73 mid-S-phase cells in the generated tetraploid ground truth dataset assigned to the same clone by SPRINTER. **b**, A replication timing profile (RTP, *y* axis) is calculated by SPRINTER for each bin (*x* axis) for the same cells by correcting RDRs for CNAs based on the copy-number segments inferred by SPRINTER, preserving clear fluctuations between bins with different replication timing (with magenta early regions having higher RDRs than green late regions on average). **c**, Replication-corrected RDRs (*y* axis) are computed by SPRINTER for each bin (*x* axis) for the same cells by correcting RDRs for replication fluctuations, such that the remaining fluctuations are likely due to CNAs and are not influenced by replication (in each segment there is no clear difference between bins with different replication timing).

**a**

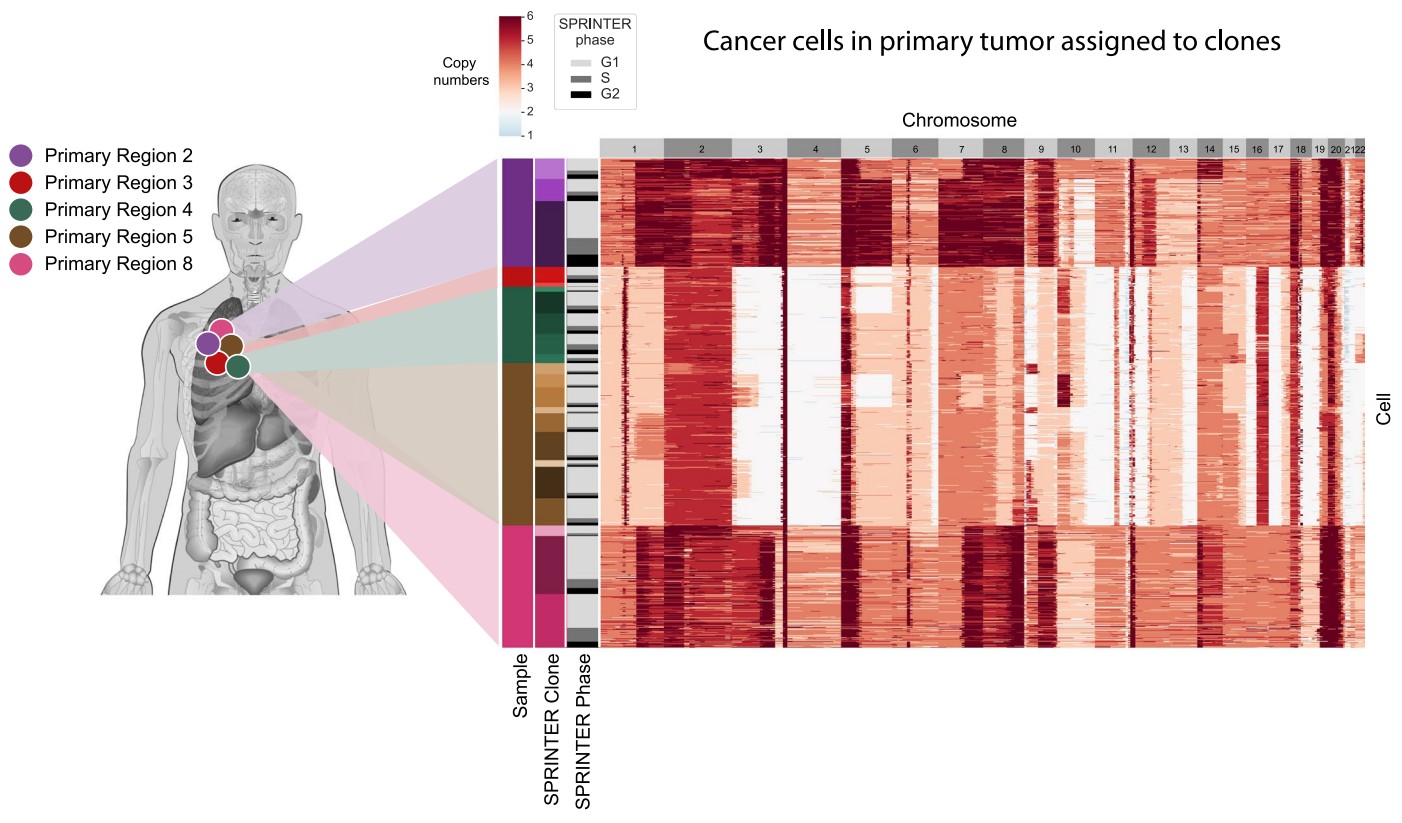

**b**

**Extended Data Fig. 5 | See next page for caption.**

**Extended Data Fig. 5 | SPRINTER's results for cells sequenced from five primary tumor samples and five metastases from patient CRUKP9145 with NSCLC.** Baseline copy numbers (heatmap colors) were inferred by SPRINTER on 7312 cancer cells assigned to clones by SPRINTER (rows, excluding normal cells and cells classified as outliers) sequenced from 10 distinct tumor samples (left bar), including (**a**) 4265 cells from five primary tumor samples obtained at surgery and (**b**) 3047 cells from five metastases sampled at autopsy, across ~1 Mb genomic bins (columns) with SPRINTER-inferred clones (middle bar) and with S- and G2-phase cells assigned to each corresponding clone (light gray for G1 phase, dark gray for S phase and black for G2 phase in right bar). The anatomical locations of the samples (colored circles) for (**a**) primary tumor regions and (**b**) metastases are displayed in corresponding body maps. The figure is created with BioRender.com.

## a  Tumor volume measured from CT and MR imaging

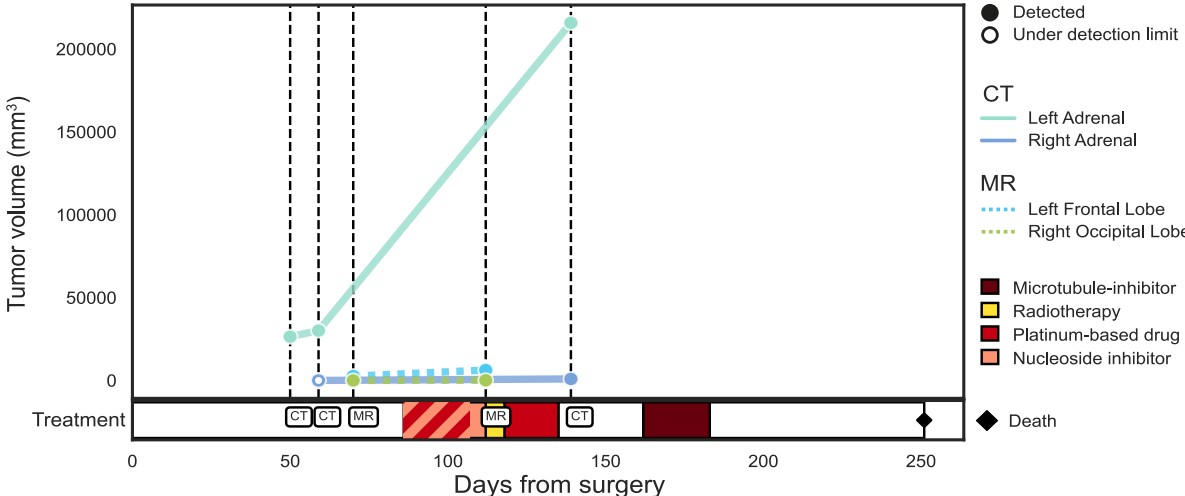

## b  Growth rate measured from CT and MR imaging

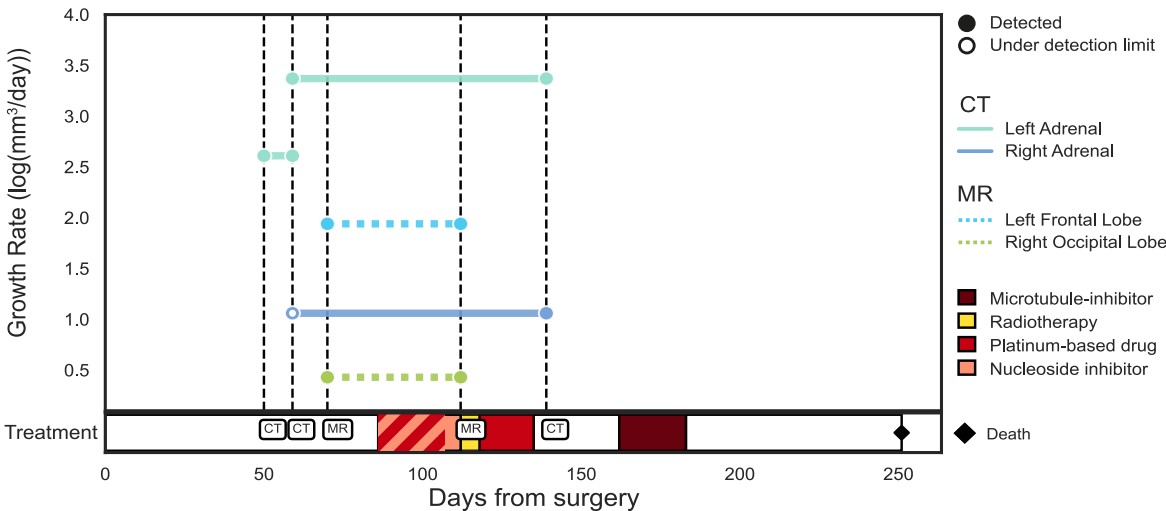

## c  CT and MR imaging

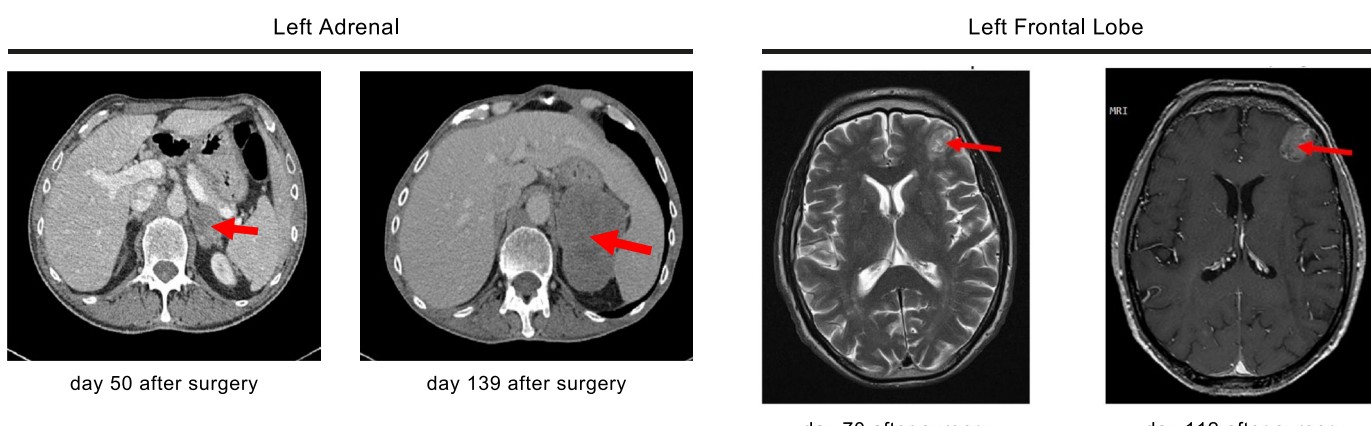

**Extended Data Fig. 6 | See next page for caption.**

**Extended Data Fig. 6 | Analysis of growth rates of metastases measured using serial clinical imaging for patient CRUKP9145.** Individual metastases were identified on computed tomography (CT) and magnetic resonance (MR) imaging scans performed during routine clinical management and collected as part of TRACERx. **a**, The volume of each metastasis (*y* axis, circle) was measured on serial scans (vertical dashed black lines) allowing changes in volume to be tracked over time (*x* axis). **b**, For each interval between two consecutive time points, the growth rate (log(mm³/day)) was calculated for each metastasis using either CT scans for the extra-cranial metastases (solid lines) or MR imaging scans for the brain metastases (dashed lines). For the right adrenal metastasis, which was only detected on the final CT scan (day 139 after surgery), the growth rate was calculated by assigning it a volume below the limit of CT detection on the preceding CT scan (day 59 after surgery, unfilled circle). **c**, Axial CT images of the left adrenal metastasis (red arrow, days 50 and 139 after surgery) and MR images of the left frontal lobe metastasis (red arrow, days 70 and 112 after surgery) are displayed.

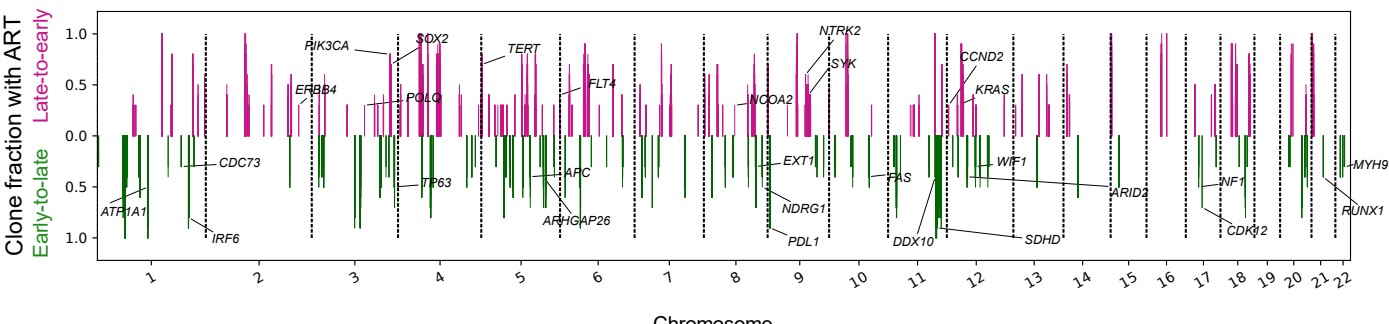

**Extended Data Fig. 7 | Clone-specific ART in the NSCLC dataset affects <10% of the genome on average as expected from previous studies.** The fraction of clones affected by ART was calculated by combining the fractions of clones affected across all samples (*y* axis) based on SPRINTER's clone-specific results in the NSCLC dataset for either late-to-early (positive values, dark magenta) or early-to-late (negative values, dark green) ART in 50 kb genomic bins along the genome (*x* axis, with autosomes separated by dashed lines). ART was inferred only in high-confidence cases (that is, only ART events that were present in most clones in >2 samples). Known cancer oncogenes in late-to-early genomic regions and known cancer tumor suppressor genes in early-to-late regions (from the COSMIC Cancer Gene Census) are annotated (black text and lines), also including tumor- and metastatic-clade-specific ART events affecting genes in the expression analysis (for example, *PDL1*, *CDK12*, *NCOA2* and *KRAS*).

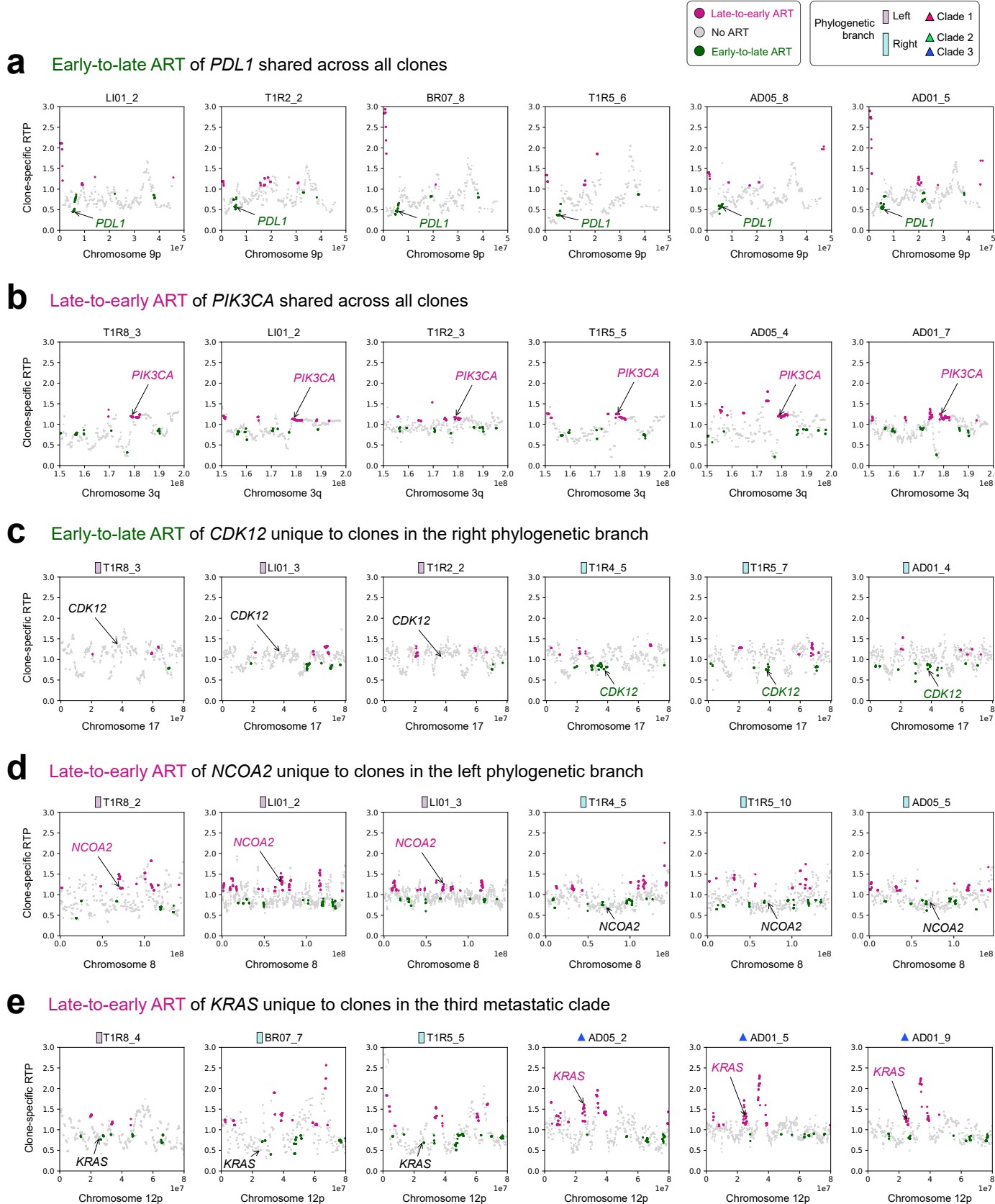

**Extended Data Fig. 8 | SPRINTER enables the identification of clone-specific ART supported by underlying read counts.** SPRINTER identifies different ART events affecting different genes (annotated text) and present in distinct clones (**a**–**e**) that belong to different phylogenetic branches (left and right indicated by lilac and light blue rectangles) or different metastatic clades (colored triangles). SPRINTER identifies clone-specific late-to-early (dark magenta) and early-to-late (dark green) ART events in genomic regions across chromosomes (*x* axis) if they have calculated values of the replication timing profile per clone (clone-specific RTP, *y* axis) that are higher or lower, respectively, than expected.

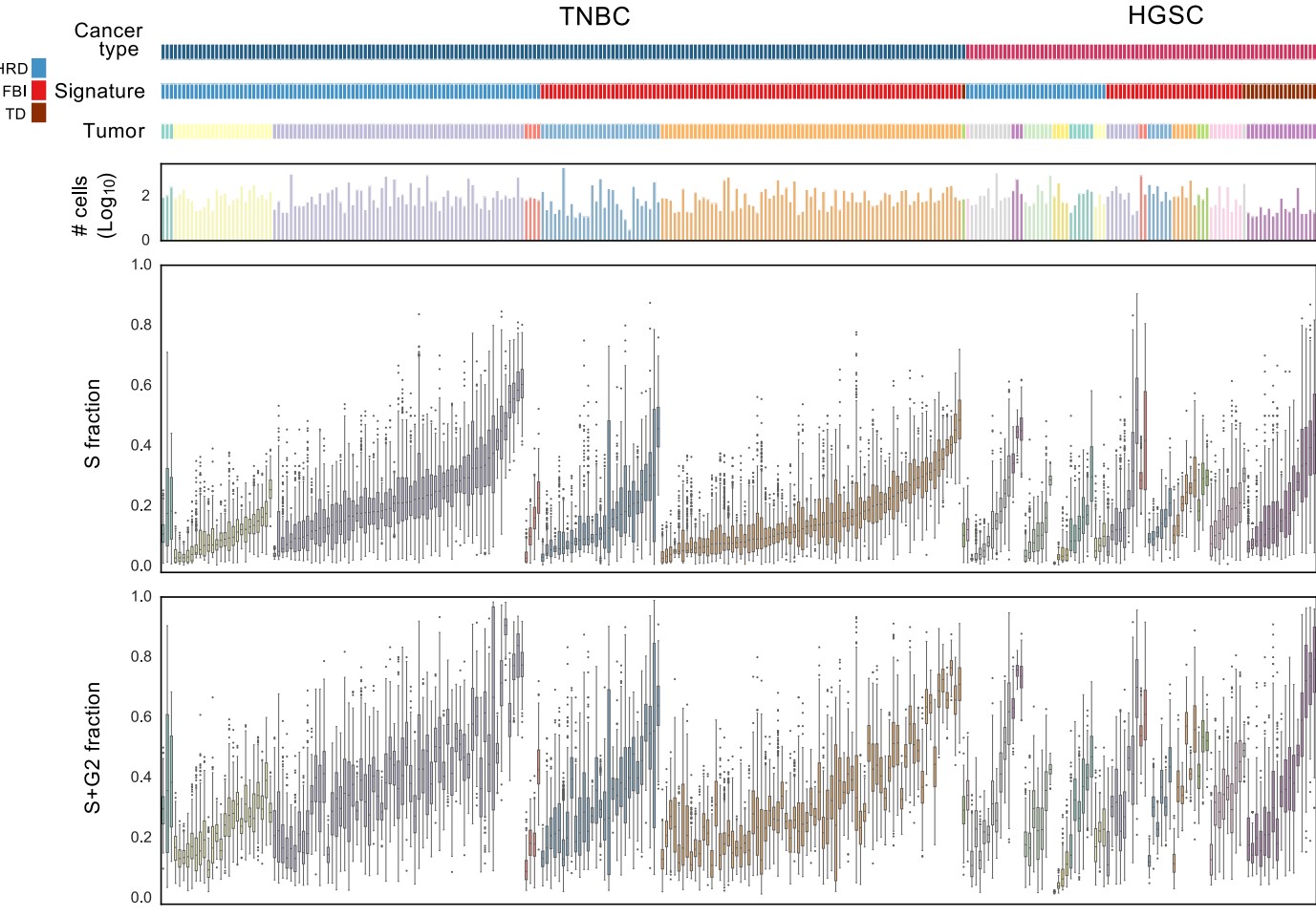

**Extended Data Fig. 9 | SPRINTER estimates clone-specific S and G2 fractions in previous TNBC and HGSC datasets.** For the TNBC and HGSC datasets (first row) with previously annotated genomic signatures (second row, with three signatures defined in the previous analysis of these datasets, that is, HRD, FBI and TD) and for each tumor in these datasets (third row), the distributions of the (middle) S fraction and (bottom) the fraction of actively replicating cells

(S + G2 fraction, y axis) for SPRINTER's inferred clones (x axis) were calculated by bootstrapping (per sample with 300 repeats) using the S- and G2-phase cells identified and assigned to clones by SPRINTER. Box plots show the median and the IQR, and the whiskers denote the lowest and highest values within 1.5 times the IQR from the first and third quartiles, respectively. HRD, homologous recombination deficiency; FBI, fold-back inversions; TD, tandem duplications.

## a   All clones

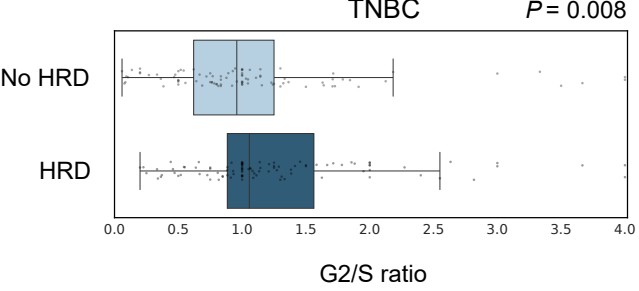
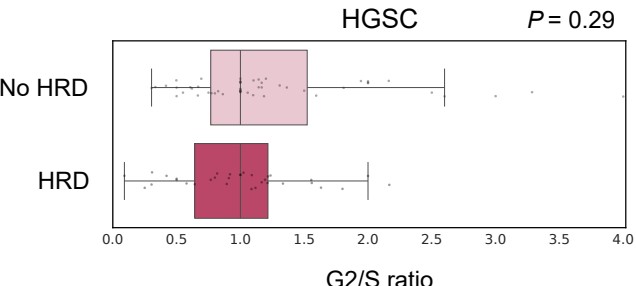

## b   Clones with >80 cells

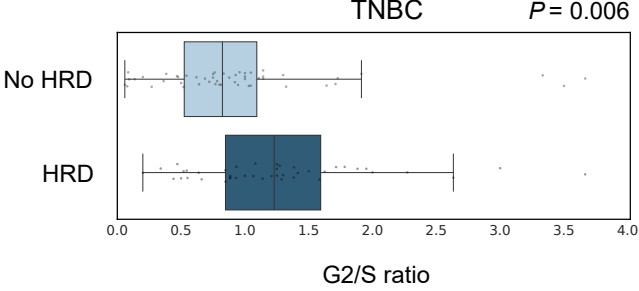
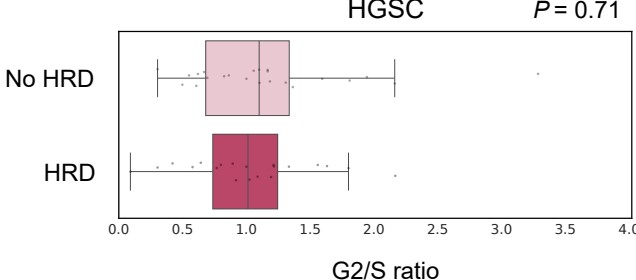

## c   Clones with >200 cells

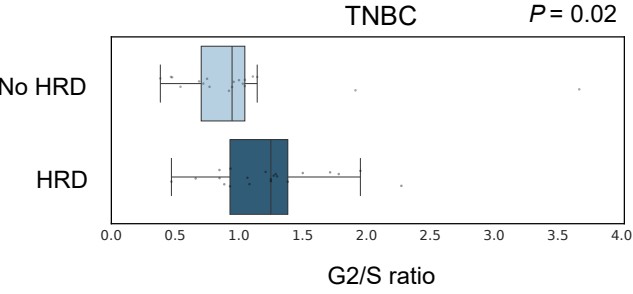
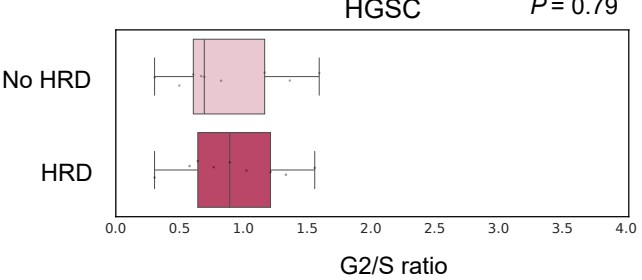

**Extended Data Fig. 10 | The G2/S ratio is significantly higher in breast cancer clones with HRD.** The G2/S ratio (*x* axis) was calculated based on the G2 and S fractions inferred by SPRINTER in the clones (dots) with or without HRD (*y* axis) in the TNBC (left) and HGSC (right) datasets, with *P* values as measured by a two-sided Mann–Whitney *U* test when considering (**a**) all 280 clones inferred by SPRINTER, (**b**) only the 137 clones with more than 80 cells and (**c**) only the 58 clones with more than 200 cells. In all panels, box plots show the median and the IQR, and the whiskers denote the lowest and highest values within 1.5 times the IQR from the first and third quartiles, respectively.

Charles Swanton
Simone Zaccaria

# Reporting Summary

## Statistics

For all statistical analyses, confirm that the following items are present in the figure legend, table legend, main text, or Methods section.

| n/a | Confirmed | |
|---|---|---|
| ☐ | ☒ | The exact sample size (*n*) for each experimental group/condition, given as a discrete number and unit of measurement |
| ☐ | ☒ | A statement on whether measurements were taken from distinct samples or whether the same sample was measured repeatedly |
| ☐ | ☒ | The statistical test(s) used AND whether they are one- or two-sided *Only common tests should be described solely by name; describe more complex techniques in the Methods section.* |
| ☐ | ☒ | A description of all covariates tested |
| ☐ | ☒ | A description of any assumptions or corrections, such as tests of normality and adjustment for multiple comparisons |
| ☐ | ☒ | A full description of the statistical parameters including central tendency (e.g. means) or other basic estimates (e.g. regression coefficient) AND variation (e.g. standard deviation) or associated estimates of uncertainty (e.g. confidence intervals) |
| ☐ | ☒ | For null hypothesis testing, the test statistic (e.g. $F$, $t$, $r$) with confidence intervals, effect sizes, degrees of freedom and $P$ value noted *Give P values as exact values whenever suitable.* |
| ☐ | ☒ | For Bayesian analysis, information on the choice of priors and Markov chain Monte Carlo settings |
| ☐ | ☒ | For hierarchical and complex designs, identification of the appropriate level for tests and full reporting of outcomes |
| ☐ | ☒ | Estimates of effect sizes (e.g. Cohen's *d*, Pearson's *r*), indicating how they were calculated |

*Our web collection on statistics for biologists contains articles on many of the points above.*

## Software and code

Policy information about availability of computer code

| Data collection | Quality control of raw sequencing data (FASTQ): |
|---|---|
| | FastQC (v0.11.8) |
| | FastQ Screen (v0.13.0, flags: --subset 100000; --aligner bowtie2) |
| | MultiQC (v1.10.1) |
| | |
| | Removing adapters: |
| | fastp (v0.20.0) |
| | |
| | Alignment to hg19 genome assembly: |
| | BWA-MEM (v0.7.17) |
| | |
| | Deduplication of sequencing reads: |
| | sambamba markdup (v0.7.0, flags: --remove-duplicates) |
| | |
| | Merging aligned sequencing reads: |
| | sambamba merge (v0.7.0) |
| | |
| | Quality control of aligned sequencing reads (BAM): |
| | Samtools (v1.9) |
| | Picard (v2.25.4) with MultiQC (v1.10.1) |

Counting single-cell sequencing reads:
CHISEL (v1.1.4)

Calling genomic variants:
Mutect2 (GATK, v4.2.0)

Variant annotation:
Ensembl Variant Effect Predictor (VEP, v109) with the plugins CADD (v16), LOFTEE, and SpliceAI
openCRAVAT34 (v2.3.0) with CHASMplus, CHASMplus LUAD, and CHASMplus LUSC modules

Data analysis

New method for identification and clone assignment of S and G2 phase cells:
SPRINTER (v1.0) available on GitHub at https://github.com/zaccaria-lab/sprinter with reproducible capsule linked to this manuscript available on CodeOcean

Existing methods for S phase identification:
cell cycle classifier (CCC) with HMMcopy (v0.6.46)
MAPD, available on GitHub at https://github.com/TheKorenLab/Single-cell-replication-timing (commit 4773a8f)

Phylogenetic analysis:
HUNTRESS (v0.1.2)
MEDICC2 (v1.0.2)

Metastatic dissemination analysis:
MACHINA (v1.2)

For manuscripts utilizing custom algorithms or software that are central to the research but not yet described in published literature, software must be made available to editors and reviewers. We strongly encourage code deposition in a community repository (e.g. GitHub). See the Nature Portfolio guidelines for submitting code & software for further information.

# Data

Policy information about availability of data

All manuscripts must include a data availability statement. This statement should provide the following information, where applicable:
- Accession codes, unique identifiers, or web links for publicly available datasets
- A description of any restrictions on data availability
- For clinical datasets or third party data, please ensure that the statement adheres to our policy

Raw scDNA-seq data generated in this study from the ground truth datasets have been deposited at the NCBI Sequence Read Archive (SRA) under accession code PRJNA1158752. Raw scDNA-seq data generated in this study from the patient enrolled in the TRACERx and PEACE studies have been deposited at the European Genome–phenome Archive (EGA) under accession code EGAD00001015411. Access is controlled by the TRACERx and PEACE data access committees, who assess whether the proposed research is allowed given patient consent and ethical approvals, as well as the scientific purpose. Details on how to apply for access are available on EGA. The processed data for the figures and analyses performed in this study are available in Zenodo at https://doi.org/10.5281/zenodo.13754278. The processed data and related genomic variants from the previous TNBC and HGSC datasets are available in Zenodo at https://doi.org/10.5281/zenodo.6998936 and https://doi.org/10.5281/zenodo.7718917 as part of previous studies. Raw scDNA-seq data generated in a previous study from phase-sorted lymphoblastoid cells are available in SRA under accession code PRJNA770772.

# Research involving human participants, their data, or biological material

Policy information about studies with human participants or human data. See also policy information about sex, gender (identity/presentation), and sexual orientation and race, ethnicity and racism.

| | |
|---|---|
| Reporting on sex and gender | Done as part of the previous TRACERx study (https://doi.org/10.1038/s41586-023-05783-5). |
| Reporting on race, ethnicity, or other socially relevant groupings | Done as part of the previous TRACERx study (https://doi.org/10.1038/s41586-023-05783-5). |
| Population characteristics | The patient was a 60-year-old male with stage IIIA squamous cell carcinoma, who was part of the TRACERx study and underwent surgical removal of the primary tumour and who subsequently relapsed and died 251 days later after receiving multiple lines of chemotherapy and radiotherapy. The patient died with metastases in multiple anatomical sites and was enrolled in the PEACE autopsy programme, through which a post-mortem examination was performed. |
| Recruitment | Done as part of the previous TRACERx study (https://doi.org/10.1038/s41586-023-05783-5). |
| Ethics oversight | Done as part of the previous TRACERx study (https://doi.org/10.1038/s41586-023-05783-5) and PEACE autopsy programme (detailed information are reported in Methods). |

Note that full information on the approval of the study protocol must also be provided in the manuscript.

# Field-specific reporting

Please select the one below that is the best fit for your research. If you are not sure, read the appropriate sections before making your selection.

☒ Life sciences ☐ Behavioural & social sciences ☐ Ecological, evolutionary & environmental sciences

For a reference copy of the document with all sections, see nature.com/documents/nr-reporting-summary-flat.pdf

# Life sciences study design

All studies must disclose on these points even when the disclosure is negative.

| | |
|---|---|
| Sample size | The number of primary tumour and metastatic samples was chosen based on tissue availability from the related TRACERx study and PEACE autopsy programme. The number of cells sequenced per sample (2000-2500) was chosen based on previous power studies for scDNA-seq data (https://doi.org/10.1038/s41587-020-0661-6). |
| Data exclusions | Cells with less than 100,000 sequencing reads have been excluded from downstream analysis in this study because this low number of sequencing reads was insufficient for copy-number analysis and may indicate failures in the process of DNA library preparation as previously reported. |
| Replication | For reproducibility, the DNA sequencing reads of every cell, as well as the SPRINTER code and related guided demos to reproduce the results, will be made publicly available after review. Currently, an automatic reproducible capsule for SPRINTER's results is available in CodeOcean at: https://codeocean.com/capsule/9392115. The capsule can be accessed to review and verify previous automatically and independently tested executions of SPRINTER and re-execute it. The processed results to reproduce every figure and downstream analysis in the manuscript, including related demos, will be made publicly available in Zenodo. |
| Randomization | Randomization is not relevant as this is an observational study. |
| Blinding | Blinding is not relevant as this is an observational study. |

# Reporting for specific materials, systems and methods

We require information from authors about some types of materials, experimental systems and methods used in many studies. Here, indicate whether each material, system or method listed is relevant to your study. If you are not sure if a list item applies to your research, read the appropriate section before selecting a response.

### Materials & experimental systems

| n/a | Involved in the study |
|---|---|
| ☒ | ☐ Antibodies |
| ☒ | ☐ Eukaryotic cell lines |
| ☒ | ☐ Palaeontology and archaeology |
| ☒ | ☐ Animals and other organisms |
| ☒ | ☐ Clinical data |
| ☒ | ☐ Dual use research of concern |
| ☒ | ☐ Plants |

### Methods

| n/a | Involved in the study |
|---|---|
| ☒ | ☐ ChIP-seq |
| ☐ | ☒ Flow cytometry |
| ☒ | ☐ MRI-based neuroimaging |

## Flow Cytometry

### Plots

Confirm that:

☒ The axis labels state the marker and fluorochrome used (e.g. CD4-FITC).

☒ The axis scales are clearly visible. Include numbers along axes only for bottom left plot of group (a 'group' is an analysis of identical markers).

☒ All plots are contour plots with outliers or pseudocolor plots.

☒ A numerical value for number of cells or percentage (with statistics) is provided.

### Methodology

| | |
|---|---|
| Sample preparation | Flow Cytometry was used for generation of the ground truth datasets using the colorectal cell line HCT116, using one diploid and one tetraploid lineage. In detail, we first labelled cells with Click-iT EdU and fixed and stained using the Click-iT Plus EdU Flow Cytometry Assay Kit (C10634 Invitrogen), halting further progression through the cell cycle. Cells were stained with 2ug/ml Hoechst 33342 before flow sorting. |

| Instrument | Flow sorting was performed on a BD Influx cell sorter (BD, San Jose, CA, USA) using a 140 micron nozzle, with pressure maintained at 14 psi. |
|---|---|
| Software | Data was analysed using BD FACS Software v1.2.0.142 (BD, San Jose, CA, USA). |
| Cell population abundance | Cells were simultaneously and electrostatically sorted into 5 uniform fractions of different cell cycle phases (G1, early S, mid S, late S, and G2). |
| Gating strategy | Cells were simultaneously and electrostatically sorted based on both EdU (Alexa Fluor 647, excited with a 642nm laser and emission collected in a 670/30BP filter) and DNA Hoechst 33342 dye (excited using a 405nm laser and emission collected in a 460/50BP filter), with both parameters displayed on a linear scale. |

☒ Tick this box to confirm that a figure exemplifying the gating strategy is provided in the Supplementary Information.

