## [Peer Review File · Nature Genetics]

Peer Review Information

Manuscript Title: Characterising the evolutionary dynamics of cancer proliferation in single-cell clones

Corresponding author name(s): Dr Simone Zaccaria, Dr Nnennaya Kanu, Professor Charles Swanton

Reviewer Comments & Decisions:

Decision Letter, initial version:

23rd Nov 2023

Dear Dr Zaccaria,

Your Article, "Linking proliferation rate to the evolution of single-cell primary and metastatic tumour clones" has now been seen by 3 referees. You will see from their comments copied below that while they find your work of considerable potential interest, they have raised quite substantial concerns that must be addressed. In light of these comments, we cannot accept the manuscript for publication, but would be very interested in considering a revised version that addresses these serious concerns.

We hope you will find the referees' comments useful as you decide how to proceed. If you wish to submit a substantially revised manuscript, please bear in mind that we will be reluctant to approach the referees again in the absence of major revisions.

If you choose to revise your manuscript taking into account all reviewer and editor comments, please highlight all changes in the manuscript text file. At this stage we will need you to upload a copy of the manuscript in MS Word .docx or similar editable format.

*2) If you have not done so already please begin to revise your manuscript so that it conforms to our Article format instructions, available here. Refer also to any guidelines provided in this letter.

Please be aware of our guidelines on digital image standards.

[redacted]

If you wish to submit a suitably revised manuscript we would hope to receive it within 6 months. If you cannot send it within this time, please let us know. We will be happy to consider your revision so long as nothing similar has been accepted for publication at Nature Genetics or published elsewhere. Should your manuscript be substantially delayed without notifying us in advance and your article is eventually published, the received date would be that of the revised, not the original, version.

Thank you for the opportunity to review your work.

Sincerely,

Safia Danovi
Editor
Nature Genetics

Referee expertise:

Referee #1: single cell genomics, cancer evo, methods

Referee #2: replication timing

Referee #3: replication timing, epigenetics, cancer genomics

Reviewers' Comments:

Reviewer #1:

Remarks to the Author:

In this manuscript, the authors present SPRINTER, an algorithm aimed at separating pseudo CNA due to S phase vs true genome CNA. They start by training their algorithm using cells with known cell cycle phase using EdU labelling and sorting of cells in specific phases of the cell cycle, followed by DLP+ sequencing. Finally, they show that they can identify different clonal populations with different phases of cell cycle within multiple tumor nodules isolated from the same patients.

The manuscript is well written and their ability to identify S phase cells is convincing. The ability to identify CNA and cell cycle phase in the same cell is also an advancement. However, there are some important limitations to the study that are not adequately addressed.

1) DLP+ only captures a fraction of the genome, and we don't know how deeply these cells were actually sequenced. The authors should include some important performance data to adequately assess the method. For example: were all cells used, and if some cells were thrown out, what percentage? What was the average fraction of the genome covered per single cell genome and the corresponding estimated CNV resolution based on that coverage? At what CNV resolution does their algorithm have difficulty discerning CNV from S phase changes?

2) The authors are really performing the construction of phylogenetic trees based on pseudobulk measurements, not at single-cell resolution. They would need much more complete genomes to do that. Depending on CNV resolution, as well as genome loci and allele dropout for SNV/Indel calling, they will be missing CNV and SNV data at single cell resolution. This will be especially problematic for variants that are present in a small number of cells, which will be excluded when performing pseudobulk calling. It is important that they highlight this limitation, as some of their biological conclusions may in fact be due to higher resolution genomic variation they are not capturing. For example, a proliferation-associated mutation with their current clones could actually explain the S phase percentage within the pseudobulk clone.

3) Can the authors estimate the fraction of clones they are actually capturing in a given sample based on the number of cells they sequence, their sensitivity, dropout, etc? This is important, as comparisons between lesions, such as comparing the surgical sample to autopsy could just be a sampling issue.

4) The movement of clones within the same patient is interesting to see. That has been done with bulk sequencing, but the authors are also able to add cell cycle status on top of that. Although interesting,

it does not provide much biological or clinical insight. Do the authors have any other observations where adding cell cycle provides new biological or clinical insights?

Reviewer #2:

Remarks to the Author:

This beautiful manuscript by Lucas et al utilizes large-scale scDNA-seq of cancer samples to identify cells in S phase and other cell cycle phases alongside CNAs, thus enabling the inference of proliferation rates of cancer subclones. This is an important problem and a powerful solution based on single cell sequencing and computational identification of subclones and the cell cycle phases of individual cells. The authors apply this approach to interesting questions regarding tumor evolution and metastasis.

There are several limitations that should be accounted for:

1. Reference replication profiles.

The crux of the method is to use reference replication profiles to identify cells in S phase. The authors justify this based on the conservation of replication profiles across samples, however the replication timing data they use is predominantly derived from non-cancer samples, cancer cell lines, and a small number of primary cancer samples. How can we know that replication profiles won't be different in actual cancer samples across patients, especially in subclones with potential genetic and/or epigenetic alterations that could impinge on DNA replication?

Sup fig 2 shows the average of several cell lines in each panel, so any differences would be masked out. A correlation matrix including all samples would be informative to include as well.

Ultimately this approach would be useful if it can be reliably applied to individual patients. The current use of averages of several samples and of reference replication profiles therefore does not fully demonstrate that this method is reliable enough to be applied on individual cancer samples and subclones, especially considering the genomic and epigenomic heterogeneity of different tumors.

2. scDNA-seq data quality.

Data in Figure 1 is shown at the whole-genome level. It would be useful to also present a zoomed-in view of a single chromosome or part of a chromosome to assist in evaluating the data. However, it seems from the figure that the ability to separate the genome to early and late replicating regions is very coarse, with very long areas sometimes spanning entire chromosome arms or even whole chromosomes being designated as "early" or "late". Figure 2g has a bit more resolution but also leads to the same conclusion. This stands in contradiction to the authors' own claim that replication timing fluctuates at the scale of ~1Mb and raises concerns about the accuracy of separating replication timing from CNAs and about the data quality in general.

If the scDNA-seq data was of high quality, I would expect the copy number fluctuations in S phase cells to be self-evident. The need to utilize reference replication timing profiles raises further concern that the data is of moderate quality; the use of external data only partially compensates for that and, while it provides a means of classifying S phase cells (and is the reason for the reported superiority over previous methods that called S phase cells *ab initio*), in certain circumstances may be suboptimal, as explained above.

3. Success of SPRINTER and comparison to other methods.

The comparison to CCC and MAPD is somewhat of an apples and oranges comparison. I agree that the use of reference replication profiles would usually provide a superior means to call S phase cells (and supports the premise of this paper), however CCC and MAPD don't do that by design so clearly the results would be different. A more useful comparison would be to use CCC or MAPD together with the utilization of reference replication profiles, or SPRINTER with or without the use of reference replication profiles. It would be useful for this manuscript to directly establish the utility of reference replication profiles this way. Currently, the presentation could mislead some readers into preferring one approach over the other without being aware of the context in which each method was developed or can be applied. I would thus recommend more careful phrasing of this section. This is not to say that SPRINTER doesn't work as well as it is advertised to; it is to say that other methods have their power as well. As part of this, I don't think "threshold-based" is an accurate way to differentiate SPRINTER from previous methods, because doesn't SPRINTER also use thresholds?

It is also important to clarify that the sorting of G1, early, mid and late S, and G2 cells (sup Figure 7b) was not continuous, i.e. the sorted bins were discrete and separated from each other, thus only a subset of all cells were sorted, rather than choosing continuous bins that together sort all cells or at least all S phase cells. This choice of sorting gates is acceptable but leads to inflation in the estimation of the accuracy of SPRINTER in calling cells at different cell cycle stages. It leads to a disconnect between the data quality (rather noisy for calling S phase cells) and the reported success in calling cell cycle stages. It also creates a meaningless comparison to previous approaches, which used different sorting gates or thresholds. This all needs to be clarified and phrased in an accurate way as to not mislead readers to over evaluate the accuracy of SPRINTER and the current scDNA-seq data (or the limitations of previous approaches).

It's also unclear how the authors used CCC and MAPD to call G1 vs G2 cells- did they rely on total sequencing depth as in their SPRINTER pipeline? Can the authors clarify this and the reason for the superior performance of SPRINTER in this respect?

Other comment:

It is important to acknowledge that fraction of cells in S phase (or S and G2 phases) isn't a direct representation of the level of proliferation rate, since the length of individual cell cycle stages could also vary between subclones or samples. Could the authors comment on that?

Reviewer #3:

Remarks to the Author:

Key results

The authors develop a new algorithm, SPRINTER, with the purpose of using single cell whole genome sequencing to deduce mutational heterogeneity and clone-to-clone proliferation rate differences in cancer samples. The authors first develop and test SPRINTER using a large scWGS dataset of HCT116 cancer cells that the authors generated themselves. SPRINTER is shown to better deconvolute S-phase DNA replication contribution to cell-to-cell copy number variability and thus improve single cell CNV calling, mutational clone assignment, and accurate measurement of clone-specific proliferation levels,

compared to previous algorithms. The authors then apply SPRINTER to patient tumour samples that they generated for this paper (NSCLC patient), showing that estimates generated by SPRINTER match clinical features and outcomes of the tumours (e.g. primary vs. metastasis). Lastly, the authors also applied SPRINTER on published breast and ovarian cancer scWGS samples. The major conclusions of the cancer studies are that higher proliferation is correlated with increased metastatic potential and/or mutational burden, as well as increased ctDNA shedding.

Validity, originality and significance

SPRINTER improves on previous methods, especially for S-phase and G2 assignment of cells. In particular, the aspects that I see as an improvement are using constitutive replication timing (RT) information to define S-phase state of each cell and the replication aware correction of GC bias. In general, the authors have made logical and effective decisions in their method for identifying S-phase cells, separating CNV signal from DNA replication signal, and using all this information to assign cells to mutational clones. There are some clarifications of the method detailed in the Major Comments section below.

The authors also generated two useful datasets for the field. The HCT116 diploid and tetraploid datasets with more accurate S-phase information could be used by the authors or others to further develop scWGS cell cycle inference methods that could include DNA replication timing, as described in the discussion. The NSCLC patient dataset is an impressive collection of sequencing and clinical data, with some temporality in the sequencing datasets e.g. samples taken at primary tumor surgery and samples taken at autopsy. This dataset has the potential to become a valuable in-depth case study, and the authors use this dataset to a good degree to showcase their method. However, the authors focused on results that agree with previous knowledge, e.g. association between high clone proliferation and metastatic seeding potential, and more proliferative clones shed more ctDNA. I assume this is because the data is only from one patient and the authors' focus was to highlight their method.

Lastly, the authors further showcase their method by applying it on previously published ovarian and breast cancer scWGS samples from Funnell et al 2022. I feel particularly here, there is room for more significant investigation, for example, by juxtaposing the mutational signature and heterogeneity findings in Funnell et al with clone proliferation data that SPRINTER can provide. What can SPRINTER add to the story that Funnell et al's methods could not? Beyond just showing that there is clone to clone proliferation rate variability.

More areas where originality and significance could be improved are detailed in the Major Comments section below.

Major comments

1. SPRINTER's bin size selection. The authors describe their minimum bin size, 50 kb, as it matches the typical bin size of replication timing data. However, I have not found description of whether there is a max bin size limit. This also relates to whether there is a minimum read count per cell cutoff. I understand the reasoning behind using the same number of reads per bin. However, it is not clear to me how SPRINTER can accurately predict CNA between cells, especially breakpoints, where they start

and end, if every cell potentially has different sized bins. How do you relate between cells where their bin sizes are very different e.g. 50kb vs. potentially ≥ 1 Mb sized bins. I can imagine that if each cell was analyzed in a sliding window manner, this would get over the issue, but from reading the methods, this doesn't seem to be the case.

2. The authors write "CNAs are expected to induce segments containing both early and late...segments that only include either early or late bins are discarded since they are likely due to replication". The authors use quite separate cutoffs for calling Early vs. Late (± 0.5). In my experience, with this type of cutoff, Early and Late regions will usually be beyond 1Mb away from each other, therefore, the authors may be selecting for very large CNAs. Replication domains are typically said to be ~400-800Kb, however, large domains of Early and Late timing, called constant timing regions which I believe the ± 0.5 would call, can be as large as 5-6Mb with an average of around 1.5Mb in more terminally differentiated cells (Rivera-Mulia et al. 2015). To help clarify, perhaps the authors could show a graph for neighboring Early and Late bin distance vs. the distribution of bin sizes chosen for individual single cells. It makes sense why the authors would restrict to CNAs that contain both Early and Late bins, so that the method remains replication aware, but the paper needs clarity on what types of CNAs the authors are missing. For example, previous papers suggest that CNAs that occur in early replicating regions relate to recombination-based repair mechanisms such as homologous recombination (HR) (De and Michor 2011; Koren et al. 2012; Morganello et al. 2016). Additionally, there is also an overall bias for CNAs, particularly deletions, to occur in late replication timing (De and Michor 2011; Du et al. 2019; Koren et al. 2012). CNAs like these that are exclusive to one replication timing may be missed in SPRINTER.

3. Missed CNAs. Similar to the comment above, the authors describe that most CNAs are over 1Mb in size, does this mean that SPRINTER is missing smaller CNAs? Or are these smaller CNAs beyond the capability of scWGS? Can the authors show e.g. using the Funnell et al data, that they are identifying the same complement of CNAs as Funnell? Or if not, what kind of CNAs are missed by SPRINTER? Alternatively, the authors could compare the clone level CNAs to CNAs found in bulk samples. The authors did compare to bulk samples for SNVs for the NSCLC dataset, but as far as I can see, they didn't compare the CNAs. Furthermore, due to the replication aware nature of this method, as far as I understand, only 50-70% of the genome is assayed as these are the regions with constant replication timing between cell types. Does this mean that CNAs in the other 30-50% of the genome are not detected? And therefore could SPRINTER potentially miss out on clones and subclones where CNAs differ in the undetectable regions?

4. As a follow on from the previous comment, I would be curious to see the NSCLC evolutionary tree delineated by CNAs instead of SNVs. This was described in the text but I do not see a figure in the main figures or the supplementary. Did the CNA phylogeny differ to the SNV phylogeny? And how so? I understand that the primary purpose of inferring replication aware CNAs is so you can better identify clones and proliferation level, however, beyond this the authors primarily used SNVs to look at evolutionary history of the NSCLC dataset. For example, one question that could be asked is whether the clones that contribute to ctDNA have a high CNA burden? The perspective here is that CNAs and SNVs can have potentially very different consequences on a cell and result from very different mutational pathways. Therefore, looking at both and how they are different could highlight different aspects of tumor history. The results here could add a more significance to this work.

5. Proliferation rate inferred from S-phase and G2 cell fractions. In general, I agree with using the fraction of S-phase cells as a proxy for how proliferative a clone is. The assumption being that the

more proliferative the clone, the more likely you are to capture cells in S-phase at any given time. However, this may oversimplify the cell cycle diversity of cancer. I was unsure of the reasoning behind showing the proportion of G2 cells or S+G2 cells, e.g. Fig. 5a, Supp Fig. 23, yet the main measure the authors use to compare proliferation between clones is the S-phase fraction. Is this because one might assume that if cells undergo S-phase, then they would eventually make it to G2? However, if you look at the S+G2 graph in Fig. 5a, the S fraction only rank is different to the S+G2 fraction ranking of clones. Therefore, there is some sort of discrepancy in agreement between S fraction and G2 fraction. This may be due to several things like perhaps SPRINTER not being able to pick up G2 cells as well as G1 and S, or more biological considerations like cell cycle arrest that can occur in cells with compromised DNA repair pathways or cells with higher ploidy (Matthews, Bertoli, and de Bruin 2022; Storchova and Pellman 2004). For example, DNA damage and replication stress can lead to S-phase and G2/M cell cycle arrest, which prolongs the time the cell spends in G2, before exiting to apoptosis or senescence (Matthews, Bertoli, and de Bruin 2022). Have the authors investigated this in their data? Would the data tell a different story if a more integrated G1:S:G2 value was used instead of just the S fraction? E.g. for Fig. 4f, Fig. 5b,c,d. Also, in the tumor datasets (HSCLC, TNBC, HGSC), the authors do have mutational information for several important genes (Supp Fig. 30, Funnell et al). The authors could look at the S:G2 ratios (or even G1:S:G2 ratios to be more comprehensive) for clones with differing mutations. This could be why there were not strong correlations found between S fraction and SNV, SV or CNA rate overall (Fig. 5b,c,d). Firstly, because maybe S fraction is only one part of the cell cycle picture, so to speak, and secondly, because you might expect clones with differing mutation backgrounds to have a different correlation between proliferation and mutation rate. Again, the results here could add a more significance to this work.

6. SPRINTER determined S-phase fraction vs. Ki-67 staining, Fig. 3a,b. The Ki-67 staining in parts of the samples are often higher than the detected clones by SPRINTER in the matched region. Can the authors explain why this occurs? Is it a tissue handling issue? E.g. adjacent tissue regions are not that close – are the authors able to mark where the SPRINTER sample was taken in the pictures in Fig. 3b? Or are there limits to how many S-phase cells a clone can contain before there are not enough G0/1/2 cells for SPRINTER to perform clone inference. Therefore, say the 80% Ki-67 stained region in the left adrenal could never be identified as a clone because the S-fraction is too large. It would be useful to understand the limits of detection of differing S-phase fractions in SPRINTER.

7. I am not sure if it is possible from the NSCLC data to infer more about the order of occurrence of mutations in Supp Fig. 30. It would definitely be interesting and potentially useful to others to know which ones were earlier, or which ones were unique to metastases or even specific metastases etc. I know the authors say that there were no known driver mutations unique to the 3rd clade, but perhaps other mutations could be informative to others and in the future. On similar lines, and also mentioned above, it would be interesting to know if there are particular CNAs unique to this 3rd clade, which may contain genes that could affect cell growth or proliferation when copy number is altered.

Minor comments

1. Can the authors please add clearer titles on their graphs for all figures? The information is in the figure legends or sometimes on the axes titles, but it would be nice to have the information more obvious at the top of each graph, to be able to understand the graph quicker, for better readability without having to shift between the graph, the main text and then the figure legends, and ultimately to avoid confusion.

- o E.g. Figure 2, c and d. Would be nice to see immediately that c is about diploid cells and d is about tetraploid cells.
- o E.g. Figure 3 b, would be nice to see that these are Ki-67 stains
- o E.g. Figure 3 c, that this is about nuclear diameter
- o This is particularly hard in the supplementary figures, where often the panels of a figure look very similar. E.g. Supp Fig 5, 6, 8, 10, 12, 13, 22, 23 etc.

2. The authors used ART to simulate read depth control sample from the sequencing reads of the sample itself. I do not understand the details of ART, so I may be missing information here. However, I do wonder about the suitability of this method for the newer Novaseq platforms that use 2-colour technology compared to the older Illumina 4-colour platforms that ART was designed on. The scWGS datasets were sequenced on the Novaseq 6000. Is there anything here to be wary about?

References

- De, S., and F. Michor. 2011. "DNA replication timing and long-range DNA interactions predict mutational landscapes of cancer genomes." *Nat Biotechnol* 29 (12): 1103-8. <https://doi.org/10.1038/nbt.2030>. <https://www.ncbi.nlm.nih.gov/pubmed/22101487>.
- Du, Q., S. A. Bert, N. J. Armstrong, C. E. Caldon, J. Z. Song, S. S. Nair, C. M. Gould, P. L. Luu, T. Peters, A. Houry, W. Qu, E. Zotenko, C. Stirzaker, and S. J. Clark. 2019. "Replication timing and epigenome remodelling are associated with the nature of chromosomal rearrangements in cancer." *Nat Commun* 10 (1): 416. <https://doi.org/10.1038/s41467-019-08302-1>. <https://www.ncbi.nlm.nih.gov/pubmed/30679435>.
- Koren, A., P. Polak, J. Nemesh, J. J. Michaelson, J. Sebat, S. R. Sunyaev, and S. A. McCarroll. 2012. "Differential relationship of DNA replication timing to different forms of human mutation and variation." *Am J Hum Genet* 91 (6): 1033-40. <https://doi.org/10.1016/j.ajhg.2012.10.018>. <https://www.ncbi.nlm.nih.gov/pubmed/23176822>.
- Matthews, H. K., C. Bertoli, and R. A. M. de Bruin. 2022. "Cell cycle control in cancer." *Nat Rev Mol Cell Biol* 23 (1): 74-88. <https://doi.org/10.1038/s41580-021-00404-3>. <https://www.ncbi.nlm.nih.gov/pubmed/34508254>.
- Morganella, S., L. B. Alexandrov, D. Glodzik, X. Zou, H. Davies, J. Staaf, A. M. Sieuwerts, A. B. Brinkman, S. Martin, M. Ramakrishna, A. Butler, H. Y. Kim, A. Borg, C. Sotiriou, P. A. Futreal, P. J. Campbell, P. N. Span, S. Van Laere, S. R. Lakhani, J. E. Eyfjord, A. M. Thompson, H. G. Stunnenberg, M. J. van de Vijver, J. W. Martens, A. L. Borresen-Dale, A. L. Richardson, G. Kong, G. Thomas, J. Sale, C. Rada, M. R. Stratton, E. Birney, and S. Nik-Zainal. 2016. "The topography of mutational processes in breast cancer genomes." *Nat Commun* 7: 11383. <https://doi.org/10.1038/ncomms11383>. <https://www.ncbi.nlm.nih.gov/pubmed/27136393>.
- Rivera-Mulia, J. C., Q. Buckley, T. Sasaki, J. Zimmerman, R. A. Didier, K. Nazor, J. F. Loring, Z. Lian, S. Weissman, A. J. Robins, T. C. Schulz, L. Menendez, M. J. Kulik, S. Dalton, H. Gabr, T. Kahveci, and D. M. Gilbert. 2015. "Dynamic changes in replication timing and gene expression during lineage specification of human pluripotent stem cells." *Genome Res* 25 (8): 1091-103. <https://doi.org/10.1101/gr.187989.114>. <https://www.ncbi.nlm.nih.gov/pubmed/26055160>.
- Storchova, Z., and D. Pellman. 2004. "From polyploidy to aneuploidy, genome instability and cancer." *Nature Reviews Molecular Cell Biology* 5 (1): 45-54. <https://doi.org/10.1038/nrm1276>. <Go to ISI>://WOS:000187993500014.

Author Rebuttal to Initial comments

Reviewer 1

Response 1.1 In this manuscript, the authors present SPRINTER, an algorithm aimed at separating pseudo CNA due to S phase vs true genome CNA. They start by training their algorithm using cells with known cell cycle phase using EdU labelling and sorting of cells in specific phases of the cell cycle, followed by DLP+ sequencing. Finally, they show that they can identify different clonal populations with different phases of cell cycle within multiple tumor nodules isolated from the same patients.

The manuscript is well written and their ability to identify S phase cells is convincing. The ability to identify CNA and cell cycle phase in the same cell is also an advancement. However, there are some important limitations to the study that are not adequately addressed.

We thank the reviewer for their appreciation of our manuscript and for recognising the advancements of our novel method. In this revision, we have now addressed the reported limitations, as described in the responses below.

Response 1.2 DLP+ only captures a fraction of the genome, and we don't know how deeply these cells were actually sequenced. The authors should include some important performance data to adequately assess the method. For example: were all cells used, and if some cells were thrown out, what percentage? What was the average fraction of the genome covered per single cell genome and the corresponding estimated CNV resolution based on that coverage? At what CNV resolution does their algorithm have difficulty discerning CNV from S phase changes?

In this revision, we have now added two new figures, Supplementary Figs. 12 and 25 (reproduced in Figs. R1 and R2 below), reporting the requested information in detail for the ground truth datasets and the NSCLC dataset, respectively. This is in addition to the information on sequencing coverage and total number of sequencing reads that was already reported in Supplementary Fig. 10 for the ground truth datasets, and Supplementary Figs. 26 and 27 for the NSCLC dataset. In the new figures, we show that the sequencing coverage per cell is approximately 0.02-0.08x across all the sequenced samples in both ground truth datasets, as well as in both the primary tumour and metastases of the NSCLC dataset (Figs. R1 and R2 below), which is a sequencing coverage higher or comparable to that used in previous single-cell studies that performed related cancer-evolutionary analyses¹⁻⁴.

In addition, the new figures include the requested information about the fraction of selected cells and the fraction of the genome covered per cell (Figs. R1 and R2 below). Specifically, for all the analyses in this study, we selected only cells with more than 100k sequencing reads, similar to previous single-cell studies with comparable sequencing coverage, because this number of reads is required for high confidence copy-number analysis of individual cells^{1,2,4,5}. In fact, since ~100 reads are generally required for high confidence estimates of read depth ratios (RDRs) in individual genomic regions^{1,2,4,5}, cells with >100k reads provide sufficient resolution for estimating CNAs that are >3Mb. Moreover, most cells in our generated datasets have more than 95% of all 50kb genomic bins covered by at least one read (Figs. R1 and R2 below) and most cells have >5 reads on average

in 50kb genomic regions, thus providing sufficient power for estimating CNAs that are larger than 1Mb (Figs. R1 and R2 below). Despite this selection, we show that >90% of all sequenced cells have been retained in all but two samples of the NSCLC dataset, and >60-80% in the remaining samples (Figs. R1 and R2 below).

Lastly, in this revision we have now performed a new spike-in, in-silico experiment to demonstrate that SPRINTER can accurately discern CNAs at a resolution of >3Mb from replication-induced changes in S phase cells (new Supplementary Fig. 23, reproduced in Fig. R19 below and described in Response 3.7). Specifically, we have used the HCT116 diploid cells sequenced as part of our generated ground truth dataset, and we have spiked-in copy-number gains and losses (by correspondingly scaling the observed read counts) into chromosomes that are known to be diploid for both G1/2 and S phase cells (Fig. R19a,b below). As such, we have applied SPRINTER to all the cells in this spiked-in dataset and we have measured the ability of SPRINTER to accurately recover CNAs of varying size (3-15Mb). We found that SPRINTER is able to recover these CNAs with high accuracy, precision, and recall (>80%) when considering the raw CNAs directly inferred by SPRINTER for individual cells in both G1/2 and S phases, and with even higher performance (>90-95%) when considering the new corrections that are now applied by the revised version of SPRINTER to the CNAs of S phase cells (Fig. R19c-e below). Note that the details of the new corrections introduced in the new version of SPRINTER are described in Response 3.7 below. While SPRINTER's performance is higher for larger CNAs, we found that SPRINTER accurately infers most CNAs >3Mb with >85% accuracy across all cells (Fig. R19f below).

We also highlight that, in response to the comments from other reviewers (Response 3.7), we have now performed new analyses in this revision (reproduced in Fig. R17 below) demonstrating that CNAs measured in single cancer cells are substantially larger (42Mb on average with 99% of segments >2Mb) than neighbouring genomic regions with the same replication timing (897kb on average with 87% of replication timing genomic regions <2Mb).

Overall, all these new analyses demonstrate that SPRINTER can accurately distinguish single-cell CNAs from replication-induced fluctuations of the observed read counts, and it can thus use this information to accurately identify S phase cells and assign them to the correct clone.

Fig. R1: scDNA-seq features of the generated ground truth datasets. Across cells from 5 cell cycle phases (x-axis) which were single-cell DLP+ sequenced from the diploid (left) and tetraploid (right) ground truth datasets, multiple sequencing features have been computed: (a) number of selected cells for analysis, (b) total number of sequenced reads, (c) fraction of selected cells with a sufficient number of sequencing reads (>100k indicating successful DNA library preparation among all cells (threshold on yielded sequencing reads to define total cell count is estimated as the median of the 2nd percentile computed across all samples for all isolated objects with above average intensity, measured by microscopy imaging), (d) sequencing coverage per cell, (e) fraction of 50kb genomic bins covered by at least one sequencing read, and (f) average number of sequencing reads in 50kb genomic bins per cell.

Fig. R2: scDNA-seq features of the generated NSCLC dataset. Across 10 samples (x-axis) single-cell DLP+ sequenced from 5 primary tumour regions (left) and 5 metastases (right) of CRUK0516, multiple sequencing features have been computed: **(a)** number of selected cells for analysis, **(b)** total number of sequenced reads, **(c)** fraction of selected cells with a sufficient number of sequencing reads ($>100k$) indicating successful DNA library preparation among all cells (threshold on yielded sequencing reads to define total cell count is estimated as the median of the 2nd percentile computed across all samples for all isolated objects with above average intensity, measured by microscopy imaging), **(d)** sequencing coverage per cell, **(e)** fraction of 50kb genomic bins covered by at least one sequencing read, and **(f)** average number of sequencing reads in 50kb genomic bins per cell.

Response 1.3 The authors are really performing the construction of phylogenetic trees based on pseudobulk measurements, not at single-cell resolution. They would need much more complete genomes to do that. Depending on CNV resolution, as well as genome loci and allele dropout for SNV/Indel calling, they will be missing CNV and SNV data at single cell resolution. This will be especially problematic for variants that are present in a small number of cells, which will be excluded when performing pseudobulk calling. It is important that they highlight this limitation, as some of their biological conclusions may in fact be due to higher resolution genomic variation they are not capturing. For example, a proliferation-associated mutation with their current clones could actually explain the S phase percentage within the pseudobulk clone.

We agree with the reviewer that the power to identify genomic variants in single-cell clones depends on the sequencing coverage per clone, which in turn depends on the related number of cells. Therefore, some SNVs and particularly short CNAs only present in small clones (especially those below the resolution of low coverage scDNA-seq data) might have been missed in the clones analysed within the NSCLC dataset. We have introduced a new sentence in the revised Discussion to acknowledge this limitation:

“Lastly, due to the low sequencing coverage of scDNA-seq, we also note that some SNVs and particularly short CNAs only present in small clones might have been missed by these analyses. Therefore, further improvements in pseudobulk analyses or sequencing with deeper coverage could reveal more genetic events associated with changes in proliferation.”

However, we also highlight that, in this study, we have adopted evolutionary methods that are specifically intended to account for missing data (and false negatives) in the analysed genomic variants. Particularly for SNVs, we identified loci that are not covered by a sufficiently high number of sequencing reads to statistically exclude the presence of a mutation using a hypothesis testing approach based on binomial models (labelled as “unknown” genotype), and we have applied the HUNTRESS algorithm that has been specifically developed to correct these missing data or false negatives during the phylogenetic reconstruction⁶. The details of this approach are described in the section “*Phylogenetic analysis*” in Methods. This approach is thus expected to be able to correctly infer SNVs that are identified in at least one sample overall but have been missed in some clones.

In addition, we have now performed a new analysis to demonstrate that the sequencing coverage per sample in the NSCLC dataset is sufficient to identify the vast majority of SNVs present with a cancer cell frequency of >10% in every sample (new Supplementary Fig. 42, reproduced in Fig. R3 below), such that their presence in the related clones can be correctly inferred later using the approach described above. Specifically, for every genomic locus with a driver mutation in any of the analysed samples, we have calculated the probability of observing at least one variant read within each sample if the SNV is present using the observed total number of reads and a binomial model as in previous studies⁴. We found that ~99.9% of all drivers across all samples have a >50% probability of detection, and ~99.4% of all drivers across all samples have a >75% probability of detection (Fig. R3a,b,f below). Therefore, although we cannot exclude that some variants were not detected, this new analysis suggests that we have sufficient power to detect variants for the vast majority of driver genes across every sample (Fig. R3f below).

Moreover, we highlight that most of the clones identified by SPRINTER in the NSCLC dataset are of medium size, containing ~160 cells on average, which corresponds to an expected sequencing coverage of >5x (Fig. R3c below). Therefore, we have also performed a new analysis to demonstrate our power to identify mutations in individual clones, using an approach similar to the one described above at the sample level and considering a mutant allele frequency of 50% (the value expected in genomic regions that are copy-number balanced). Specifically, for every genomic locus with a driver mutation in any sample, we have calculated the probability of observing at least one variant read for each clone if the SNV is present using the observed total number of reads in the specific clone and a binomial model as in previous studies⁴. We found that ~80% of all drivers across all clones have a >50% probability of detection, and ~70% of all drivers across all clones have a >75% probability of detection (Fig. R3c,d,e below). Therefore, even though we adopted an approach to correct for false negatives across clones as described above, we demonstrated that for most mutations in most clones we have sufficient power to accurately identify the presence of mutations per clone (Fig. R3e below). Note that this is on top of the correction for missing data and false negatives that is applied by HUNTRESS in the downstream phylogenetic analyses, as described above.

Lastly, regarding CNAs, SPRINTER identifies the presence of CNAs in individual cells when considering events that are above the resolution of low coverage scDNA-seq, i.e., >1Mb for data generated with a sequencing coverage 0.02-0.08x like those in this and previous studies¹⁻⁴. Even though some CNAs may be missed in certain cells, in the new version of SPRINTER we have introduced a method to correct the presence of small CNAs using all the cells assigned to the same clone (details are reported in Response 3.7 below). We have performed a new spike-in experiment to demonstrate that this new correction allows SPRINTER to accurately recover nearly all CNAs of size >3Mb in both S and non-S phase cells (Fig. R19 below, described in Response 3.7). Therefore, we do not expect that the identification of these CNAs is substantially affected by the size of the clones. However, particularly short CNAs that are below the resolution of low coverage scDNA-seq might be missed, and we have therefore added a related acknowledgment in the revised Discussion as described at the start of this response.

Fig. R3: The generated NSCLC dataset provides sufficient power to identify most mutations in samples and clones. a-b, For each driver mutation in each gene (x-axis) and in every sample (y-axis), the total number of reads covering the corresponding genomic locus (a) and the probability of observing more than one variant read computed using a binomial model with an underlying cancer cell fraction of 10% (b) are reported. **c-d,** For each driver mutation in each gene (x-axis) and in every SPRINTER clone (y-axis), the total number of reads covering the corresponding genomic locus (c) and the probability of observing more than one variant read computed using a binomial model with an underlying mutant allele frequency of 50% (d) are reported. **e-f,** The proportion (y-axis) and number (labels) of all loci with non-zero read counts, >50% probability of observing a mutation if present, and >75% probability of observing a mutation if present (x-axis) were computed either across all SPRINTER clones (e) or across all samples (f).

Response 1.4 Can the authors estimate the fraction of clones they are actually capturing in a given sample based on the number of cells they sequence, their sensitivity, dropout, etc? This is important, as comparisons between lesions, such as comparing the surgical sample to autopsy could just be a sampling issue.

We have performed a new analysis to investigate the power and precision of capturing tumour clones with varying frequencies (i.e., with varying numbers of cancer cells) in each sample independently, considering the different total number of cells sequenced in each sample as well as the different sequencing features of each sample (e.g., sequencing coverage, error rates, CNA rates). To do this, we used the in-silico, subsampling approach that has been previously introduced by similar single-cell studies of cancer evolution⁴, and we thus generated 125,000 in-silico datasets across samples and across clones of increasing frequency (100 repeats for 25 tumour clones with 50 values of varying numbers of cells between 1 and 100). Specifically, we generated each in-silico dataset by randomly subsampling a varying number of cells 1-100 (corresponding to a tumour fraction of 0.1-1% on average) from each sufficiently large tumour clone (i.e., >5% tumour fraction) identified by SPRINTER and for each primary tumour and metastatic sample containing at least two of these clones. The subsampled cells were then added to all other remaining cells sequenced from the same sample (i.e., cells in different clones from the same sample), effectively obtaining a new dataset containing the cells from all the original clones but with one subsampled clone with known frequency. Note that this subsampling procedure preserves all other features of the single-cell dataset, including the rates and sizes of the CNAs in different clones as well as errors and biases in the DNA sequencing signals.

We applied SPRINTER to each subsampled dataset and quantified SPRINTER's ability to detect the small, subsampled clones with varying frequencies in terms of both precision and recall. While clones in some samples could be identified with higher accuracy than others, we found that SPRINTER accurately recovers most clones with >30 cells in every case (new Supplementary Fig. 30, reproduced in Fig. R4 below). Moreover, we showed that SPRINTER accurately recovers all clones containing more than 40-60 cells across all samples, corresponding to tumour fractions >0.4% on average. Note that these results are in line with the results obtained in previous similar single-cell studies⁴. Since the clones analysed in this study have tumour fractions between 3-81% (with an average of ~20%), corresponding to ~160 cells on average per clone, these results indicate that all the analysed clones could be accurately recovered by SPRINTER. We report the details of this in the new Supplementary Fig. 30 (reproduced in Fig. R4 here below).

Fig. R4: SPRINTER accurately recovers most clones containing more than 30 cells. SPRINTER was run on 125,000 datasets obtained by randomly subsampling a varying number of cells (x-axis, with 50 values between 1 and 100) from each clone (colour, 25 in total) in each NSCLC sample (row) for 100 repeats. Each dataset is obtained by adding the subsampled cells to all other remaining cells sequenced from the same sample. Recall (y-axis on left-side plots) and precision (y-axis on right-side plots) are measured for each subsampled clone in each generated dataset (each dot represents the mean across repeats and whiskers indicate the related standard deviation).

Response 1.5 The movement of clones within the same patient is interesting to see. That has been done with bulk sequencing, but the authors are also able to add cell cycle status on top of that. Although interesting, it does not provide much biological or clinical insight. Do the authors have any other observations where adding cell cycle provides new biological or clinical insights?

The novel clone assignment of replicating cells in SPRINTER did provide novel biological insights in this study as, for the first time, it reveals that genomically-distinct tumour clones with significantly different proliferation rates co-exist within the same tumour in the same patient. Previous approaches to measure proliferation from RNA-sequencing or pathological Ki-67 data do not capture clone-level information⁷⁻¹⁵, and therefore could not detect the presence of minor clones with different proliferation rates that we identified within individual primary tumour regions and also within each individual metastasis. Details of this comparison are described in the section *“Proliferation heterogeneity in a primary tumour and metastases”* in Results (supported by Fig. 3a in the revision). Even more remarkably, we showed that the clones with the highest proliferation within the same tumour are those that are more likely to disseminate and seed metastases, i.e., are those with increased metastatic potential (Fig. 4e in the revision). These results are now more clearly highlighted in the text, for example in the revised Introduction and Discussion:

“Through evolutionary analysis of SPRINTER’s results, we uncovered a link between high clone proliferation and increased metastatic seeding potential of specific tumour clones, as well as increased clone-specific shedding of ctDNA.”

“By generating a novel single-cell, longitudinal, primary-metastasis matched dataset, we found that SPRINTER reveals an association between high clone proliferation and increased metastatic seeding potential of specific tumour clones, with high proliferation clones comprising the specific subsets of cancer cells that are more likely to seed metastases.”

These results correspond to important biological insights because they provide key information to improve our understanding of the cellular mechanisms underpinning metastatic potential. Additionally, they provide clinical insights because they suggest that a therapeutic approach targeting the most proliferative clones may have a higher chance of success in delaying metastatic progression. Importantly, while these results are consistent with the poor prognosis observed for high proliferation tumours^{8-12,14,15}, they were not necessarily expected, given some previous studies have suggested that disseminating cells may undergo epithelial mesenchymal transition (EMT), and thus may have a more invasive but less proliferative phenotype¹⁶⁻¹⁹. We have now added this important consideration in the revised Discussion:

“Importantly, while these results are consistent with the worse outcomes observed for high proliferation tumours^{8-12,14,15}, they were not necessarily expected based on some previous biological studies which suggest that disseminating cells may undergo epithelial mesenchymal transition (EMT), which has been associated with a more invasive but less proliferative phenotype¹⁶⁻¹⁹. Our results could be consistent with highly proliferative clones undergoing EMT but then returning to a proliferative state in a target organ plastically, or with these clones subverting signals that reduce proliferation during EMT.”

In addition to the results explained above, SPRINTER’s clone assignment of replicating cells has also enabled novel findings related to the differential shedding of circulating-tumour DNA (ctDNA) into

the bloodstream by clones with different proliferation rates. In fact, we demonstrated that high proliferation tumour clones shed more ctDNA than low proliferation tumour clones within the same tumour (Fig. 4f and Supplementary Fig. 49 in the revision). While previous studies^{20,21} based on bulk approaches have shown related observations for distinct tumours across different patients, our study shows for the first time that this relationship holds for distinct tumour clones present within the same patient. We have now clarified this important consideration in the revised Introduction:

“While previous bulk-based studies²⁰⁻²² indicated that more proliferative tumours in different patients shed more ctDNA and are associated with worse outcomes, our study demonstrates that similar associations hold for distinct tumour clones that co-exist within the same tumour in the same patient, and especially that they hold for the specific metastatic seeding clones that comprise the subset of cancer cells responsible for metastasis. In addition to the introduction of SPRINTER, this finding has been made possible by the availability of the newly generated longitudinal, primary-metastasis matched scDNA-seq dataset with detailed clinical annotation.”

This result is also potentially important both biologically and clinically because it impacts the interpretation of ctDNA in liquid biopsy-based approaches: specifically, it suggests that the prevalence of ctDNA within a patient might not only relate to the volume of the corresponding clone (i.e., number of related cells) but may also be influenced by its proliferation. Furthermore, it suggests that ctDNA might also be used to identify the presence of high proliferation tumour clones within a patient. We have now added these important considerations in the revised Discussion:

“In particular, ctDNA-based predictive approaches are motivated by this study since we demonstrated differential shedding of ctDNA into the bloodstream by distinct tumour clones, with high clone proliferation associated with increased ctDNA shedding, thus suggesting that ctDNA shedding could be used as a proxy for clone proliferation. Given our results indicate that the prevalence of ctDNA within a patient might not only relate to the volume of the corresponding clone (i.e., number of related cells), but may also be influenced by its proliferation rate, they also warrant the careful interpretation of ctDNA prevalence in related studies.”

To further demonstrate that SPRINTER can enable biological discoveries, in this revision we have also introduced a new feature of the SPRINTER algorithm that aims to estimate clone-specific altered replication timing (ART), i.e., changes in the replication timing of a region of the genome in the tumour compared to a reference replication profile obtained from normal cells. ART is a key epigenetic mechanism in cancer that correlates with gene expression and chromatin structure²³⁻²⁹. The details of this new feature are described in Response 2.2 below. In addition to confirming that ART affects <10% of the genome on average per clone, as expected based on previous studies^{23,25-27,29}, this new analysis allowed us to reveal the presence of ART in key cancer genes known to affect proliferation or metastatic potential, and especially to elucidate ART events that were shared or unique to different metastatic clades in the NSCLC dataset (Fig. 4d in the revision, reproduced in Fig. R5 below). Importantly, while we did not identify driver mutations or particular CNAs unique to the third metastatic clade which contained the tumour clones with the highest proliferation, we did identify ART events unique to this clade, especially affecting *KRAS* (Fig. R5 below).

To support this result and the identified ART events, we have also performed a new analysis of gene expression using matched RNA-sequencing data from the same tumour³⁰, since ART is known to be

associated with differential expression (with higher or lower expression in genes affected by late-to-early or early-to-late ART compared to normal tissue without ART, respectively)^{23,24}. We confirmed that the ART events found in *KRAS* and other genes relevant to proliferation and metastasis were associated with expected changes in gene expression when comparing the related samples to primary tumour regions from different clades, to cancer and normal tissue samples from 347 other TRACERx patients (130 normal and 915 tumour samples)³⁰, or to a pre-mortem relapse sample of the left adrenal metastasis belonging to the third metastatic clade (Fig. 4d in the revision, reproduced in Fig. R5 below). This new analysis provides preliminary insight into the notion that non-genetic alterations, such as ART, might play an important role in metastatic clones with increased proliferation, and some of these (those associated with ART) can now be revealed and analysed in individual clones using SPINTER. We describe this new analysis in a new paragraph in the section “Increased metastatic seeding potential and ctDNA shedding of high proliferation clones” in the revised Results:

*“We investigated the presence of genetic alterations in key cancer genes associated with these clades. Despite the high proliferation of the third clade, we did not identify any known driver mutation unique to this clade and shared by most of its clones, and the driver mutations identified, e.g., in TP53, NF1, and CDK12 genes, were shared with other clades (Supplementary Fig. 41). Similarly, we did not identify any CNA specific to this clade (Supplementary Fig. 44). In addition to genetic alterations, non-genetic alterations also play a key role in cancer progression^{25-27,29} and may be present in different metastatic clades. To investigate this, we demonstrated that SPINTER’s clone assignments can be used to reveal the presence of clone-specific altered replication timing (ART, i.e., changes in the replication timing of a genomic region in the tumour compared to the default reference replication timing profile derived from normal cell lines) by using the S phase cells assigned to each clone independently and applying approaches similar to previous replication timing studies³¹ (Methods). Overall, the fraction of the genome affected by ART was <10% on average across clones, which matches previous measurements^{23,25-27,29} (Supplementary Fig. 45) and does not affect SPINTER’s results (Supplementary Fig. 46). In particular, we found ART events affecting genes that are known to impact proliferation or metastatic potential in cancer and that are shared or unique to different metastatic clades (Fig. 4d). For example, we found ART events in genes that are shared by all clones, e.g., *PDL1*, *TERT*, and *PIK3CA*, ART events unique to only one or two of the metastatic clades on distinct branches of the phylogenetic tree, e.g., *CDK12* in the second and third clades and *NCOA2* in the first clade, and ART events mostly exclusive to the third, most proliferative clade, e.g., *KRAS*. Since ART is known to be associated with differential expression (higher or lower expression for genes affected by late-to-early or early-to-late ART, respectively)^{23,24}, we showed that these ART events were supported by related expression changes by comparing previous bulk RNA-sequencing data³⁰ from matched primary tumour regions to primary regions from different clades, to cancer and normal tissue samples from 347 other TRACERx patients, or to a pre-mortem relapse sample from the left adrenal metastasis in the third clade (Fig. 4d and Supplementary Fig. 47). For example, the comparison across clades and with the pre-mortem relapse sample confirmed a significant increase in *KRAS* expression in the third clade, and particularly in the left adrenal metastasis, which contained the highest proliferation clones, most of which displayed late-to-early ART of *KRAS* (Fig. 4d). Moreover, reduced *PDL1* expression correlating to early-to-late ART in almost all clones across the tumour was further confirmed by the clinal immunohistochemistry report for this patient (*PDL1* 0%, Methods).”*

Inferred clone-specific altered replication timing

Fig. R5: Altered replication timing in genes known to impact proliferation and metastatic potential in cancer. In the two main phylogenetic branches containing different metastatic clades (top row), the clone-specific ART (coloured rectangles) inferred in this tumour is shown for each clone (second row) for a set of genes (left) that are known to impact proliferation and metastatic potential. For each gene, the reference replication timing derived from normal cell lines is shown (left column). ART is supported by related expression changes measured using bulk RNA-sequencing (right heatmap, for matched primary tumour regions compared to normal tissue samples, to other TRACERx tumours, to primary regions from CRUK0516 from different clades, or to a pre-mortem relapse sample from the left adrenal metastasis in the third clade), with late-to-early ART associated with increased expression and early-to-late ART associated with decreased expression.

We highlight that all the novel biological and clinical insights described above for the NSCLC case could only be revealed thanks to the novel single-cell, whole-genome DNA sequencing dataset generated in this study by sequencing cells belonging to matched primary tumour and metastasis samples obtained longitudinally from the same patient with detailed clinical annotation (e.g., integrating the use of matched patient clinical imaging and ctDNA). The novelty of these insights is thus partly explained by the fact that this study provides the first dataset of this kind, especially in humans and for metastatic tissues. We highlight this important consideration in the revised Introduction:

“In addition to the introduction of SPRINTER, this finding has been made possible by the availability of the newly generated longitudinal, primary-metastasis matched scDNA-seq dataset with detailed clinical annotation.”

Lastly, we also highlight that our new proliferation analysis of the previously generated TNBC and HGSC datasets with 61,914 single cells also enabled novel biological findings related to the association between increased clone proliferation and increased clone-specific rates of different genomic variants (SNVs, SVs, and CNAs). In this revision, we have performed a new, more powerful analysis of the rates of genomic variants at the individual cell level, and we now clearly demonstrate that high proliferation clones have significantly higher rates of SNVs, SVs, and CNAs in individual cells consistently in both datasets (new Figs. 5b-d in the revision, reproduced in Fig. R25 below). The details of this new analysis are described in Response 3.11 below. This result represents an important biological insight because it suggests that high proliferation clones might have an advantage in acquiring key genomic alterations during tumour progression. We have now clarified this important point in a new paragraph in the revised Discussion:

“In this study, we showed that higher proliferation clones have a significantly increased rate of multiple genomic variants (SNVs, SVs, and CNAs). This result may suggest that the increased proliferation of these clones might be advantageous for the acquisition of new key genetic alterations that drive cancer progression, and other phenotypes like resistance, further explaining the worse outcomes reported in highly proliferative tumours^{8-12,14,15}.”

Moreover, guided by a recommendation from another reviewer (Reviewer 3 with details in Response 3.11), we have also performed two new analyses of the TNBC and HGSC datasets, further enhancing our study’s novel findings. First, we have leveraged SPRINTER’s ability to jointly infer clone genomic alterations and proliferation rates to investigate the presence of amplifications of oncogenes, deletions of tumour suppressors genes (TSGs), and driver mutations that are enriched in clones with high proliferation (new Figs. 5e,f and Supplementary Fig. 53 in the revision, reproduced in Figs. R26 and R27 below). As such, we found several amplifications of known oncogenes that are significantly associated with increased clone proliferation (e.g., CDK4, CCND2, ERBB2, ERBB3, EGFR, AKT3, HRAS, shown in Fig. 5e). This association is further supported by the fact that these genes are enriched in relevant pathways related to the cell cycle and proliferation (e.g., PI3K/AKT/mTOR signalling, KRAS signalling upregulation) as shown using a new gene set enrichment analysis³² (Fig. 5f, details in Methods). Moreover, we found a smaller number of deletions in TSGs (e.g., SMAD4, KEAP1) and driver mutations (e.g., KMT2D, EPHA7) which are significantly associated with high clone proliferation (Supplementary Fig. 53). For some of these genes (e.g., ARID2, KEAP1, PTPRD, TTN, MROH2B), the anti-proliferative and tumour suppressive effects of related deletions or mutations match the validation measurements performed in previous cell-line siRNA experiments³³. Overall, these results highlight that the joint analysis of clone proliferation and cancer evolution enabled by SPRINTER can elucidate mechanisms underlying changes in cancer proliferation in patient tumours. More details about this new analysis are provided in Response 3.11 below.

In the second new analysis, we explored potential changes in the duration of G2 phase relative to the duration of S phase using the fraction of G2 phase cells over the fraction of S phase cells (G2/S ratio) inferred by SPRINTER. While increased or decreased S fractions are expected to yield increased or decreased G2 fractions, respectively (since the presence of more/less S phase cells generally determines if more/less cells enter G2), an increase or decrease in the G2 fraction without a corresponding variation in the S fraction could indicate a change in the G2 phase length relative to the length of the S phase. For example, in the TNBC dataset, we found that SPRINTER tumour

clones with previously identified homologous recombination deficiency (HRD) have a significantly higher G2/S ratio than other clones ($p = 0.008$, Supplementary Fig. 54, reproduced in Fig. R28 below). This result is thus consistent with a possible prolonged G2 phase relative to the length of S phase in HRD clones, as reported in previous studies³⁴, which indicate prolonged G2/M cell cycle arrest may occur in the absence of the ability to properly repair double strand breaks via homologous recombination. Further details of this new analysis are also provided in Response 3.11 below.

Overall, our study (including the new additions from this revision) provides seven novel biological insights that are summarised here below:

1. Tumour clones with significantly different proliferation rates co-exist within the same tumour, both within the same primary tumour region and even within the same metastasis.
2. High proliferation clones have increased metastatic seeding potential compared to clones with lower proliferation within the same tumour, with high proliferation clones comprising the specific subsets of cancer cells that are more likely to seed metastases. This association was not necessarily expected based on some of the biological results reported in previous literature.
3. High clone proliferation is associated with increased clone-specific shedding of ctDNA for different clones within the same tumour. Differential ctDNA shedding has only been shown before for distinct tumours in different patients, not individual clones, and our result has implications for ctDNA interpretation.
4. Distinct ART events are present in different clones within the same tumour and in different metastatic clades, they are associated with corresponding changes in gene expression, and affect key genes involved in proliferation and metastatic potential.
5. High clone proliferation is associated with increased accumulation of SNVs, CNAs and SVs at the individual cell level, suggesting that highly proliferative clones may have an advantage in acquiring key genomic alterations during tumour progression.
6. At the clone level, there are amplifications of oncogenes, deletions of TSGs, and driver mutations that are significantly associated with high proliferation, and in particular these oncogene amplifications are enriched in proliferation pathways.
7. Tumour clones in TNBC with HRD have evidence of a prolonged G2 phase relative to the length of S phase, which is consistent with cell line studies which revealed possible G2/M arrest in the presence of HRD.

Reviewer 2

Response 2.1 This beautiful manuscript by Lucas et al utilizes large-scale scDNA-seq of cancer samples to identify cells in S phase and other cell cycle phases alongside CNAs, thus enabling the inference of proliferation rates of cancer subclones. This is an important problem and a powerful solution based on single cell sequencing and computational identification of subclones and the cell cycle phases of individual cells. The authors apply this approach to interesting questions regarding tumor evolution and metastasis.

There are several limitations that should be accounted for:

We thank the reviewer for their appreciation of our manuscript and for recognising the advancements of our novel method, as well as the importance of the questions that we have addressed in this study. In this revision, we have now addressed the reported limitations, as described in the responses below.

Response 2.2 Reference replication profiles. The crux of the method is to use reference replication profiles to identify cells in S phase. The authors justify this based on the conservation of replication profiles across samples, however the replication timing data they use is predominantly derived from non-cancer samples, cancer cell lines, and a small number of primary cancer samples. How can we know that replication profiles won't be different in actual cancer samples across patients, especially in subclones with potential genetic and/or epigenetic alterations that could impinge on DNA replication?

Ultimately this approach would be useful if it can be reliably applied to individual patients. The current use of averages of several samples and of reference replication profiles therefore does not fully demonstrate that this method is reliable enough to be applied on individual cancer samples and subclones, especially considering the genomic and epigenomic heterogeneity of different tumors.

The SPRINTER algorithm does not require the analysed cells to have the exact same early and late replication timing regions as those defined by the reference replication profiles. In fact, the algorithm is based on multiple statistical approaches that account for, and are robust to, the presence of genomic regions with different replication timing than what is expected from the reference, with two key examples. First, SPRINTER identifies S phase cells by using a novel summary statistic that is robust to the presence of genomic regions with unexpected read counts. This is because the new statistic only requires a sufficiently large subset of early/late genomic regions with higher/lower read counts than expected respectively, and it is therefore not influenced by the presence of other early/late genomic regions with unexpected read counts (details are in the section "*Identifying S phase cells*" in Methods). Second, SPRINTER's statistical approach calculates these statistics in each chromosome independently and aggregates these values using the harmonic mean, which is robust to the presence of errors or focal alterations in certain chromosomes. Overall, the SPRINTER algorithm is thus applicable to heterogeneous tumours with replication timing alterations or errors in the classifications of replication timing from the reference profile. We have now clarified these

important considerations in multiple points throughout the revised Results and Methods sections, for example:

“SPRINTER achieves this goal by leveraging prior information on genomic regions that are expected to have early or late replication timing, which is known to be conserved for >50% of the genome across different cell types^{28,35,36} and cancer cells^{23-27,29} (Supplementary Figs. 1-4). Since the replication timing of some genomic regions can still vary in the analysed cells, SPRINTER employs probabilistic approaches and statistical tests that do not fully rely on this prior information, but rather account for the presence of potential changes or errors.”

“Note that this statistic is expected to be robust to the presence of alterations or errors in replication timing classifications, since it requires only a subset of bins, not all early or late bins, to display the expected signal in RTPs.”

“Since variations of t_b are expected to occur along the entire genome during S phase, we perform the test on each chromosome independently and the resulting values of each statistic are combined using the harmonic mean; this approach helps overcome noise and errors that can be localised to certain genomic regions.”

To demonstrate SPRINTER’s robustness in this regard, we performed an in-silico experiment in which SPRINTER was executed on S phase cells after introducing increasing fractions of errors in the classification of early/late replication timing. In this revision, we have performed an extended version of this experiment on 500 diploid and 500 tetraploid S phase cells from the ground truth datasets, and we clearly showed that SPRINTER is robust to replication timing errors or changes up to 40% (new Supplementary Fig. 15 in the revision, reproduced in Fig. R6 below). As such, SPRINTER is robust to the fraction of replication timing differences expected for both cancer and non-cancer cells. In fact, we have also performed a new analysis to estimate the expected fraction of replication timing differences by comparing every available replication profile (experimentally measured in multiple previous cancer and non-cancer studies) to the early/late classifications used by SPRINTER (Fig. R6a,b below). In this new analysis, we thus demonstrated that the expected fraction of replication timing differences and alterations is always <15%, well within the range at which SPRINTER remains robust.

Furthermore, the fractions of replication timing differences estimated with our new analysis are in line with previous studies, which estimated that altered replication timing (ART) between tumour and normal tissue is present across 5-18% of the genome in various cancer types^{25-27,29}. Moreover, some of the authors of this study have also previously estimated the degree of ART in lung and breast cancer compared to matched normal cells-of-origin as 6-18% in a recent replication timing study²³. Even at a single-cell level, replication timing has been shown to be highly conserved, including in cancer cell lines^{31,37,38}. Given replication timing alterations occur in <18% of the genome in cancer, the degree of error in the classification of replication timing from the reference profile is estimated to be <18%, and SPRINTER is thus expected to be clearly robust to these variations in replication timing in individual tumours.

Fig. R6: SPRINTER is robust to up to 40% of errors or alterations in replication timing. **a**, The expected fraction of replication timing errors (x-axis) was estimated for each available replication profile (y-axis) by quantifying the fraction of the genome with a classification of early/late replication timing different to SPRINTER's reference replication timing classifications used by default. **b**, The expected fraction of replication timing errors (x-axis) was computed with the same method used in **(a)** but using 20 bootstrapped repeats for each profile, such that in every repeat the genomic regions to be used in the comparison were randomly sampled with replacement. A beta distribution (dotted line) was fitted to the bootstrapped values to estimate the density of the distribution. **c-d**, The proportion of S phase cells that are correctly identified by SPRINTER (y-axis) after introducing a varying fraction of errors in the replication timing classifications (x-axis) was calculated using 500 S phase cells randomly sampled from either the diploid **(c)** or tetraploid **(d)** ground truth dataset for 100 bootstrapped repeats (lines). The 95% confidence interval of the expected fraction of errors as calculated in **(b)** is shown (red box). **e-f**, For each experiment in **(c)** and **(d)**, the proportions of S phase cells that are correctly identified by SPRINTER (y-axis) after introducing a varying fraction of errors in the replication timing classifications (x-axis) are separated by cell cycle phase for diploid **(e)** and tetraploid **(f)** cases, respectively.

It is important also to remark that genomic regions that do not display conserved replication timing are discarded during the identification of S phase cells by SPRINTER, since these regions might be under developmental control and thus are more likely to have varied replication timing across different tissue types^{24,28,35,36,39,40}. SPRINTER identifies and ignores these regions during S phase identification using the reference replication profiles (Supplementary Fig. 1), hence the degree of errors in replication timing classifications compared to the reference are expected to be mostly limited to ART seen in individual tumours. Importantly, we have also performed experiments showing that SPRINTER's accuracy in identifying replicating cells in the ground truth datasets does not significantly change when using different reference replication profiles, further indicating that SPRINTER's performance is not affected by limited changes in the input replication timing information (Supplementary Fig. 16).

Despite the expectations of SPRINTER's robustness demonstrated above, we agree with the reviewer that it remains important to estimate the degree of replication timing heterogeneity in the analysed tumour samples, and especially between different tumour clones with potentially different genetic and/or epigenetic alterations. To do this, we have now introduced a new feature of the SPRINTER algorithm in this revision that allows the estimation of ART, i.e., changes in replication timing between the tumour and the reference replication profile derived from normal cells, in distinct clones. Specifically, we leveraged the S phase cells assigned to each clone independently by SPRINTER to identify clone-specific ART by applying approaches similar to previous replication timing studies^{31,38} to each clone separately. To further support the identified ART, we have also performed a new gene set variation analysis⁴¹ using previous matched bulk RNA-sequencing data generated for regions obtained from the same primary tumour³⁰ (new Supplementary Fig. 47, reproduced in Fig. R7 below). This is because late-to-early and early-to-late ART is known to be associated with a relative increase or decrease in gene expression compared to normal tissue without ART, respectively^{23,24}. As such, we observed that gene sets with late-to-early or early-to-late ART identified in any of the analysed samples in this tumour have high and low gene expression enrichment scores, respectively, in every available sample from patient CRUK0516 (Fig. R7a below). To show that these results are specific to the inferred ART for this patient, we also showed that arbitrary scores are obtained when using gene expression data obtained from 915 tumour samples from 347 other TRACERx patients, which are likely to have different tumour-specific ART events (Fig. R7b below). The details of this new method that leverages SPRINTER's results for the identification of clone-specific ART are now described in the new section "*Identifying clone-specific altered replication timing*" in the revised Methods.

Fig. R7: Gene set variation analysis using bulk RNA-sequencing in the TRACERx cohort supports the inferred altered replication timing for patient CRUK0516. **a**, Gene set variation analysis enrichment scores (colours, with higher scores indicating increased expression, and lower scores decreased expression) for the set of genes inferred to have ART in at least one sample from patient CRUK0516 in either the late-to-early or the early-to-late direction (y-axis) in each CRUK0516 tumour sample with available bulk RNA-sequencing data (including primary tumour regions and a pre-mortem recurrence sample in the left adrenal, x-axis). **b**, Gene set variation analysis enrichment scores (colours) calculated for the same gene sets with ART as defined in CRUK0516 in (a), but for all tumour samples (total 915 samples) with available RNA-sequencing data (x-axis) from 347 other TRACERx patients (top bar).

Using this approach, we have estimated that up to 10% of the genome on average across all clones in the NSCLC dataset is affected by ART (new Supplementary Fig. 45, reproduced in Fig. R8 below). Therefore, SPRINTER is expected to be highly robust to tumour-specific ART in the analysed cancer cells, since the measured ART fraction of <10% is well below SPRINTER's level of robustness (<40%) and is clearly consistent with the expected rate of ART in cancer from previous studies (5-18%), as detailed above. Additionally, we identified ART events affecting genes involved in cancer proliferation and metastatic potential that are shared or unique to different metastatic clades in the NSCLC dataset, also supported by related changes in gene expression. This new result further enhances the key novel insights into metastatic evolution in our study. The details of this new analysis are described in Response 1.5 above.

Fig. R8: Clone-specific altered replication timing (ART) in the NSCLC dataset affects <10% of the genome on average as expected from previous studies. The fraction of clones affected by ART was calculated by combining the fractions of clones affected across all samples (y-axis) based on SPRINTER's clone-specific results in the NSCLC dataset for either late-to-early (positive values, magenta) or early-to-late (negative values, green) ART in 50kb genomic bins along the genome (x-axis, with autosomes separated by dashed lines). ART was inferred only in high-confidence cases (i.e., only ART events present in most clones in >2 samples). Known cancer oncogenes in late-to-early genomic regions and known cancer tumour suppressor genes in early-to-late regions (from the COSMIC Cancer Gene Census) are annotated (black text and lines), also including tumour- and metastatic-clade-specific ART events affecting genes in the performed expression analysis (e.g., PDL1, CDK12, NCOA1, KRAS).

Finally, to further address the reviewer's concern regarding the use of reference replication profiles in individual tumours with ART, we have performed a new experiment by re-running SPRINTER in each sample of the NSCLC dataset after excluding bins for which high-confidence tumour-specific ART has been identified (i.e., present in most clones from >2 samples). As such, we showed that SPRINTER infers nearly identical estimates of S fractions for most clones analysed in the NSCLC dataset when excluding tumour-specific ART, indicating that SPRINTER's results are robust to the presence of tumour- and clone-specific ART (new Supplementary Fig. 46, reproduced in Fig. R9 below).

Fig. R9: SPRINTER's estimates of S fractions are not affected by clone-specific altered replication timing (ART). The bootstrapped estimate of the S fraction of each clone (y-axis) was calculated by sampling with replacement cells in each SPRINTER-inferred clone from each sample of the NSCLC dataset (x-axis) either using the default replication timing classification (green) or excluding genomic regions in which high-confidence ART events (found in most clones of >2 samples) have been identified (orange).

Overall, we have demonstrated that SPRINTER is robust to high rates (up to 40%) of errors and alterations in the classification of early/late replication timing. Since we also showed that the expected fraction of replication timing differences in cancer and non-cancer samples is <18% in line with previous studies, SPRINTER is applicable to samples with ART and its results are robust to tumour- and clone-specific ART events. This is further demonstrated by the consistency of SPRINTER's results when excluding inferred tumour-specific ART. Given this, we believe the use of reference replication profiles is in fact a key strength of SPRINTER, since they are used within a statistical framework that accounts for the presence of tumour-specific alterations and errors in replication timing. Further support for this observation is provided in Response 2.6 below.

Response 2.3 Sup fig 2 shows the average of several cell lines in each panel, so any differences would be masked out. A correlation matrix including all samples would be informative to include as well.

Following the reviewer's recommendation, we have now calculated correlation matrices for each set of cell lines that can be used as a reference replication profile by SPRINTER and we have added these results in a new Supplementary Fig. 4 (reproduced in Fig. R10 below). Overall, this new analysis confirms that every pair of reference profiles displays high correlation, with a Pearson correlation coefficient of ~0.79 on average when calculated using all possible genomic regions (Fig. R10 below). Importantly, the correlation is even higher, with a Pearson correlation coefficient of ~0.83 on average, when only considering the genomic regions with conserved replication timing selected by SPRINTER using the permissive thresholds (± 0.5 in the replication score) applied by default in this study (Fig. R10 below).

Fig. R10: All pairs of reference replication profiles display high correlation. For the three subsets of replication profiles that can be used by SPRINTER (top, middle, and bottom), the Pearson correlation coefficient (colour) is calculated for each pair of profiles (rows and columns) either considering all genomic regions (left) or only genomic regions with conserved replication timing selected by SPRINTER using thresholds of +/-0.5 in the replication score (right).

Response 2.4 scDNA-seq data quality. Data in Figure 1 is shown at the whole-genome level. It would be useful to also present a zoomed-in view of a single chromosome or part of a chromosome to assist in evaluating the data. However, it seems from the figure that the ability to separate the genome to early and late replicating regions is very coarse, with very long areas sometimes spanning entire chromosome arms or even whole chromosomes being designated as “early” or “late”. Figure 2g has a bit more resolution but also leads to the same conclusion. This stands in contradiction to the authors’ own claim that replication timing fluctuates at the scale of ~1Mb and raises concerns about the accuracy of separating replication timing from CNAs and about the data quality in general.

We agree with the reviewer that the visualisation of the whole-genome data in the previous version of Fig. 1 was confusing. This is because the per cell read depth ratios (RDRs) previously displayed in Fig. 1 were aggregated within large early/late-specific bins per cell, making bins with the same replication timing appear closer in the genome than they actually are and not showing the correct alternation of early/late genomic regions along the genome. In this revision and following the

suggestion of another reviewer (see Response 3.6 to Reviewer 3), we have now completely reworked the binning method of SPRINTER using a sliding window approach, which allows the new version of SPRINTER to estimate RDRs for the same 50kb genomic bins that are used to estimate early/late replication timing. As a result, we can now provide a much clearer and more accurate visualisation of the data, with a direct link between RDRs and replication timing, since they are both measured for the same 50kb bins (details of the new approach are reported in Response 3.6).

Using the new binning approach of SPRINTER, we have now introduced an improved visualisation of the data in Fig. 1 (new Supplementary Fig. 7, reproduced in Fig. R11 below) and we have also added the requested examples of zoomed-in figures for some chromosomes in a new Supplementary Fig. 8 (reproduced in Fig. R12 below). Both of these new figures now clearly show that early/late genomic regions alternate across the entire genome and confirm the high quality of the generated data: while early/late genomic regions are more prevalent in certain domains, both these new figures clearly show that the alternation of early/late genomic regions is frequent across most of the genome. To formally demonstrate this observation, we have also performed new analyses in this revision to measure the size of consecutive genomic regions with the same replication timing, showing that the size of consecutive early/late genomic regions is <1Mb on average across the entire genome. The details of these new analyses are reported in Response 3.7 below. We highlight that this observation is not new or unexpected, but it is consistent with multiple previous studies of replication timing^{23,24,28,35,36,40}, as also confirmed by another reviewer of this study (see Response 3.7 below).

Overall, these new analyses further confirm our claim that consecutive genomic regions with the same replication timing are generally <1Mb in size and are substantially shorter than the CNAs measured in single-cell studies, which are nearly always >3Mb (also formally shown in a new analysis reported in Fig. R17 below and described in Response 3.7). To provide even further evidence for SPRINTER's accuracy in discerning CNAs from replication timing fluctuations, we have also performed a new spike-in experiment of small CNAs of >3Mb in both S and non-S phase cells, confirming that SPRINTER can recover these CNAs with high accuracy even in S phase cells (details are reported in Response 3.7 and Fig. R19 below). Moreover, we have introduced a new feature in SPRINTER that allows the correction of CNAs that might be missed in certain S phase cells by using the copy numbers inferred for the G0/1/2 phase cells assigned to the same clone (details in Response 3.7 and Fig. R19 below).

a Diploid ground truth: **G1/2 phase cells**

b Diploid ground truth: **mid-S phase cells**

c Tetraploid ground truth: **G1/2 phase cells**

d Tetraploid ground truth: **mid-S phase cells**

Fig. R11: S phase cells display a clear difference in read depth ratios (RDRs) between early and late genomic regions in contrast to G1/2 cells. Average RDRs (y-axis) are estimated by SPRINTER in 50kb genomic bins with early (magenta) or late (green) replication timing along all autosomes (x-axis) in either the diploid (**a** and **b**) or tetraploid (**c** and **d**) ground truth datasets and across either G1/2 (**a** and **c**) or S (**b** and **d**) phase cells.

Fig. R12: Early and late genomic regions are distributed across the entire genome and within chromosomes, displaying clear differences in read depth ratios (RDRs) between early and late genomic regions in S phase cells in contrast to G1/2 cells. RDRs (y-axis) are estimated by SPRINTER in 50kb genomic bins with early (magenta) or late (green) replication timing along all autosomes (x-axis in top panels) or in example chromosomes (bottom panels) for different example cells that belong to either the diploid (a and b) or tetraploid (c and d) ground truth datasets and are either in the G1/2 (a and c) or S (b and d) phase of the cell cycle.

Response 2.5 If the scDNA-seq data was of high quality, I would expect the copy number fluctuations in S phase cells to be self-evident. The need to utilize reference replication timing profiles raises further concern that the data is of moderate quality; the use of external data only partially compensates for that and, while it provides a means of classifying S phase cells (and is the reason for the reported superiority over previous methods that called S phase cells *ab initio*), in certain circumstances may be suboptimal, as explained above.

The new, improved visualisation of the whole-genome data (details explained in the previous Response 2.4 above) clearly shows that replication fluctuations are self-evident both across cells (Fig. R11b,d above) and at the individual cell level (Fig. R12b,d above) in S phase cells in our generated data. In addition to replication fluctuations, RDRs in S phase cells are also affected by CNAs similarly affecting both early and late regions (e.g., see chromosome 7 in Fig. R12d).

Conversely, G1/2 phase cells display a clean profile without significant differences between early/late genomic regions and with a genome clearly partitioned into only a few distinct segments due to the effect of CNAs (Figs. R11a,c and R12a,c above). Taken together, these observations clearly demonstrate the high quality of the generated data.

In addition, we highlight that previous methods that do not use reference replication profiles were able to identify mid S phase cells in our generated ground truth datasets with an accuracy comparable to, or higher than, that achieved in datasets generated in previous studies³¹ (see Fig. R13 below for mid S phase accuracy in our datasets, which is similar to, or higher than, the accuracy achieved in previously generated datasets in Fig. R14 below). In addition to further confirming the high quality of our generated data, this observation also confirms that the use of reference replication profiles is not required to achieve medium-high accuracy in recovering mid S phase cells from our generated dataset (in contrast to early and late S phase cells), similar to previous studies. For example, MAPD recovers mid S phase cells with ~60-70% accuracy in the diploid ground truth dataset without the use of replication profiles (Fig. R13 below), which is similar to the accuracy achieved by the same method on previous datasets (Fig. R14 below). Furthermore, we note that the use of replication timing profiles is not suboptimal when the expected fraction of replication timing differences compared to the reference profiles is below the levels of robustness observed in SPRINTER, as demonstrated for cancer and non-cancer cells (see Response 2.2 above) and, especially, when applying SPRINTER's robust statistical approach to specifically account for the presence of these differences (related details are described in Response 2.2 above).

Response 2.6 Success of SPRINTER and comparison to other methods. The comparison to CCC and MAPD is somewhat of an apples and oranges comparison. I agree that the use of reference replication profiles would usually provide a superior means to call S phase cells (and supports the premise of this paper), however CCC and MAPD don't do that by design so clearly the results would be different. A more useful comparison would be to use CCC or MAPD together with the utilization of reference replication profiles, or SPRINTER with or without the use of reference replication profiles. It would be useful for this manuscript to directly establish the utility of reference replication profiles this way.

We agree that the integration of reference replication profiles is one of the novel features of the SPRINTER algorithm, allowing it to achieve improved accuracy in identifying S phase cells compared to previous methods. However, the SPRINTER algorithm is not a straightforward extension of previous methods that simply integrates reference replication profiles. Rather, the SPRINTER algorithm is a novel and comprehensive multi-step algorithm in which every step of traditional single-cell copy-number analysis has been fully re-designed to incorporate prior knowledge of early/late replicating genomic regions and to specifically deal with the expected read count differences between early and late genomic regions in replicating cells. As an example, SPRINTER's replication-aware segmentation is a completely different approach compared to those applied by previous methods, since it is composed of multiple parallel segmentations (details and comparison with previous approaches are reported in the subsection "*Replication-aware copy-number segmentation*" in Methods). Furthermore, we highlight that SPRINTER identifies S phase cells by using a novel statistical hypothesis testing approach that is applied to each cell independently

(details and comparison with previous approaches are reported in the subsection “*Identifying S phase cells*” in Methods). This cell-specific test is also a different approach to those applied by previous methods, which identify S phase cells by aggregating all the cells sequenced from the same sample and classifying replicating cells as those that clearly deviate from the mode of the sample based on different summary statistics (more details about this point and limitations of previous approaches are described in Response 2.7 below).

Following the reviewer’s suggestion, in this revision we have now performed a new analysis to demonstrate the advantages of SPRINTER’s novel features by introducing to our benchmarking comparison a new version of a previous method that integrates reference replication profiles. Specifically, we have extended the existing MAPD method into a new version called replication timing MAPD (rtMAPD), which calculates the median absolute deviation of pairwise differences (i.e., the summary statistic used by MAPD) only between neighbouring genomic windows with different replication timing (i.e., early vs late). Also, note that the CCC method was already using a form of reference profiles because it is a machine learning algorithm that has been trained on an extensive collection of FACS-sorted datasets with labelled replicating cells, thus representing the corresponding reference. As such, we compared all methods on the diploid and tetraploid ground truth datasets and found that SPRINTER still provides a substantial improvement in performance compared to all other methods (Fig. 2 with all updated analyses, reproduced in Fig. R13 below). While rtMAPD improved performance compared to MAPD, demonstrating the utility of reference replication profiles, SPRINTER identifies a substantially higher number of S phase cells without sacrificing precision. Overall, these results demonstrate that a straightforward integration of reference replication profiles into previous methods only results in a marginal improvement in performance and the sophisticated algorithm introduced in SPRINTER is needed to substantially improve the accuracy of identifying S phase cells.

Fig. R13: SPRINTER outperforms previous methods in the identification of S phase cells even when previous methods are extended to integrate replication timing. **a-b**, The proportion of correctly identified G1/2 and S phase cells (y-axis) was computed for SPRINTER (green) and three previous methods (CCC in blue, MAPD in orange, and rtMAPD, which extends MAPD with replication timing, in red) across four cell cycle phases (x-axis) for 100 subpopulations of cells (each dot), each formed by randomly sampling 500 cells from a total of either **(a)** 4,410 diploid or **(b)** 4,434 tetraploid HCT116 cells. **c-d**, ROC curves (false positive vs. true positive rate) were computed to measure performance in distinguishing G1 cells from actively replicating cells by using the classification scores computed by existing methods (blue, orange, and red) or combining SPRINTER's S and G2 p-values (using the minimum in green) for either **(c)** each of the 100 diploid subpopulations in **(a)**, or **(d)** each of the 100 tetraploid subpopulations in **(b)**.

We also note that SPRINTER cannot be executed without replication timing by design. However, we have performed new extended experiments in which SPRINTER is executed with varying fractions of errors in the replication timing classifications (details are in Response 2.2 above) and showed that SPRINTER's performance is robust to a high rate of errors up to 40% (Fig. R6 above). Therefore, this experiment demonstrates that SPRINTER does not rely on perfect reference replication profiles and is robust to a high rate of alterations or errors in replication timing classifications as a result of the statistical framework employed (details are in Response 2.2 above).

Moreover, we highlight that we have demonstrated in previous experiments that the identification of S phase cells based on the cell aggregation used by previous methods (especially MAPD) is ill-suited to the identification of S phase cells in samples that contain admixtures of cells with different ploidy (see experiment in Supplementary Fig. 20, described at the end of the third paragraph of the section "SPRINTER exhibits high accuracy and sensitivity" in Results). This is a limitation that

cannot be fixed by integrating reference replication profiles into previous methods, but is overcome by the cell-specific test introduced by SPRINTER (Supplementary Fig. 20). We highlight this important point in a new sentence in the related paragraph of Results:

“Furthermore, by applying all methods to an additional dataset of 4,163 diploid and tetraploid unsorted cells, we found that methods that aggregate all cells together for S phase identification, especially MAPD, failed to accurately identify S phase cells in heterogeneous samples containing cells with different ploidy levels (both diploid and tetraploid cells), in contrast to SPRINTER, which preserved its accuracy (Supplementary Fig. 20). This is because distinct subpopulations of cancer cells with different ploidy levels within the same sample might require the use of different thresholds for accurately classifying S phase cells. This result further highlights that it is not only the use of replication scores but also the other novel features of SPRINTER, especially the cell-specific test for S phase cells, that are key to enabling accurate S phase identification in heterogeneous cancer samples.”

Lastly, we also highlight that the key novel feature introduced by SPRINTER is the assignment of replicating cells to the corresponding tumour clone. This is a key feature that is not provided by any other existing method, and it is made possible by the specific integration of replication timing information into other steps of the SPRINTER algorithm in addition to S phase identification. Therefore, the specific way in which SPRINTER uses the reference replication profiles (which is different to a straightforward integration in previous methods, like the proposed rtMAPD) does not only provide improved identification of S phase cells, but is also needed to enable the assignment of replicating cells to tumour clones.

Response 2.7 Currently, the presentation could mislead some readers into preferring one approach over the other without being aware of the context in which each method was developed or can be applied. I would thus recommend more careful phrasing of this section. This is not to say that SPRINTER doesn't work as well as it is advertised to; it is to say that other methods have their power as well. As part of this, I don't think “threshold-based” is an accurate way to differentiate SPRINTER from previous methods, because doesn't SPRINTER also use thresholds?

We agree with the reviewer that the results in this study do not imply that previous methods should be discarded. In fact, our study demonstrates that previous methods can still provide a medium-high accuracy of 60-90% in the identification of mid S phase cells (Fig. R13 above). Therefore, existing methods like CCC and MAPD are still useful, especially in studies that mostly require the identification of mid S phase cells, for example in studies of replication timing in single cells. Following the reviewer's suggestion, we clarified this point in a new sentence in Results:

“At the same time, previous methods displayed generally high accuracy in the identification of mid S phase cells, indicating that these methods remain useful in studies that mostly rely on the identification of these cells, such as replication timing studies³¹.”

Additionally, we also agree with the reviewer that the expression “threshold-based” was confusing and incorrect in the differentiation between SPRINTER and previous methods. Our point is that most previous methods base their classification of S phase cells on the use of a sample-specific threshold that is calculated by aggregating all the cells sequenced from the same sample. While this approach has been shown to be successful in homogeneous samples with cells of equal ploidy, we expect that

the same approach could fail in heterogeneous samples, or samples containing cells with different ploidy levels. This is because distinct subpopulations of cancer cells with different ploidy levels within the same sample might require the use of different thresholds to accurately classify S phase cells. We demonstrated this issue for MAPD in a previous analysis of a mixed population of cells (Supplementary Fig. 20). In contrast, SPRINTER overcomes this challenge by introducing a cell-specific statistical test for the identification of S phase cells that can be applied to each cell independently. As such, we showed that SPRINTER's approach is not affected by the same limitations (Supplementary Fig. 20) and it can be applied to heterogeneous samples or samples containing cells with different ploidy levels, a frequent occurrence in cancer. We improved the explanation of this observation in both the Introduction and Results:

“Second, they assume that the sequenced cells belong to a homogenous population and thus aggregate all the cells together, identifying S phase cells as those with some sequencing signal that deviates from the rest^{2,7,31}. While this assumption may be true in cell lines (mostly used in previous studies^{2,7,31}), this is not the case in cancer tissues that are often heterogeneous mixtures of normal and different cancer cell subpopulations^{4,42-44}, such that each subpopulation may need to be treated differently for S phase identification.”

“Furthermore, by applying all methods to an additional dataset of 4,163 diploid and tetraploid unsorted cells, we found that methods that aggregate all cells together for S phase identification, especially MAPD, failed to accurately identify S phase cells in heterogeneous samples containing cells with different ploidy levels (both diploid and tetraploid cells), in contrast to SPRINTER, which preserved its accuracy (Supplementary Fig. 20). This is because distinct subpopulations of cancer cells with different ploidy levels within the same sample might require the use of different thresholds for accurately classifying S phase cells. This result further highlights that it is not only the use of replication scores but also the other novel features of SPRINTER, especially the cell-specific test for S phase cells, that are key to enabling accurate S phase identification in heterogeneous cancer samples.”

Response 2.8 It is also important to clarify that the sorting of G1, early, mid and late S, and G2 cells (sup Figure 7b) was not continuous, i.e. the sorted bins were discrete and separated from each other, thus only a subset of all cells were sorted, rather than choosing continuous bins that together sort all cells or at least all S phase cells. This choice of sorting gates is acceptable but leads to inflation in the estimation of the accuracy of SPRINTER in calling cells at different cell cycle stages. It leads to a disconnect between the data quality (rather noisy for calling S phase cells) and the reported success in calling cell cycle stages. It also creates a meaningless comparison to previous approaches, which used different sorting gates or thresholds. This all needs to be clarified and phrased in an accurate way as to not mislead readers to over evaluate the accuracy of SPRINTER and the current scDNA-seq data (or the limitations of previous approaches).

We have clarified the discrete sorting of our ground truth datasets by adding a related sentence in the section *“SPRINTER exhibits high accuracy and sensitivity”* in Results:

“Compared to previous studies^{2,31}, our approach is based on the use of discrete gates rather than continuous sorting. While this discrete approach does not sample cells as comprehensively as previous approaches, it provides improved precision in the classification of the different phases.”

However, we note that all the benchmarked methods in this study have been executed on the same data using the same true labels for cells in G1, G2, early S, mid S, and late S phase (i.e., defined by

the same gates). Therefore, the accuracy of SPRINTER has not been inflated compared to all the other methods benchmarked in this study.

Even more importantly, we highlight that, in this study, we have also used a previous phase-sorted scDNA-seq dataset³¹ of 5,970 cells from the lymphoblastoid cell line GM12878 to compare SPRINTER with all other methods. In this revision, we have now re-executed all methods on this previous dataset, using the new version of SPRINTER and including rtMAPD, i.e., the replication timing version of MAPD (requested as part of Response 2.6 above). We report this new analysis in the updated Supplementary Fig. 18 (reproduced in Fig. R14 below). Similar to the results on our ground truth datasets, the analysis of this previous dataset confirms SPRINTER's improved accuracy in identifying S phase cells compared to all other methods (~70-80% for SPRINTER vs 40-60% for other methods in Fig. R14 below). Moreover, both MAPD and SPRINTER identify likely S phase cells infiltrating the experimentally sorted G1/2 phase cells (Fig. R14b,c below), which have been demonstrated to correspond to sorting errors in the previous analysis of this dataset³¹. This analysis based on a previous dataset provides further evidence that the improved accuracy observed with SPRINTER is not specific to the ground truth datasets generated in this study, but rather generalises to other datasets generated with different sorting approaches.

Related to the data quality, we note that we have now clarified previous issues with data visualisation and have demonstrated the high quality of the newly generated data in previous responses (see Response 2.4 above), especially for S phase cells (Fig. R11 and R12 above).

Fig. R14: SPRINTER exhibits higher accuracy and sensitivity in the identification of S phase cells than existing methods in a previously published 10X CNV Solution dataset. **a**, The proportion of correctly identified G1/2 and S phase cells computed for SPRINTER and existing approaches (CCC, MAPD, and rtMAPD, which extends MAPD with replication timing) for 100 subpopulations of cells (each dot), each formed by randomly sampling 500 cells from a previous ground truth dataset of 5,970 cells from the lymphoblastoid cell line GM12878 sorted with standard FACS and sequenced using the 10x CNV Solution. **b-c**, The average RDR (y-axis) was measured in 250kb bins with either early (magenta) or late (green) replication timing across autosomes in the genome (x-axis) for all G1/2 cells in the previous ground truth dataset that were inferred as either **(b)** G1/2 phase or **(c)** S phase by SPRINTER. In contrast to cells in **(b)** that were inferred as G1/2 cells by SPRINTER, the remaining cells that were inferred as S phase by SPRINTER and also by MAPD and rtMAPD in **(c)** exhibit clear replication fluctuations across the genome (early bins have consistently higher RDR than late bins), indicating that these cells are the result of known FACS infiltrating errors in the classification in the previous dataset and likely correspond to true S phase cells, in line with SPRINTER's results and previous reports in the same study. **d-e**, The average RDR (y-axis) was measured in 250kb bins with either early (magenta) or late (green) replication timing across autosomes in the genome (x-axis) for all S phase cells in the previous ground truth dataset that were inferred as either **(d)** G1/2 phase or **(e)** S phase by SPRINTER. In contrast to cells in **(e)** that were inferred as S phase cells by SPRINTER, the remaining cells in **(d)** that were inferred as G1/2 phase by SPRINTER do not exhibit clear replication fluctuations across the genome, similar to other G1/2 cells in the dataset in **(b)**, indicating that these cells are the result of known FACS infiltrating errors in the classification in the previous dataset and likely correspond to true G1/2 phase cells, in accordance with SPRINTER's results.

Response 2.9 It's also unclear how the authors used CCC and MAPD to call G1 vs G2 cells- did they rely on total sequencing depth as in their SPRINTER pipeline? Can the authors clarify this and the reason for the superior performance of SPRINTER in this respect?

While the previous methods benchmarked in this study do not explicitly infer G2 phase cells, the classification scores calculated by these methods account for, and are influenced by, the total sequencing depth per cell. Therefore, as suggested in previous studies², one can hypothesise that these scores might also be used to separate G2 from G1 phase cells, in addition to separating S from G1 phase cells. In our benchmark, we have thus tried every possible threshold for the scores computed by previous methods using an ROC analysis to assess the potential of these methods to separate G1 phase cells from S and G2 phase cells (updated Figs. 2c,d in the revision, reproduced in Figs. R13c,d above). To clarify this aspect, we have now added a new sentence in the section “*Bioinformatics analysis of single-cell data*” in Methods:

“For each previous method, all the possible thresholds for the classification scores calculated by each method have been tested as part of the ROC analysis to assess the performance of separating G1 from S and G2 phase cells.”

In this analysis, we found that SPRINTER provides improved accuracy in separating G1 phase cells from G2 and S phase cells compared to the classification scores computed by the other methods (ROC AUC 87-88% for SPRINTER vs 64-82% for previous methods, Figs. R13c,d above). The reason for the superior performance is that SPRINTER is the only algorithm that includes a specific probabilistic model for the total read counts expected from G2 phase cells, which can deal with both specific sequencing features of different samples and the presence of clones with different ploidies. In fact, SPRINTER introduces an importance sampling method based on a Negative Binomial model to specifically model and predict the expected read counts of both G1 and G2 phase cells per clone, while being robust to variations in different samples and clones. The details of this algorithm are reported in the subsection “*Identifying G2 cells in distinct clones*” in Methods.

Moreover, we previously used the default value of the classification threshold for previous methods to quantify the fraction of correctly identified G2 phase cells, such that cells in G2 phase in the ground truth datasets were classified as correctly identified by previous methods if their classification scores were higher than the default threshold. However, we agree that this approach was confusing, as these methods do not explicitly infer G2 phase cells and their thresholds have not been specifically designed for this purpose. Therefore, while preserving the previous ROC analysis that is still informative for investigating the potential for classifying G2 phase cells (Figs. R13c,d above), we have now focused the analysis of the exact accuracy of identifying G2 phase cells on SPRINTER. Specifically, we have now performed a new analysis to investigate both the precision and recall of SPRINTER in identifying G2 phase cells, finding that SPRINTER achieves accurate results with >80% precision and recall on both the diploid and tetraploid ground truth datasets (new Supplementary Fig. 19, reproduced in Fig. R15 below).

a SPRINTER's performance for G2 diploid cells

b SPRINTER's performance for G2 tetraploid cells

Fig. R15: SPRINTER accurately infers G2 phase cells in both the diploid and tetraploid ground truth datasets. (Left plots) The precision and recall for G2 phase cells were computed by bootstrapping for 100 repeats the G1/G2 phase cells identified by SPRINTER from a total of either (a) 4,410 diploid or (b) 4,434 tetraploid HCT116 cells. (Right plots) ROC curves were calculated to measure the performance of SPRINTER in distinguishing G1 vs G2 phase cells for either the diploid or tetraploid cells analysed in (a) or (b), respectively, by using the G2 p-values computed by SPRINTER.

Response 2.10 Other comment: It is important to acknowledge that fraction of cells in S phase (or S and G2 phases) isn't a direct representation of the level of proliferation rate, since the length of individual cell cycle stages could also vary between subclones or samples. Could the authors comment on that?

We agree with the reviewer that variations in the fraction of cells in S or G2 phase can generally be influenced by relative changes in the length of individual cell cycle phases if these lengths are altered across distinct clones or tumours. At the same time, we also note that the same limitation holds for Ki-67-based approaches, which are considered the gold standard approach for measuring proliferation at the sample level and are implemented in routine clinical practice^{9,10,12,14}. However, in our NSCLC case, we were able to confirm that the changes identified in S fractions among clones correspond to changes in proliferation rates, as they match the tumour growth rates measured using computed tomography (CT) and magnetic resonance (MR) imaging acquired for five longitudinal timepoints. We have now added a new sentence to clarify this important point in the related section of Results:

“Consistent with the expectation that higher proliferation results in increasing disease burden, we found the average growth rates of the left adrenal, left frontal lobe, right adrenal, and right occipital lobe metastases (3.34, 1.94, 1.1, and 0.43 log(mm³/day), Supplementary Fig. 37) followed the same order as the average S fractions estimated by SPRINTER (40.9%, 37.1%, 30.8%, and 13.4%). Additionally, these results confirm that increased sample S fractions relate to increased proliferation, rather than changes in the length of cell cycle phases.”

Moreover, we have now performed an improved analysis of the correlation between SPRINTER's estimated fractions of S and G2 phase cells (details in Response 3.11 below), confirming their high correlation across samples and clones, thus indicating that higher S fractions are more likely to relate to increased proliferation than changes in the relative length of cell cycle phases in different clones (new Supplementary Fig. 33, reproduced in Fig. R24 below, with further explanations in Response 3.11 below).

In the TNBC and HGSC datasets, we have now also performed a new analysis of the ratio between G2 fractions and S fractions to specifically investigate potential changes in the length of G2 phase relative to the length of the S phase, as suggested by another reviewer (Reviewer 3 below). The details of this new analysis are reported in the Response 3.11 below.

We also agree with the reviewer that acknowledging this observation is important for future users of the SPRINTER algorithm, who should generally account for this consideration in their future analyses. Therefore, we have added a new related comment in the revised Discussion:

“In this study, we showed that investigating the relative fractions of S and G2 phase cells can provide evidence of potential changes in the relative length of cell cycle phases. Therefore, improvements in G2 identification and more robust models of the cell cycle could enable further investigation of these changes throughout cancer evolution and across different tumours.”

Reviewer 3

Response 3.1 Key results. The authors develop a new algorithm, SPRINTER, with the purpose of using single cell whole genome sequencing to deduce mutational heterogeneity and clone-to-clone proliferation rate differences in cancer samples. The authors first develop and test SPRINTER using a large scWGS dataset of HCT116 cancer cells that the authors generated themselves. SPRINTER is shown to better deconvolute S-phase DNA replication contribution to cell-to-cell copy number variability and thus improve single cell CNV calling, mutational clone assignment, and accurate measurement of clone-specific proliferation levels, compared to previous algorithms. The authors then apply SPRINTER to patient tumour samples that they generated for this paper (NSCLC patient), showing that estimates generated by SPRINTER match clinical features and outcomes of the tumours (e.g. primary vs. metastasis). Lastly, the authors also applied SPRINTER on published breast and ovarian cancer scWGS samples. The major conclusions of the cancer studies are that higher proliferation is correlated with increased metastatic potential and/or mutational burden, as well as increased ctDNA shedding.

Validity, originality and significance. SPRINTER improves on previous methods, especially for S-phase and G2 assignment of cells. In particular, the aspects that I see as an improvement are using constitutive replication timing (RT) information to define S-phase state of each cell and the replication aware correction of GC bias. In general, the authors have made logical and effective decisions in their method for identifying S-phase cells, separating CNV signal from DNA replication signal, and using all this information to assign cells to mutational clones. There are some clarifications of the method detailed in the Major Comments section below.

We thank the reviewer for recognising the advancements and motivation of our novel method. In this revision, we have addressed the reported questions in the responses below by adding new clarifications, new features to the SPRINTER algorithm, and new analyses and experiments to demonstrate the robustness of the method.

Response 3.2 The authors also generated two useful datasets for the field. The HCT116 diploid and tetraploid datasets with more accurate S-phase information could be used by the authors or others to further develop scWGS cell cycle inference methods that could include DNA replication timing, as described in the discussion.

We thank the reviewer for recognising the importance and utility to the community of the newly generated ground truth datasets.

Response 3.3 The NSCLC patient dataset is an impressive collection of sequencing and clinical data, with some temporality in the sequencing datasets e.g. samples taken at primary tumor surgery and samples taken at autopsy. This dataset has the potential to become a valuable in-depth case study, and the authors use this dataset to a good degree to showcase their method.

We thank the reviewer for recognising the novelty and value of the newly generated, longitudinal, primary-metastasis matched cancer dataset with detailed clinical annotation.

Response 3.4 However, the authors focused on results that agree with previous knowledge, e.g. association between high clone proliferation and metastatic seeding potential, and more proliferative clones shed more ctDNA. I assume this is because the data is only from one patient and the authors' focus was to highlight their method.

We agree with the reviewer that the focus has been on highlighting the novel biological applications enabled by the SPRINTER algorithm on a single metastatic patient used as a proof of concept. However, our results on the association between high clone proliferation and increased metastatic seeding potential of these specific clones (i.e., showing that high proliferation clones comprise the subset of cancer cells most likely to be responsible for seeding metastases), as well as the link between high clone proliferation and increased clone-specific shedding of ctDNA, have an important novelty compared to previous bulk-based studies: our results demonstrate that these associations are true for genomically distinct tumour clones present within the same tumour in the same patient. While previous studies^{20,21} have shown that high proliferation tumours in different patients shed more ctDNA and that this is associated with worse outcomes, our study shows for the first time that these associations also apply to distinct tumour clones that co-exist within the same primary tumour in the same patient. Importantly, our results which suggest that high proliferation clones have an increased metastatic seeding potential were not necessarily expected based on some previous biological observations. While these results are consistent with the worse outcomes observed for high proliferation tumours^{8-12,14,15}, some studies have suggested that disseminating cells may undergo epithelial mesenchymal transition (EMT), which has been associated with a more invasive but less proliferative phenotype¹⁶⁻¹⁹. Moreover, our results are also important because they suggest that the prevalence of ctDNA within a patient might not necessarily relate to the volume of the corresponding clone (i.e., number of related cells) but is also influenced by its proliferation rate. Furthermore, they suggest that ctDNA might also be used to identify the presence of high proliferation tumour clones within a patient. These are all novel biological insights that advance previous knowledge. We have now added these important considerations in new sentences in the Discussion:

“By generating a novel single-cell, longitudinal, primary-metastasis matched dataset, we found that SPRINTER reveals an association between high clone proliferation and increased metastatic seeding potential of specific tumour clones, with high proliferation clones comprising the specific subsets of cancer cells that are more likely to seed metastases. Importantly, while these results are consistent with the worse outcomes observed for high proliferation tumours^{8-12,14,15}, they were not necessarily expected based on some previous biological studies which suggest that disseminating cells may undergo epithelial mesenchymal transition (EMT), which has been associated with a more invasive but less proliferative phenotype¹⁶⁻¹⁹. Our results could be consistent with highly proliferative clones undergoing EMT but then returning to a proliferative state in a target organ plastically, or with these clones subverting signals that reduce proliferation during EMT.”

“If the correlation between proliferation and the metastatic seeding potential of individual clones is further confirmed in larger datasets, SPRINTER's results will motivate the development of scalable approaches to predict the metastatic potential of different tumour clones based on their proliferation, for example based on ctDNA²⁰⁻²², circulating tumour cells⁴⁵, or inexpensive methylation assays⁴⁵, thus informing future precision medicine approaches^{46,47}. In particular, ctDNA-based predictive approaches are motivated by this study since we demonstrated differential shedding of ctDNA into the bloodstream by distinct tumour clones, with high clone proliferation associated with increased ctDNA shedding, thus suggesting that ctDNA shedding could be used as a proxy for clone proliferation. Given our results indicate that the prevalence of ctDNA within a patient might not only relate to the volume of the corresponding clone (i.e., number of related cells), but may also be influenced by its proliferation rate, they also warrant the careful interpretation of ctDNA prevalence in related studies.”

In this revision, we have also performed a new analysis that for the first time reveals the presence of clone-specific altered replication timing (ART) that affects known cancer driver genes implicated in cancer proliferation or metastatic potential. We have additionally used a new analysis of matched RNA-sequencing data from the same tumour to support these findings. Furthermore, we showed that ART events are sometimes shared between different metastatic clades, or in some cases are unique to the most proliferative metastatic clade (described in detail in Response 1.5 above), adding even more novelty to the biological insights that we obtained from the analysis of the NSCLC metastatic dataset. A summary of all these novel insights in comparison to previous knowledge are reported in detail in Response 1.5 above.

Response 3.5 Lastly, the authors further showcase their method by applying it on previously published ovarian and breast cancer scWGS samples from Funnell et al 2022. I feel particularly here, there is room for more significant investigation, for example, by juxtaposing the mutational signature and heterogeneity findings in Funnell et al with clone proliferation data that SPRINTER can provide. What can SPRINTER add to the story that Funnell et al's methods could not? Beyond just showing that there is clone to clone proliferation rate variability. More areas where originality and significance could be improved are detailed in the Major Comments section below.

We agree with the reviewer that there was an opportunity for more significant investigation of the previous TNBC and HGSC datasets. Therefore, we have now performed multiple additional analyses that add novel and significant biological insights to the findings reported in the previous study from Funnell et al¹. We summarise the main new analyses and findings here in this response, and the related details are reported in the responses below (especially in Response 3.11 below):

1. We have fully reworked and improved the analysis of the associations between clone proliferation and the clone-specific rates of different genetic variants (SNV, SVs, and CNAs) by computing these rates at the individual cell level. The new analysis now clearly shows that the rates of all of these variants is significantly higher in individual cells belonging to high proliferation clones compared to cells belonging to low proliferation clones in both the TNBC and HGSC datasets.
2. We have performed a new statistical analysis to investigate the presence of CNAs and driver mutations that are enriched in high proliferation clones. As such, we identified several amplifications of oncogenes associated with high clone proliferation and demonstrated an enrichment of pathways related to proliferation in these genes. Similarly, we have also identified several driver mutations and deletions affecting tumour suppressor genes that are enriched in high proliferation clones, some of which are concordant with previous validation measurements using siRNA cell-line screens³³.
3. We have compared the ratio between the G2 fraction and S fraction (G2/S ratio) across clones, and found that tumour clones in which homologous recombination deficiency (HRD) was previously identified also have a significantly higher G2/S ratio, suggesting a prolonged length of G2 phase relative to the length of S phase in these clones, as expected by the reviewer based on previous studies^{34,48}.

Response 3.6 Major comments. 1. SPRINTER's bin size selection. The authors describe their minimum bin size, 50 kb, as it matches the typical bin size of replication timing data. However, I have not found description of whether there is a max bin size limit. This also relates to whether there is a minimum read count per cell cutoff. I understand the reasoning behind using the same number of reads per bin. However, it is not clear to me how SPRINTER can accurately predict CNA between cells, especially breakpoints, where they start and end, if every cell potentially has different sized bins. How do you relate between cells where their bin sizes are very different e.g. 50kb vs. potentially ≥ 1 Mb sized bins. I can imagine that if each cell was analyzed in a sliding window manner, this would get over the issue, but from reading the methods, this doesn't seem to be the case.

We agree with the reviewer that a sliding window approach would provide the most direct way of unifying different copy-number breakpoints independently inferred across cells with genomic bins of varying size. Therefore, in this revision we have now fully reworked SPRINTER's binning method by using a sliding window approach to estimate the read depth ratio (RDR) and the related copy number of each 50kb genomic bin in every cell. This new approach has now been implemented in two new software libraries in the SPRINTER software package, namely "estrdrs.py" and "rtcorrect.py". We describe this new binning approach by updating the related text throughout the entire manuscript, for example by updating the description in the section "*The SPRINTER algorithm*" in Results and in the section "*Deriving replication-specific DNA sequencing signals*" in Methods:

"SPRINTER improves RDR calculation in two ways: (1) aggregating the read counts of neighbouring bins within windows of cell-specific size to account for varying total read counts for cells in different phases, and (2) introducing a replication-aware correction of RDRs for GC sequencing bias, separately obtained for early and late bins."

"Since the small bins used to identify replication timing are not sufficiently long for the analysis of low-coverage scDNA-seq data^{1,2,4}, RDRs are calculated by aggregating read counts from multiple neighbouring bins. As such, the RDR of a genomic bin b is computed for each cell as follows

$$x_b = \frac{r_b}{\bar{r}_b} \cdot \frac{\bar{R}}{R}$$

where r_b is the read count for b calculated by aggregating the counts from neighbouring bins using a sliding window approach within a window of cell-specific size (explained below)[...]."

The previous version of SPRINTER inferred copy numbers for each cell independently and then mapped each result back to every individual 50kb genomic bin (based on maximum overlap) in order to unify breakpoints across cells. Since 50kb genomic bins are generally much shorter than the larger bins used to estimate RDRs and copy numbers in individual cells, the previous and newly implemented binning approaches resulted in nearly identical results. In fact, we showed that the estimated RDRs from the previous and new version of SPRINTER are nearly identical in the ground truth datasets (new Fig. R16 below). Therefore, the new binning version of SPRINTER did not change the results of our analyses. However, we agree that the sliding window approach might be better suited to future analyses of higher sequencing-depth datasets where the copy-number bins could be

closer in size to the 50kb genomic bins used for the replication data, and we have thus chosen the sliding window approach for the new version of SPRINTER.

Fig. R16: The new sliding window approach leads SPRINTER to estimate RDRs nearly identical to those obtained with the previous binning approach. RDRs are calculated using either the previous binning approach of SPRINTER (x-axis) or the new sliding window approach (y-axis) for every genomic bin considered by both versions for (a) 4,410 diploid or (b) 4,434 tetraploid HCT116 cells in the generated ground truth datasets. The colour indicates the density estimate of these values (in log₁₀ scale).

Lastly, we confirm that SPRINTER uses a minimum read count per cell cut-off, which has been set to 100k reads per cell in all the analyses in this study based on previous single-cell studies analysing data with similar per-cell sequencing coverage^{1,2,4}. We have now clarified this point in the section “*Bioinformatics analysis of single-cell data*” in Methods:

“In all cases, the default set of input replication scores has been used as input to SPRINTER and only cells with >100k sequencing reads have been selected for SPRINTER’s analysis based on recommendations from previous scDNA-seq studies⁴.”

In addition, SPRINTER includes parameters that can be used to control the maximum size of the bins used throughout the analysis. For example, the default maximum bin size for measuring the RDRs of early and late genomic regions during the identification of S phase cells has been set to 500kb for all the analyses in this study (even though nearly no cell has reached this maximum in these analyses). Note that all these parameters (including the minimum read count per cell cut-off described above) are adjustable by the user, and they can be easily changed to fit different experimental settings.

Response 3.7 2. The authors write “CNAs are expected to induce segments containing both early and late...segments that only include either early or late bins are discarded since they are likely due to replication”. The authors use quite separate cutoffs for calling Early vs. Late (± 0.5). In my experience, with this type of cutoff, Early and Late regions will usually be beyond 1Mb away from each other, therefore, the authors may be selecting for very large CNAs. Replication domains are typically said to be ~400-800Kb, however, large domains of Early and Late timing, called constant timing regions which I believe the ± 0.5 would call, can be as large as 5-6Mb with an average of around 1.5Mb in more terminally differentiated cells (Rivera-Mulia et al. 2015). To help clarify, perhaps the authors could show a graph for neighboring Early and Late bin distance vs. the distribution of bin sizes chosen for individual single cells. It makes sense why the authors would restrict to CNAs that contain both Early and Late bins, so that the method remains replication aware, but the paper needs clarity on what types of CNAs the authors are missing. For example, previous papers suggest that CNAs that occur in early replicating regions relate to recombination-based repair mechanisms such as homologous recombination (HR) (De and Michor 2011; Koren et al. 2012; Morganella et al. 2016). Additionally, there is also an overall bias for CNAs, particularly deletions, to occur in late replication timing (De and Michor 2011; Du et al. 2019; Koren et al. 2012). CNAs like these that are exclusive to one replication timing may be missed in SPRINTER.

In this revision, we have performed an extensive new analysis comparing the size of neighbouring genomic regions with the same replication timing (as measured by SPRINTER with the default cut-offs of ± 0.5 of the average replication score) to the size of the CNAs that have been inferred in previous single-cell studies¹. Specifically, we calculated the size distribution of 2,895,173 CNA segments measured from 43,105 cancer cells in the TNBC and HGSC datasets (new Supplementary Fig. 2, reproduced in Fig. R17 below). Based on this new analysis, we found that CNAs are significantly larger than genomic regions with the same replication timing, and nearly all measured CNAs (i.e., >97%) are larger than the vast majority (i.e., >90%) of the genomic regions with the same replication timing (Fig. R17a below). While the size of neighbouring genomic regions with the same replication timing has a median of 300kb (~897kb on average with 87% of regions being <2Mb in size) in agreement with the reviewer’s reports, the size of CNAs measured from previous single-cell studies has a median of 23Mb (~42Mb on average with >99.9% of CNA segments being >2Mb in size) and is thus substantially larger. These results support our claim that most CNAs inferred from single cells are expected to contain both early and late genomic bins; in fact, nearly all the CNAs measured in these datasets cover both early and late genomic bins (Fig. R17b below) and >98.7% of the genome across all these cells belong to CNA segments that contain both early and late genomic bins (Fig. R17c below). Overall, these observations support the ability of SPRINTER to recover the vast majority of CNAs present in S phase cells. Importantly, we also note that the identification of CNAs in non-S phase cells (including G0/1/2 phase cells) is not affected by replication timing and no genomic region is discarded during CNA identification in these cells (more details are provided in Response 3.8 below).

Fig. R17: CNAs inferred in single-cell studies are substantially larger than neighbouring genomic regions with the same replication timing. **a**, The size (x-axis in bases, b) of neighbouring genomic regions with the same replication timing and of 2,968,788 previously inferred CNA segments (y-axis) is reported without (left panel) or with values capped at 25Mb (right panel) as measured from 43,106 cells in 35 patients from the previous TNBC and HGSC datasets. **b**, For the same CNA segments in **(a)**, the number of segments (x-axis) within three groups with different segment sizes (y-axis) is reported (left panel) and, for each of these groups, the proportion of CNA segments covering both early and late genomic bins is also calculated (right panel). **c**, For each cell (dot) in each patient (x-axis), the fraction of the genome within CNA segments covering both early and late bins is reported (y-axis).

We have also performed a new analysis to investigate the type of CNAs that occur in genomic regions with the same replication timing in the previous TNBC and HGSC datasets, which could therefore potentially be missed by SPRINTER in S phase cells. Overall, we did not find a clear difference in the types of copy-number events that occur in segments with exclusively early or late replication timing (that may potentially be missed by SPRINTER in S phase cells) compared to segments with mixed early and late replication timing (new Fig. R18 below).

Fig. R18: CNA segments with only early or late bins are affected by similar CNA events to those with both early and late bins. The fraction of the 2,968,788 previously inferred CNA segments (y-axis) affected by five different CNA events (colour) was computed for either CNA segments overlapping only early or late bins exclusively, or both early and late bins (x-axis) as measured from 43,106 cells in 35 patients from the previous TNBC and HGSC datasets.

We also note that the distance between early and late genomic regions does not vary with bin size because the RDRs are individually estimated for each 50kb genomic bin in the revised SPRINTER version that implements the new sliding window approach. The distance does not change even when aggregating these bins within larger cell-specific windows because only regions with the same replication timing are aggregated in the same window by SPRINTER. In fact, the average per-cell distance between early and late genomic regions was always between 850-950kb for all cells in the ground truth datasets despite their varying bin sizes (matching the expectations reported by the reviewer).

Despite these observations, we agree with the reviewer that small CNAs or rare CNAs occurring within the few large early/late domains could still be missed by SPRINTER in S phase cells. Therefore, we have now introduced a new method in SPRINTER to specifically correct CNAs in S phase cells that might be missed because they are exclusive to early or late genomic regions, or because they are too small. This new method is based on the key observation that while SPRINTER could have missed certain replication-timing-specific CNAs in S phase cells, these events are not missed in G0/1/2 phase cells because RDR fluctuations are not corrected for replication timing during the CNA analysis of G0/1/2 phase cells. In fact, CNA segments are directly inferred from uncorrected RDRs for these cells as per standard single-cell copy-number methods. We have now clarified this important point in Methods:

“Note that CNA segments are re-inferred directly from RDRs in this step for G0/1/2 phase cells; therefore, rare CNA segments that might have been erroneously excluded from the second step of SPRINTER because they occur in genomic regions with only early or late replication timing can be correctly recovered here, and can also be used later to correct potentially missed CNAs in S phase cells assigned to the same clone.”

Therefore, this new method in SPRINTER corrects replication-timing-specific CNAs in S phase cells by using the CNAs inferred for G0/1/2 phase cells assigned to the same clone. Similarly, small CNAs

are corrected with the same method as they might be more difficult to infer in noisy S phase cells. This new method is now implemented in a new software library “cncorrect.py” in the SPRINTER software package, and we describe it in the revised subsection “Assigning S phase cells to distinct clones” in Methods:

“Moreover, small, rare CNAs that exclusively occur in genomic regions with only early or late replication timing cannot be identified with this method in S phase cells, since the fluctuations induced by these rare CNAs would be erroneously corrected as replication-induced fluctuations and related segments would thus have been excluded in the second step of SPRINTER. Therefore, SPRINTER adds a specific correction for these cases, correcting the CNAs inferred for each S phase cell whenever a replication-timing-exclusive CNA is inferred for the assigned clone (i.e., in the G0/1/2 phase cells belonging to the assigned clone). The same approach is also adopted to correct small CNAs (<5Mb by default) that could be more difficult to identify in S phase cells. Through a copy number spike-in experiment in the generated ground truth datasets, we showed that this approach allows SPRINTER to accurately recover most CNAs of >3Mb in size in both S and non-S phase cells (Supplementary Fig. 23).”

Importantly, in this revision we have also performed a new spike-in experiment in which we have demonstrated that the revised version of SPRINTER is able to accurately recover most CNAs of >3Mb size, especially in S phase cells (new Supplementary Fig. 23, reproduced in Fig. R19 below). Specifically, we have used the HCT116 diploid cells sequenced as part of the ground truth dataset, and we have spiked-in copy-number gains and losses by correspondingly scaling the observed read counts (using a factor of 1/2 for losses and a factor of 2 for gains, Fig. R19a). Moreover, we have only spiked these copy-number events into chromosomes that are known to be diploid in all of these cells. As such, we have applied SPRINTER to all the cells in this spiked-in dataset and we have measured the ability of SPRINTER to accurately recover CNAs of varying size (3-15Mb). We found that SPRINTER is able to recover these CNAs with high accuracy, precision, and recall (>80%) when considering the raw CNAs directly inferred by SPRINTER for individual cells in both G1/2 and S phase, and with even higher performance (>90-95%) when considering the new corrections that are now applied by the revised version of SPRINTER to the CNAs of S phase cells (and even when considering copy numbers corrected per clone, Fig. R19b-d below). While SPRINTER’s performance is higher for larger CNAs, we found that SPRINTER accurately infers most CNAs with >85% accuracy across all cells (Fig. R19e below).

Fig. R19: Spike-in experiment of CNAs demonstrates that SPRINTER accurately recovers most events of >3Mb in size in both S and non-S phase cells. **a-b**, Copy-number gains (red arrows) and losses (blue arrows) of varying size have been spiked into chromosomes that are known to be diploid in the cells of the diploid ground truth dataset by scaling the original read counts (**a**) by different scaling factors (2 and 1/2, respectively) to obtain spike-in datasets with known events (**b**). **c-e**, Accuracy (y-axis in **c**), recall (y-axis in **d**), and precision (y-axis in **e**) have been measured per cell (each dot) when applied to 500 non-S (blue) and S (orange) phase cells with 55,000 spike-in events (110 events per cell) when considering the raw, uncorrected CNAs inferred directly by SPRINTER in each cell (left), the corrected CNAs inferred by SPRINTER (middle, including correction of replication-timing-specific CNAs in S phase cells and small CNAs), and the CNAs inferred by SPRINTER at the clone-level for each cell (right). **f**, The accuracy (y-axis) for the same CNAs inferred by SPRINTER with different corrections (colours) is measured for events of different sizes (x-axis).

Response 3.8 3. Missed CNAs. Similar to the comment above, the authors describe that most CNAs are over 1Mb in size, does this mean that SPRINTER is missing smaller CNAs? Or are these smaller CNAs beyond the capability of scWGS? Can the authors show e.g. using the Funnell et al data, that they are identifying the same complement of CNAs as Funnell? Or if not, what kind of CNAs are missed by SPRINTER? Alternatively, the authors could compare the clone level CNAs to CNAs found in bulk samples. The authors did compare to bulk samples for SNVs for the NSCLC dataset, but as far as I can see, they didn't compare the CNAs.

As shown in related previous single-cell studies^{2-4,22,31}, including the Funnell *et al.*¹ study, the low sequencing coverage of these scDNA-seq technologies provides a resolution that is sufficient to identify CNAs of >1Mb in size on average in individual cells. Therefore, the size of the CNAs inferred by SPRINTER in this study is limited by the resolution of the scDNA-seq technologies and the corresponding sequencing coverage (required to scale to large numbers of cells). Theoretically, SPRINTER could be applied to identify smaller CNAs in individual cells that are sequenced with a deeper sequencing coverage, since there is no methodological limitation to this possibility. Moreover, smaller CNAs might potentially be identified by performing a per-clone pseudobulk analysis in which all the sequencing reads belonging to cells assigned to the same clone are collapsed together². As such, SPRINTER's results (especially clone identification and assignment) could also enable the identification of smaller CNAs within each clone. We have clarified this opportunity for future analyses by adding a new related sentence in the revised Discussion:

“Lastly, due to the low sequencing coverage of scDNA-seq, we also note that some SNVs and particularly short CNAs only present in small clones might have been missed by these analyses. Therefore, further improvements in pseudobulk analyses or sequencing with deeper coverage could reveal more genetic events associated with changes in proliferation.”

As requested by the reviewer, in this revision we have now performed a new analysis comparing the CNAs inferred by SPRINTER with the CNAs inferred in the previous study of Funnell *et al.*¹ on the TNBC and HGSC datasets, revealing high consistency between the results in the two studies (new Supplementary Fig. 50, reproduced in Fig. R20 below). In fact, SPRINTER infers the same single cell copy numbers as in the previous analysis by Funnell *et al.*¹ for 89.8% of the genome in median across different samples in cells inferred with the same ploidy, corresponding to ~90% of the total number of non-S phase cells analysed (Fig. R20a,b below). Furthermore, when normalising by the inferred ploidy to make the copy numbers inferred across all cells comparable, we found that SPRINTER classifies ~93% of the genome in median across samples into either copy-number gained, lost, or neutral states equal to the classifications yielded by the previous analysis of Funnell *et al.*¹. In fact, the normalised copy numbers in the two studies have a very high Spearman correlation of 0.95 with high similarity across all cells (Fig. R20c below).

a Previous analysis of CNAs from Funnell *et al.*

b New SPRINTER analysis of CNAs

c Correlation between CNAs inferred by SPRINTER and in the previous analysis of Funnell *et al.*

Fig. R20: SPRINTER infers CNAs that are highly consistent with previous analysis by Funnell *et al.* **a-b**, Heatmaps of copy number normalised by ploidy (colours) in every cell (y-axis) across the genome (x-axis) as inferred previously in the Funnell *et al.*¹ analysis in **(a)** and in non-S phase cells inferred by SPRINTER in **(b)**. **c**, (Left) Spearman correlation between copy numbers normalised by ploidy inferred previously in the Funnell *et al.* analysis and by SPRINTER in non-S phase cells (x-axis) measured per cell (y-axis showing count of cells). (Right) Copy numbers normalised by ploidy inferred by SPRINTER in non-S phase cells (x-axis) and by previous Funnell *et al.* analysis (y-axis) for every cell (dots), demonstrating an overall Spearman correlation of 0.95.

Similarly, as requested by the reviewer, we have also performed a new analysis comparing the CNAs inferred by SPRINTER with the CNAs inferred by the previous bulk analysis of primary tumour samples from the NSCLC case⁴³ (new Supplementary Fig. 39, reproduced in Fig. R21 below). Even in this case, we found that the copy numbers inferred by SPRINTER across all cells in each sample very closely match the fractional copy numbers inferred in the previous bulk analysis when normalised by the tumour purity and ploidy of the related bulk samples (Fig. R21a,b below). In fact,

the copy numbers inferred by SPRINTER and the previous bulk analysis across the whole genome display a very high correlation between 0.89-0.94 for all samples (Fig. R21c below).

a Previous bulk analysis of CNAs

b SPRINTER analysis of CNAs

c Comparison of bulk and SPRINTER copy numbers

Fig. R21: SPRINTER infers CNAs in the NSCLC dataset that are highly consistent with previous bulk studies of matched primary tumour regions. **a**, Normalised fractional copy numbers (colours, representing averages across all cells within the bulk sample) from previous bulk studies⁴³ for five primary tumour regions (rows) in 1Mb genomic bins across all autosomes (columns) were calculated after correcting for tumour sample purity and ploidy. **b**, For each matched primary tumour region (rows), normalised copy numbers (colours) for the same 1Mb genomic bins across all autosomes (columns) were computed as the average of the copy numbers inferred by SPRINTER for all cancer cells sequenced from the same sample. **c**, For each primary tumour region (each plot), the normalised copy numbers inferred in previous bulk studies (y-axis) for 1Mb genomic bins (dots) is compared to normalised copy numbers calculated using SPRINTER. A Pearson correlation is computed for each sample (coefficient labelled at the top and linear relation represented as a line).

Response 3.9 Furthermore, due to the replication aware nature of this method, as far as I understand, only 50-70% of the genome is assayed as these are the regions with constant replication timing between cell types. Does this mean that CNAs in the other 30-50% of the genome are not detected? And therefore could SPRINTER potentially miss out on clones and subclones where CNAs differ in the undetectable regions?

The 50kb genomic bins that are excluded by SPRINTER during the analysis of S phase cells using default parameters (based on the permissive default thresholds and the default input replication profiles) only represent <20% of the genome (as shown in Supplementary Fig. 1) and, importantly, they are distributed across the whole genome: in fact, the median size of consecutive genomic regions that have been discarded is <150kb on average across all cells. These discarded genomic regions are thus substantially smaller than the CNAs that are inferred in single cells (as explained in detail in Response 3.7 above), and they are not expected to affect SPRINTER's identification of CNAs.

Even more importantly, we highlight that genomic bins without consistent early/late replication timing are discarded by SPRINTER only in the CNA analysis of S phase cells, and they are preserved in the analysis of G0/1/2 phase cells. Since SPRINTER infers tumour clones only using G0/1/2 phase cells and S phase cells are later assigned to these clones, the excluded genomic regions do not affect the identification of clones. We have added a new sentence in the subsection "*Deriving replication-specific DNA sequencing signals*" in Methods to clarify this point:

"The remaining bins are classified as unknown and they are only used in the CNA analysis of G0/1/2 phase cells. In fact, we expect that preserving >50% of the bins with conserved early/late replication timing is sufficient for S phase identification and clone assignment since CNAs are large (mostly >2Mb⁴⁹⁻⁵¹, Supplementary Fig. 2), and replication fluctuations are substantially shorter (Supplementary Fig. 2) and occur across the whole genome (Supplementary Figs. 7 and 8)."

Moreover, we note that the CNAs that could be hypothetically missed in discarded regions in S phase cells can now be recovered using the CNAs accurately inferred in the G0/1/2 phase cells assigned to the same clone using the new correction introduced in the revised version of SPRINTER (related details in Response 3.7 above).

Response 3.10 4. As a follow on from the previous comment, I would be curious to see the NSCLC evolutionary tree delineated by CNAs instead of SNVs. This was described in the text but I do not see a figure in the main figures or the supplementary. Did the CNA phylogeny differ to the SNV phylogeny? And how so? I understand that the primary purpose of inferring replication aware CNAs is so you can better identify clones and proliferation level, however, beyond this the authors primarily used SNVs to look at evolutionary history of the NSCLC dataset. For example, one question that could be asked is whether the clones that contribute to ctDNA have a high CNA burden? The perspective here is that CNAs and SNVs can have potentially very different consequences on a cell and result from very different mutational pathways. Therefore, looking at both and how they are different could highlight different aspects of tumor history. The results here could add a more significance to this work.

We apologise for forgetting to include the tumour phylogenetic tree delineated by CNAs in the previous version of this manuscript. In this revision, we have now introduced the new Supplementary Fig. 44 (reproduced in Fig. R22 below) that represents the inferred phylogenetic tree in terms of CNAs. Following the approach described in the section "*Phylogenetic analysis*" in Methods, this CNA phylogeny has been reconstructed by first fixing the same topology as the SNV phylogeny and then applying the MEDICC algorithm to reconstruct the evolutionary history of CNAs using this fixed topology. Nevertheless, we found that the CNA phylogeny obtained with this approach is highly consistent with the one that is inferred using only CNAs: in fact, the numbers of events measured along the evolutionary path between any pair of clones is highly similar between the two phylogenies (new Fig. R23 below). Moreover, we found that reconstructing CNA evolution without fixing the topology resulted in a phylogeny with a similar total number of events (<8% difference) and a similar median number of events per edge (8 vs 9).

Fig. R22: Reconstructed evolution of CNAs for the NSCLC dataset. CNA evolution was reconstructed by the MEDICC algorithm, which took as input the SNV tree topology and the copy numbers (colours) inferred by SPRINTER across the genome (columns) for extant clones (rows with coloured circles) in order to infer the copy numbers of the ancestral clones (rows without coloured circles).

Fig. R23: The reconstructed CNA phylogeny is very similar to the CNA phylogeny that would have been reconstructed without using the SNV topology. Between every pair of extant clones (dot), the overall number of CNA events along all the connecting edges was calculated when the phylogeny was reconstructed by fixing the same topology as the reconstructed SNV phylogeny (x-axis) or without fixing the topology (y-axis).

We highlight that CNAs have not only been used to identify clones and their proliferation in this study, but the reconstructed evolutionary history of CNAs has also been used in addition to SNVs to show the significant association between high clone proliferation and increased clone metastatic seeding potential. The details of this analysis are described in the third paragraph of the section *“Increased metastatic seeding potential and ctDNA shedding of high proliferation clones”* in Results (revised Fig. 4e). In fact, we found a significant negative correlation between the S fractions of extant clones and the genetic distance to seeding clones in terms of both SNVs and CNAs (revised Fig. 4e). The consistency between SNV and CNA evolution provide further support to our results from the analysis of the NSCLC dataset.

Related to ctDNA, we observed that the clones that mostly contribute to ctDNA shedding are those present in the third metastatic clade, which include the highest proliferation tumour clones found in the two adrenal and left frontal lobe brain metastases (as shown in the revised Fig. 4f and in the new Supplementary Fig. 49). However, these clones do not appear to have a CNA burden higher than other clones found in other samples with lower ctDNA shedding. For example, while the clones found in the liver metastasis and primary tumour regions T1R2 and T1R8 (purple and magenta leaves in Fig. R22 above) seem to have a higher burden of CNAs and underwent more whole-genome doublings than the clones found in the adrenal and left frontal lobe metastases (blue leaves in Fig. R22 above), the former belong to a clade shedding substantially less ctDNA than clones in the other clades (with ctDNA shedding index -0.1 vs 0.4 on average, respectively, corresponding to the brown vs blue dots in revised Fig. 4f). Therefore, proliferation seems to be more of a determinant of ctDNA shedding than CNA burden in this metastatic NSCLC case, as reported in the results of our study.

Response 3.11 5. Proliferation rate inferred from S-phase and G2 cell fractions. In general, I agree with using the fraction of S-phase cells as a proxy for how proliferative a clone is. The assumption being that the more proliferative the clone, the more likely you are to capture cells in S-phase at any given time. However, this may oversimplify the cell cycle diversity of cancer. I was unsure of the reasoning behind showing the proportion of G2 cells or S+G2 cells, e.g. Fig. 5a, Supp Fig. 23, yet the main measure the authors use to compare proliferation between clones is the S-phase fraction. Is this because one might assume that if cells undergo S-phase, then they would eventually make it to G2? However, if you look at the S+G2 graph in Fig. 5a, the S fraction only rank is different to the S+G2 fraction ranking of clones. Therefore, there is some sort of discrepancy in agreement between S fraction and G2 fraction. This may be due to several things like perhaps SPRINTER not being able to pick up G2 cells as well as G1 and S, or more biological considerations like cell cycle arrest that can occur in cells with compromised DNA repair pathways or cells with higher ploidy (Matthews, Bertoli, and de Bruin 2022; Storchova and Pellman 2004). For example, DNA damage and replication stress can lead to S-phase and G2/M cell cycle arrest, which prolongs the time the cell spends in G2, before exiting to apoptosis or senescence (Matthews, Bertoli, and de Bruin 2022). Have the authors investigated this in their data? Would the data tell a different story if a more integrated G1:S:G2 value was used instead of just the S fraction? E.g. for Fig. 4f, Fig. 5b,c,d. Also, in the tumor datasets (HSCLC, TNBC, HGSC), the authors do have mutational information for several important genes (Supp Fig. 30, Funnell et al). The authors could look at the S:G2 ratios (or even G1:S:G2 ratios to be more comprehensive) for clones with differing mutations. This could be why there were not strong correlations found between S fraction and SNV, SV or CNA rate overall (Fig. 5b,c,d). Firstly, because maybe S fraction is only one part of the cell cycle picture, so to speak, and secondly, because you might expect clones with differing mutation backgrounds to have a different correlation between proliferation and mutation rate. Again, the results here could add a more significance to this work.

The reviewer is correct in recognising that the comparison of S and G2 fractions was motivated by the most basic assumption that, in general and in the simplest scenario, a higher fraction of S phase cells would also induce a higher fraction of G2 phase cells in the same clone. As such, the significant positive correlation between S and G2 fractions identified in the NSCLC case provided further support to the estimated proliferation rates. At the same time, we recognise that SPRINTER's estimates of S fractions are expected to be more robust than the estimates of G2 fractions because SPRINTER's identification of S phase cells leverages signals from thousands of genomic bins across the whole genome of each cell, while G2 identification only relies on a single noisy signal per cell, namely the total number of reads sequenced from each cell. We have now clarified this important point in the Discussion:

“For example, while most G2 phase cells yield higher total read counts than G0/1 phase cells, a non-negligible fraction of G2 phase cells do not necessarily display this increase (Supplementary Fig. 10). For this reason, S fraction estimates are expected to be more robust than G2 fraction estimates, since the identification of S phase cells leverages signals from thousands of genomic regions across the entire genome, while G2 phase identification only relies on a single signal related to total read count.”

Therefore, based on these important considerations, we have mainly used the estimated clone S fractions as a proxy for proliferation rate in this study, in agreement with the reviewer and in line with the gold standard pathological approaches used for measuring proliferation at the sample level that are implemented in routine clinical practice^{9,10,12,14}. In the NSCLC case, the strong correlation

between S and G2 fractions (which has been revised in a more robust analysis described below in this response) across the clones in the same tumour is thus used to provide further support to the estimates obtained from S fractions. At the same time, we agree that a joint analysis of the clone S and G2 fractions estimated by SPRINTER can now provide a more comprehensive view of the cell cycle and could reveal further useful insights in some cases. Therefore, we have also performed new related analyses leveraging the availability of multiple patients in the previous TNBC and HGSC datasets (details and results are described below in this response).

Furthermore, we have improved the method to identify G2 phase cells in the revised version of SPRINTER in order to estimate G2 fractions more robustly. Specifically, we have introduced a quantile sigmoid regression to correct the observed total read counts for differences in cell ploidy across different clones, and we have implemented an approach that uses a confidence threshold when selecting G2 phase cells, which is an approach more robust to sample-specific variations in contrast to the previous maximum-likelihood approach. These new improvements are now described in the section “*Identifying G2 cells in distinct clones*” in Methods:

“As such, the probability of each cell being in G0/1 or G2 phase is computed by using the likelihoods of the fitted model and a uniform prior, and G2 phase cells are defined as those with a probability below a certain threshold of being in G0/1 phase (<0.3 by default). Moreover, to account for the presence of CNAs unique to certain cells and different cell ploidies, the total read counts of all cells are corrected for cell ploidy using a quantile sigmoid regression.”

Overall, following the insightful suggestions from the reviewer and integrating the new improved approach in SPRINTER for G2 identification, we have performed new and comprehensive analyses of the NSCLC case and, especially, of the previous TNBC and HGSC datasets. Specifically, we have performed four main new analyses that include both a more robust version of previous analyses that now support our novel findings more clearly (i.e., (1) a new revised analysis of G2 fractions in the NSCLC case and (2) a new revised analysis of the rates of genomic variants in the TNBC and HGSC datasets), and completely new analyses of the TNBC and HGSC datasets that further reveal novel biological insights (i.e., (3) a new analysis of CNAs and driver mutations enriched in high proliferation clones, and (4) a new analysis of G2 and S fractions in HRD vs non-HRD clones). These new analyses are described below in four subsections in the remainder of this response.

(1) A new revised analysis of G2 fractions in the NSCLC case.

In the first new analysis, we have recomputed the correlation between S and G2 fractions in the NSCLC dataset using the updated results from the new version of SPRINTER and investigated the presence of potential discrepancies. We found a significantly high correlation between the newly computed S and G2 fractions per clone in the NSCLC case (Pearson correlation coefficient 0.74 and p -value $5.32 \cdot 10^{-10}$, new Supplementary Fig. 33a, reproduced in Fig. R24a below). While a few clones display fractions that slightly deviate from the expected relationship, these small differences are explained by stochastic variations expected due to the random sampling process of cells in different phases from each clone. In fact, when considering the 95% confidence interval for both the S and G2 fractions, we see that nearly every clone overlaps with the expected values (Fig. R24a below). Moreover, we found that every clone in the NSCLC dataset has a G2 fraction to S phase fraction ratio that does not significantly differ from the expectation per sample (test of odd ratios in

Supplementary Fig. 33b, reproduced in Fig. R24b below). Overall, these results indicate a high consistency between the estimated S and G2 fractions across all clones in the NSCLC case, further supporting the inferred proliferation rates in our results for this dataset.

a Relationship between S and G2 fractions in CRUK0516

b Comparison of S and G2 fractions to overall expectation per sample

Fig. R24: The S and G2 fractions in the NSCLC dataset have a high significant linear correlation. **a**, The estimated S (x-axis) and G2 (y-axis) fractions are estimated by SPRINTER for each clone in the NSCLC dataset (coloured circle with size proportional to number of cells) without (left) or with (right) estimated 95% confidence intervals. The best linear regression is calculated (black line with 95% confidence interval represented by a shaded area). **b**, The 95% confidence interval for the odds ratio (vertical lines on y-axis) of the relative rate of G2 over S fractions was calculated for each clone (x-axis) and compared with the corresponding sample expectations (dashed line, with the consistency for each sample indicated by the vertical lines of similar colours crossing the dashed horizontal line).

(2) A new revised analysis of the rates of genomic variants in the TNBC and HGSC datasets.

In the second new analysis, we have performed a revised and more powerful analysis of the association between clone proliferation and the clone-specific rates of different genomic variants (SNVs, SVs, and CNAs) in the TNBC and HGSC datasets. Compared to the previous version of this analysis, we have now performed an analysis with increased power by computing the rates of clone-specific SNVs, SVs, and CNAs at the resolution of individual cells. Moreover, we have separated all cells into two groups of high vs low proliferation according to the proliferation rate estimated for the

originating clone. The details of this new analysis are reported in a new paragraph in the section “Dynamics of genomic variants in high proliferation clones” in the revised Results:

“We calculated the clone-specific rate of SNVs, structural variants (SVs), and CNAs for each cell individually by normalising the number of variants per cell by the number of clonal variants for SNVs (i.e., SNVs present in all cells of the clone), or by the number of overall variants per tumour for SVs and CNAs. Moreover, we separated all cells into two groups according to whether they belong to clones with high or low proliferation (bi-partitioning S fractions using the median measured per cancer type, Methods). In both the TNBC and HGSC datasets, we found that cells from high proliferation clones have significantly higher rates of all types of variants compared to cells belonging to low proliferation clones ($p < 2 \cdot 10^{-27}$ and Cohen’s $d \in [0.12, 0.94]$, Fig. 5b-d).”

Note that this analysis has increased power because it accounts for the varying numbers of cells per clone, which can be quite different across the tumours in both the TNBC and HGSC datasets. Compared to the previous version of this analysis, we now observe that high proliferation clones have significantly higher rates of all genomic variants in both the TNBC and HGSC datasets ($p < 2e-27$, corresponding to new Figs. 5b-d in the revision, reproduced in Fig. R25 below) with a high effect size (Cohen’s coefficient between 0.12–0.94 for all tests). Overall, this new, improved analysis more robustly supports the important result already reported in the previous version of this study, indicating that the cells within high proliferation clones acquire a significantly higher number of genomic variants. Overall, this could translate into an evolutionary advantage, with these clones accumulating more key genomic alterations.

Fig. R25: Clones with high proliferation have increased rates of SNVs, SVs and CNAs. a-c, Clone-specific rates of genomic variants were measured in individual cells (y-axis) for (a) SNVs, (b) SVs, and (c) CNAs in high and low proliferation rate clones (as separated by the median of inferred S fractions, x-axis) in the TNBC (left panel) and HGSC (right panel) datasets, with p-values as measured by a Mann Whitney U test and Cohen’s d effect sizes shown.

(3) A new analysis of CNAs and driver mutations enriched in high proliferation clones.

Following the reviewer’s suggestion, we have performed a third, completely new analysis in the TNBC and HGSC datasets to investigate whether certain driver mutations or CNAs affecting known cancer genes are enriched in high proliferation clones. In particular, we describe the new analysis

in a paragraph of the new section “*Identifying genomic alterations enriched in high proliferation clones*” in the revised Methods:

“In the TNBC and HGSC datasets, a hypothesis testing approach has been used to identify amplifications of known oncogenes, deletions of known tumour suppressor genes (TSGs), and driver mutations enriched in high proliferation clones. Specifically, amplifications have been identified as genomic regions with an inferred copy number higher than 1.5 times the median copy number per cell, deletions as those with an inferred copy number lower than 0.5 times the median copy number per cell, and driver mutations have been identified using a similar approach to that applied to the NSCLC dataset (Supplementary Methods 10). Also, known oncogenes and TSGs have been obtained from the COSMIC Cancer Gene Census⁵² (v99). As such, for each of these identified events, a one-sided Mann–Whitney U test has been performed comparing SPRINTER’s inferred S fractions for clones without the event to the S fraction for clones harbouring the event. Moreover, we have only considered events present in clones from at least two patients and the analysis has been performed for the TNBC and HGSC datasets separately. After applying a multiple-hypothesis correction using the Benjamini-Hochberg method, a p-value for each test is obtained and each event passing the test is classified as significant and selected as enriched in high proliferation clones. Lastly, a gene set enrichment analysis⁵⁶ has been performed for the selected amplifications that are enriched in high proliferation clones with GSEAp⁵³ (v.1.1.2) using the Molecular Signatures Database (MSigDB) Hallmark 2020 pathway list.”

We investigated the presence of amplifications of oncogenes, deletions of tumour suppressor genes (TSGs), and driver mutations that are enriched in clones with high proliferation (new Figs. 5e,f and Supplementary Fig. 53 in the revision, reproduced in Figs. R26 and R27 below). As such, we found several amplifications of known oncogenes that are significantly associated with increased clone proliferation (e.g., CDK4, CCND2, ERBB2, ERBB3, EGFR, AKT3, HRAS, shown in Fig. 5e, reproduced in Fig. R26a below). This association is further supported by the fact that these genes are enriched in relevant pathways related to the cell cycle and proliferation (e.g., PI3K/AKT/mTOR signalling, KRAS signalling upregulation) as shown using a gene set enrichment analysis³² (Fig. 5f, reproduced in Fig. R26b below, with details in Methods). Moreover, we found a small number of deletions in TSGs (e.g., SMAD4, KEAP1) and driver mutations (e.g., KMT2D, EPHA7) which are significantly associated with high clone proliferation (Supplementary Fig. 53, reproduced in Fig. R27 below). For some of these genes (e.g., ARID2, KEAP1, PTPRD, TTN, MROH2B), the anti-proliferative and tumour suppressive effects of related deletions or mutations match the validation measurements performed in previous cell-line siRNA experiments³³. Overall, these results highlight that the joint analysis of clone proliferation and cancer evolution enabled by SPRINTER can elucidate mechanisms underlying changes in cancer proliferation in patient tumours.

a Amplifications in oncogenes associated with proliferation

b Pathway analysis for amplified oncogenes associated with proliferation

Fig. R26: Amplifications of oncogenes associated with proliferation pathways are enriched in high proliferation clones.

a, For each known oncogene (dots, with oncogenes obtained from the COSMIC Cancer Gene Census excluding tumour suppressor genes), a one-sided Mann Whitney U test was used to identify amplifications that are present in clones with significantly higher S fractions than other clones, with p -values corrected for multiple hypotheses using the Benjamini-Hochberg method (y-axis, negative log scale) and the related differences between the average S fractions (x-axis) shown for each test. Genes passing the test (red, with the minimum corrected threshold indicated with the dotted line) are enriched in clones with increased proliferation, with genes relevant to cancer proliferation annotated. **b**, Cancer-relevant pathways (y-axis) enriched for genes with amplifications significantly associated with high clone proliferation from **(a)** were identified using a gene set enrichment analysis (combined scores on x-axis).

a Deletions in TSGs associated with proliferation

b Driver mutations associated with proliferation

Fig. R27: Driver mutations and deletions of tumour suppressor genes (TSGs) associated with high clone proliferation. **a**, For each known TSG (dots, with TSGs obtained from the COSMIC Cancer Gene Census excluding oncogenes), a left-sided Mann Whitney U test was performed between the S fractions of clones with and without a deletion of the TSG to determine a corrected p-value (y-axis, negative log scale), and the related difference between the average S fractions was calculated (x-axis). After applying the Benjamini-Hochberg multiple hypothesis correction, genes passing the test (red with annotations, with the minimum corrected threshold indicated with the dotted line) are enriched in clones with increased proliferation. **b**, For each driver mutation (dots), a left-sided Mann Whitney U test was performed between the S fractions of clones with and without the mutation, and the related difference between the average S fractions was calculated (x-axis). After applying the Benjamini-Hochberg multiple hypothesis correction, genes passing the test (red with annotations, with the minimum corrected threshold indicated with the dotted line) are enriched in clones with increased proliferation.

(4) A new analysis of G2 and S fractions in HRD vs non-HRD clones.

Lastly, following a further suggestion from the reviewer, we have also performed another completely new analysis of the TNBC and HGSC datasets to examine whether variations and discrepancies in the estimated S and G2 fractions could relate to different biological processes, and particularly cell cycle dynamics, by leveraging the availability of multiple distinct tumours from different patients in these large datasets. In fact, as suggested by the reviewer, the estimated clone fractions of S and G2 phase cells can also provide information about changes in the relative length of different cell cycle phases that might occur in cancer. While increased or decreased S fractions are expected to yield increased or decreased G2 fractions, respectively (since the presence of more/less S phase cells generally determines if more/less cells enter G2), an increase or decrease in the G2 fraction without a corresponding variation in the S fraction could indicate a change in the length of G2 phase relative to the length of S phase. Following the reviewer's recommendations, we thus quantified these changes using the fraction of G2 phase cells over the fraction of S phase cells (G2/S ratio) in both these datasets. We compared the G2/S ratio found in clones in which HRD was previously identified in the Funnell *et al.* study¹ to the G2/S ratio in clones without HRD. Remarkably, in the TNBC dataset, we found that HRD clones have a significantly higher G2/S ratio compared to clones without evidence of HRD ($p = 0.008$ in the new Supplementary Fig. 54, reproduced in Fig. R28 below). This result is supported by its consistency when considering clones containing larger numbers of cells (Fig. R28a-c below). In line with the expectation expressed by the reviewer, this result is thus consistent with a possible prolonged G2 phase relative to the length of S phase in HRD clones, as reported in previous studies^{34,48} which indicated that prolonged G2/M cell cycle arrest may occur in the absence of the ability to properly repair double strand breaks via homologous recombination. We did not observe a similar difference in the clones from the HGSC dataset, possibly because of a lack of power due to a substantially lower number of cells per tumour (~837 on average in the HGSC dataset vs ~4035 on average in the TNBC dataset), or due to biological differences in these HGSC tumours.

Overall, these results provide additional novel biological insights and further improve the significance of our study. At the same time, they demonstrate that SPRINTER will also enable future studies investigating changes in the relative lengths of cell cycle phases during cancer evolution, in addition to investigating cancer proliferation. We have added a new related comment in the revised Discussion:

"In this study, we showed that investigating the relative fractions of S and G2 phase cells can provide evidence of potential changes in the relative length of cell cycle phases. Therefore, improvements in G2 identification and more robust models of the cell cycle could enable further investigation of these changes throughout cancer evolution and across different tumours."

a All clones

b Clones with >80 cells

c Clones with >200 cells

Fig. R28: The G2/S ratio is significantly higher in breast cancer clones with homologous recombination deficiency (HRD). The G2/S ratio (x-axis) was calculated based on the G2 and S fractions inferred by SPRINTER in the clones (dots) with or without HRD (y-axis) in the TNBC (left panel) and HGSC (right panel) datasets, with p-values as measured by a Mann Whitney U test when considering all clones inferred by SPRINTER (**a**), only those with more than 80 cells (**b**), and only those with more than 200 cells (**c**). In all panels, box plots show the median and the interquartile range (IQR) with whiskers denoting values within 1.5 times the IQR from the first and third quartiles.

Response 3.12 6. SPRINTER determined S-phase fraction vs. Ki-67 staining, Fig. 3a,b. The Ki-67 staining in parts of the samples are often higher than the detected clones by SPRINTER in the matched region. Can the authors explain why this occurs? Is it a tissue handling issue? E.g. adjacent tissue regions are not that close – are the authors able to mark where the SPRINTER sample was taken in the pictures in Fig. 3b? Or are there limits to how many S-phase cells a clone can contain before there are not enough G0/1/2 cells for SPRINTER to perform clone inference. Therefore, say the 80% Ki-67 stained region in the left adrenal could never be identified as a clone because the S-fraction is too large. It would be useful to understand the limits of detection of differing S-phase fractions in SPRINTER.

In the benchmarking section of this study, we showed that the identification of S phase cells from sequencing data generally results in underestimating the fraction of S phase cells (Fig. R13a,b above). While SPRINTER substantially improves the fraction of correctly recovered S phase cells compared to previous methods, SPRINTER also tends to miss a fraction of early S phase cells (Fig. R13a,b above, also confirmed in previous datasets as shown in Fig. R14 above), thus leading to potential underestimation of the true S fraction. Since Ki-67 might have a higher specificity than sequencing-based methods, this potential underestimation is a likely explanation of the observed tendency towards higher values measured with Ki-67 staining.

As explained in the section “*Pathological assessment of Ki-67*” in Methods, the FFPE samples tested with Ki-67 were taken as close as possible to the sequenced area. However, slides for metastases tended to have a greater distance (generally <10mm) from the sequenced area than that seen for primary tumour samples (~1mm) due to the larger size of the metastatic lesions and their division into FFPE and frozen sections. Given this distance, sampling differences between the tissue used to create FFPE slides and the tissue used for sequencing could also influence the variations observed between the S fractions estimated by Ki-67 and SPRINTER. For the same reasons, the reported slides cannot be marked with the location of the sequenced areas.

Lastly, we highlight that the power of SPRINTER to accurately recover tumour clones mostly depends on the absolute number of G0/1/2 phase cells sampled, as these are the cells used to identify clones. We agree with the reviewer that investigating the limits of clone detection is important. Therefore, in this revision, we have performed a new analysis to investigate the power and precision of SPRINTER to identify tumour clones with varying numbers of G0/1/2 phase cells in each sample independently. The details of this new analysis are described in detail in Response 1.4 above (results reported in Fig. R4 above). Overall, we found that SPRINTER can accurately recover clones with >30 cells identified as being in G0/1/2 phase (matching the measurements obtained in previous related single-cell studies⁴). Since the average number of cells per clone is ~160 in the NSCLC dataset, clones with an S fraction of 80% can thus be accurately recovered by SPRINTER in this dataset because they are expected to contain at least ~58 cells identified as G0/1/2 phase by SPRINTER (considering SPRINTER’s recovery rate of ~80% for S phase cells as shown in the ground truth benchmark in Fig. R13), for which SPRINTER displays high precision and recall in nearly all scenarios (Fig. R4 above).

Response 3.13 7. I am not sure if it is possible from the NSCLC data to infer more about the order of occurrence of mutations in Supp Fig. 30. It would definitely be interesting and potentially useful to others to know which ones were earlier, or which ones were unique to metastases or even specific metastases etc. I know the authors say that there were no known driver mutations unique to the 3rd clade, but perhaps other mutations could be informative to others and in the future. On similar lines, and also mentioned above, it would be interesting to know if there are particular CNAs unique to this 3rd clade, which may contain genes that could affect cell growth or proliferation when copy number is altered.

We have added a new Supplementary Fig. 43 in the revision (reproduced in Fig. R29 below) to depict the inferred ordering of all the identified driver mutations on the reconstructed tumour phylogenetic tree for the NSCLC case. The ordering of the mutations has been inferred using the method described in the section “*Phylogenetic analysis*” in the revised Methods. Moreover, we did not observe any particular CNA unique to the third metastatic clade of the NSCLC case and we reported the detailed reconstruction of CNA evolution in another new Supplementary Fig. 44 in the revision (reproduced in Fig. R22 above). However, we have introduced a new feature in the SPRINTER algorithm to estimate clone-specific altered replication timing (ART) (details are described in Response 2.2 above). Using this new feature of SPRINTER, we identified ART that is either shared or unique to different metastatic clades, and that affects known cancer genes that have been shown to impact proliferation or metastatic potential (new Fig. 4d, reproduced in Fig. R5 above, with details described in Response 1.5 above). For example, we found ART events that are shared by all clones (e.g., *PDL1*, *TERT*, and *PIK3CA*), unique to only one or two of the metastatic clades on distinct branches of the phylogenetic tree (e.g., *CDK12* and *NCOA2*), or mostly exclusive to the third, most proliferative clade (e.g., *KRAS*). Since ART is known to be associated with differential gene expression (higher/lower expression in regions that have late-to-early or early-to-late ART compared to normal tissue without ART, respectively)²⁴, we showed that these ART events are supported by related gene expression changes by comparing previous bulk RNA-sequencing data³⁰ from matched primary tumour regions to primary regions from different clades, to cancer and normal tissue samples from 347 other TRACERx patients, or to a pre-mortem relapse sample from the left adrenal metastasis in the third clade (Fig. 4d and Supplementary Fig. 47, reproduced in Figs. R5 and R7 above). For example, the comparison across clades and with the pre-mortem relapse sample confirmed a significant increase in *KRAS* expression in the third clade, and particularly in the left adrenal metastasis, which contained the highest proliferation clones, most of which displayed late-to-early ART affecting *KRAS* (Fig. 4d, reproduced in Fig. R5 above). Moreover, reduced *PDL1* expression correlating to early-to-late ART in almost all clones across the tumour was further confirmed by the clinal immunohistochemistry report for this patient (*PDL1* 0%). The details of this new analysis are reported in Response 1.5 above.

Fig. R29: Reconstructed evolution of driver mutations in the NSCLC dataset. The tumour phylogeny was reconstructed using the SNVs from SPRINTER’s single-cell clones (leaves of the tree) in the primary tumour and metastatic samples from patient CRUK0516 (clones are uniquely coloured with source samples represented by shades of the same colour). Seeding clones (dark grey) and the anatomical location of the remaining ancestral clones (white internal colour with border coloured according to the inferred anatomical site) were inferred using the MACHINA algorithm. Some ancestral clones (roman numerals) had SNVs tracked in a previous ctDNA study. The edges of the phylogeny are labelled with the driver mutations that were inferred to occur in those edges.

Response 3.14 Minor comments. 1. Can the authors please add clearer titles on their graphs for all figures? The information is in the figure legends or sometimes on the axes titles, but it would be nice to have the information more obvious at the top of each graph, to be able to understand the graph quicker, for better readability without having to shift between the graph, the main text and then the figure legends, and ultimately to avoid confusion. E.g. Figure 2, c and d. Would be nice to see immediately that c is about diploid cells and d is about tetraploid cells. E.g. Figure 3 b, would be nice to see that these are Ki-67 stains E.g. Figure 3 c, that this is about nuclear diameter This is particularly hard in the supplementary figures, where often the panels of a figure look very similar. E.g. Supp Fig 5, 6, 8, 10, 12, 13, 22, 23 etc.

We agree with the reviewer and we have now updated several figures by adding descriptive titles to most panels, including:

- Fig. 2a,b,c,d,e,f: we added descriptive titles, especially distinguishing experiments on diploid vs tetraploid datasets.
- Fig. 3b,c,d: we added descriptive titles, especially for Ki-67 staining and analysis of nuclear diameter.
- Fig. 4a,b,c,e,f: we added new descriptive titles.
- Supplementary Fig. 9 (previous Supplementary Fig. 5): we added titles to each panel defining raw RDRs, replication timing profiles, and replication-corrected RDRs.
- Supplementary Fig. 10 (previous Supplementary Fig. 6): we added titles to each panel, especially distinguishing data from diploid and tetraploid cells.
- Supplementary Fig. 13 (previous Supplementary Fig. 8): we added titles to each panel, especially distinguishing data from diploid and tetraploid cells.
- Supplementary Fig. 15 (updated version of Supplementary Fig. 12): we added titles to each panel, especially distinguishing results obtained from either diploid or tetraploid cells.
- Supplementary Fig. 16 (previous version of Supplementary Fig. 13): we added titles to each panel explaining the different sets of input profiles used in each case and distinguishing results from either diploid or tetraploid data.
- Supplementary Fig. 17 (previous Supplementary Fig. 10): we added titles to each panel, especially distinguishing the newly generated data from previously generated data.
- Supplementary Fig. 21 (previous Supplementary Fig. 15): we added titles to distinguish diploid and tetraploid data.
- Supplementary Fig. 31 (previous Supplementary Fig. 22): we added titles to each panel distinguishing data from either primary tumour or metastases.
- Supplementary Fig. 32 (previous Supplementary Fig. 23): we added titles to each panel distinguishing S fractions, G2 fractions, and S+G2 fractions.
- Supplementary Fig. 34 (previous Supplementary Fig. 24): we added titles to each panel clarifying that images correspond to Ki-67 staining.
- Supplementary Fig. 35 (previous Supplementary Fig. 25): we added titles to each panel clarifying nuclear diameters inferred from either the diploid or tetraploid datasets.
- Supplementary Fig. 36 (previous Supplementary Fig. 26): we added titles to each panel clarifying nuclear diameters inferred from either the primary tumour, metastases, or per clone.
- Supplementary Fig. 37 (previous Supplementary Fig. 27): we added titles to each panel clarifying the data measured from CT and MR imaging.
- Supplementary Fig. 40 (previous Supplementary Fig. 29): we added titles to each panel clarifying the different sets of SNVs visualised in each.

Response 3.15 2. The authors used ART to simulate read depth control sample from the sequencing reads of the sample itself. I do not understand the details of ART, so I may be missing information here. However, I do wonder about the suitability of this method for the newer Novaseq platforms that use 2-colour technology compared to the older Illumina 4-colour platforms that ART was designed on. The scWGS datasets were sequenced on the Novaseq 6000. Is there anything here to be wary about?

The sequencing data simulated by the ART tool are mostly used to correct mappability biases across different genomic regions by simulating control read counts using sequencing features (like read

length, insert size, etc.) that match those in the sequenced cells for more accurate estimates. While the reviewer is correct in stating that the ART tool cannot produce the same sequencing error profiles as those from new instruments like Novaseq 6000, different sequencing error profiles of different Illumina technologies are not expected to generate significantly different sequencing read counts when considering genomic bins larger than hundreds of thousands of base pairs, as in this study. In fact, we have now performed a new analysis showing that the RDRs estimated using simulated reads from the ART tool are nearly identical to those generated using normal cells as controls, using the same pseudonormal approach applied by the CHISEL algorithm⁴ (Fig. R30 below). Moreover, we highlight that the advantage of the approach based on the ART tool is its applicability to samples without normal-cell admixtures in contrast to previous approaches⁴.

Fig. R30: Read count normalisation using data simulated by the ART tool yields nearly identical RDRs to those obtained by normalising read counts using normal cells. RDRs are measured for each 50kb genomic bin by normalising read counts either using the data simulated with the ART tool as default (x-axis) or considering read counts from normal cells (y-axis) as in previous studies.

References

1. Funnell, T. *et al.* Single-cell genomic variation induced by mutational processes in cancer. *Nature* **612**, 106-115 (2022).
2. Laks, E. *et al.* Clonal Decomposition and DNA Replication States Defined by Scaled Single-Cell Genome Sequencing. *Cell* **179**, 1207-1221 e22 (2019).
3. Minussi, D.C. *et al.* Breast tumours maintain a reservoir of subclonal diversity during expansion. *Nature* **592**, 302-308 (2021).
4. Zaccaria, S. & Raphael, B.J. Characterizing allele- and haplotype-specific copy numbers in single cells with CHISEL. *Nat Biotechnol* **39**, 207-214 (2021).
5. Garvin, T. *et al.* Interactive analysis and assessment of single-cell copy-number variations. *Nat Methods* **12**, 1058-60 (2015).
6. Kızılkale C. *et al.* Fast intratumor heterogeneity inference from single-cell sequencing data. *Nat Comput Sci* **2**, 577-583 (2022).
7. Andor, N. *et al.* Joint single cell DNA-seq and RNA-seq of gastric cancer cell lines reveals rules of in vitro evolution. *NAR Genom Bioinform* **2**, lqaa016 (2020).
8. Andrisani, O.M., Studach, L. & Merle, P. Gene signatures in hepatocellular carcinoma (HCC). *Semin Cancer Biol* **21**, 4-9 (2011).
9. Beresford, M.J., Wilson, G.D. & Makris, A. Measuring proliferation in breast cancer: practicalities and applications. *Breast Cancer Res* **8**, 216 (2006).
10. Brown, D.C. & Gatter, K.C. Ki67 protein: the immaculate deception? *Histopathology* **40**, 2-11 (2002).
11. Cuzick, J. *et al.* Prognostic value of an RNA expression signature derived from cell cycle proliferation genes in patients with prostate cancer: a retrospective study. *Lancet Oncol* **12**, 245-55 (2011).
12. Feitelson, M.A. *et al.* Sustained proliferation in cancer: Mechanisms and novel therapeutic targets. *Semin Cancer Biol* **35 Suppl**, S25-S54 (2015).
13. Scialdone, A. *et al.* Computational assignment of cell-cycle stage from single-cell transcriptome data. *Methods* **85**, 54-61 (2015).
14. van Diest, P.J., van der Wall, E. & Baak, J.P. Prognostic value of proliferation in invasive breast cancer: a review. *J Clin Pathol* **57**, 675-81 (2004).
15. Wistuba, II *et al.* Validation of a proliferation-based expression signature as prognostic marker in early stage lung adenocarcinoma. *Clin Cancer Res* **19**, 6261-71 (2013).
16. Akhmetkaliyev, A., Alibrahim, N., Shafiee, D. & Tulchinsky, E. EMT/MET plasticity in cancer and Go-or-Grow decisions in quiescence: the two sides of the same coin? *Mol Cancer* **22**, 90 (2023).
17. Mejlvang, J. *et al.* Direct repression of cyclin D1 by SIP1 attenuates cell cycle progression in cells undergoing an epithelial mesenchymal transition. *Mol Biol Cell* **18**, 4615-24 (2007).
18. Shin, S. *et al.* ERK2 regulates epithelial-to-mesenchymal plasticity through DOCK10-dependent Rac1/FoxO1 activation. *Proc Natl Acad Sci U S A* **116**, 2967-2976 (2019).
19. Vega, S. *et al.* Snail blocks the cell cycle and confers resistance to cell death. *Genes Dev* **18**, 1131-43 (2004).
20. Abbosh, C. *et al.* Tracking early lung cancer metastatic dissemination in TRACERx using ctDNA. *Nature* **616**, 553-562 (2023).
21. Magbanua, M.J.M. *et al.* Clinical significance and biology of circulating tumor DNA in high-risk early-stage HER2-negative breast cancer receiving neoadjuvant chemotherapy. *Cancer Cell* **41**, 1091-1102 e4 (2023).

22. Abbosh, C. *et al.* Phylogenetic ctDNA analysis depicts early-stage lung cancer evolution. *Nature* **545**, 446-451 (2017).
23. Dietzen, M. *et al.* Replication timing alterations are associated with mutation acquisition during tumour evolution in breast and lung cancer. *Nature Communications* (*in press*) (2024).
24. Donley, N. & Thayer, M.J. DNA replication timing, genome stability and cancer: late and/or delayed DNA replication timing is associated with increased genomic instability. *Semin Cancer Biol* **23**, 80-9 (2013).
25. Du, Q. *et al.* Replication timing and epigenome remodelling are associated with the nature of chromosomal rearrangements in cancer. *Nat Commun* **10**, 416 (2019).
26. Rivera-Mulia, J.C. *et al.* Replication timing alterations in leukemia affect clinically relevant chromosome domains. *Blood Adv* **3**, 3201-3213 (2019).
27. Ryba, T. *et al.* Abnormal developmental control of replication-timing domains in pediatric acute lymphoblastic leukemia. *Genome Res* **22**, 1833-44 (2012).
28. Ryba, T. *et al.* Evolutionarily conserved replication timing profiles predict long-range chromatin interactions and distinguish closely related cell types. *Genome Res* **20**, 761-70 (2010).
29. Sasaki, T. *et al.* Stability of patient-specific features of altered DNA replication timing in xenografts of primary human acute lymphoblastic leukemia. *Exp Hematol* **51**, 71-82 e3 (2017).
30. Martinez-Ruiz, C. *et al.* Genomic-transcriptomic evolution in lung cancer and metastasis. *Nature* **616**, 543-552 (2023).
31. Massey, D.J. & Koren, A. High-throughput analysis of single human cells reveals the complex nature of DNA replication timing control. *Nat Commun* **13**, 2402 (2022).
32. Subramanian, A. *et al.* Gene set enrichment analysis: a knowledge-based approach for interpreting genome-wide expression profiles. *Proc Natl Acad Sci U S A* **102**, 15545-50 (2005).
33. Hobor, S. *et al.* Mixed responses to targeted therapy driven by chromosomal instability through p53 dysfunction and genome doubling. *Nature Communications* (*in press*) (2024).
34. Kostyrko, K., Bosshard, S., Urban, Z. & Mermod, N. A role for homologous recombination proteins in cell cycle regulation. *Cell Cycle* **14**, 2853-61 (2015).
35. Rhind, N. & Gilbert, D.M. DNA replication timing. *Cold Spring Harb Perspect Biol* **5**, a010132 (2013).
36. Yaffe, E. *et al.* Comparative analysis of DNA replication timing reveals conserved large-scale chromosomal architecture. *PLoS Genet* **6**, e1001011 (2010).
37. Dileep, V. & Gilbert, D.M. Single-cell replication profiling to measure stochastic variation in mammalian replication timing. *Nat Commun* **9**, 427 (2018).
38. Gnan, S. *et al.* Kronos scRT: a uniform framework for single-cell replication timing analysis. *Nat Commun* **13**, 2329 (2022).
39. Hansen, R.S. *et al.* Sequencing newly replicated DNA reveals widespread plasticity in human replication timing. *Proc Natl Acad Sci U S A* **107**, 139-44 (2010).
40. Rivera-Mulia, J.C. *et al.* Dynamic changes in replication timing and gene expression during lineage specification of human pluripotent stem cells. *Genome Res* **25**, 1091-103 (2015).
41. Hanzelmann, S., Castelo, R. & Guinney, J. GSEA: gene set variation analysis for microarray and RNA-seq data. *BMC Bioinformatics* **14**, 7 (2013).
42. Al Bakir, M. *et al.* The evolution of non-small cell lung cancer metastases in TRACERx. *Nature* **616**, 534-542 (2023).

43. Frankell, A.M. *et al.* The evolution of lung cancer and impact of subclonal selection in TRACERx. *Nature* **616**, 525-533 (2023).
44. Gerstung, M. *et al.* The evolutionary history of 2,658 cancers. *Nature* **578**, 122-128 (2020).
45. Gabbutt, C. *et al.* Fluctuating methylation clocks for cell lineage tracing at high temporal resolution in human tissues. *Nat Biotechnol* **40**, 720-730 (2022).
46. Pich, O. *et al.* The translational challenges of precision oncology. *Cancer Cell* **40**, 458-478 (2022).
47. Turajlic, S. & Swanton, C. Metastasis as an evolutionary process. *Science* **352**, 169-75 (2016).
48. Matthews, H.K., Bertoli, C. & de Bruin, R.A.M. Cell cycle control in cancer. *Nat Rev Mol Cell Biol* **23**, 74-88 (2022).
49. Dentre, S.C. *et al.* Characterizing genetic intra-tumor heterogeneity across 2,658 human cancer genomes. *Cell* **184**, 2239-2254 e39 (2021).
50. Watkins, T.B.K. *et al.* Pervasive chromosomal instability and karyotype order in tumour evolution. *Nature* **587**, 126-132 (2020).
51. Zack, T.I. *et al.* Pan-cancer patterns of somatic copy number alteration. *Nat Genet* **45**, 1134-40 (2013).
52. Sondka, Z. *et al.* The COSMIC Cancer Gene Census: describing genetic dysfunction across all human cancers. *Nat Rev Cancer* **18**, 696-705 (2018).
53. Fang, Z., Liu, X. & Peltz, G. GSEAPy: a comprehensive package for performing gene set enrichment analysis in Python. *Bioinformatics* **39**(2023).

Decision Letter, first revision:

18th Jul 2024

Dear Dr Zaccaria,

Thank you for submitting your revised manuscript "Characterising the evolutionary dynamics of cancer proliferation in single-cell clones" (NG-A63436R). It has now been seen by the original referees and their comments are below. The reviewers find that the paper has improved in revision, and therefore we'll be happy in principle to publish it in Nature Genetics, pending minor revisions to satisfy the referees' final requests and to comply with our editorial and formatting guidelines.

Sincerely,

Safia Danovi, PhD
Senior Editor, Nature Genetics
ORCID: 0009-0007-7822-5479

Reviewer #1 (Remarks to the Author):

In this revised manuscript, the authors have done extensive work to address the reviewers' concerns. This additional work has significantly strengthened the study— both from a technical standpoint and the biological insights enabled by SPRINTER. I believe that these improvements now make the manuscript suitable for publication in Nature Genetics.

One continued suggestion is that the authors make it clear that this is not a single-cell study but a single-clone study in which clones are defined by CNV. For example, if an SNV/SV is present in 10% of cells of a 30-cell CNV clone, it is unlikely to be captured with this approach. At the extreme, based on studies of normal tissue, each cell likely harbors unique SNV/SV/Indels, and none of those will be captured at the clone level. Thus, this approach is not missing a small fraction but the vast majority of the small variant genetic diversity at the single-cell level. I believe that is an important limitation of the study that should be further highlighted.

Reviewer #3 (Remarks to the Author):

The authors should be commended for an impressive and thorough rebuttal. The authors have addressed all the concerns I have raised. The new ART analysis is particularly interesting and impactful for the field. Below I have a few comments pertaining to the new ART analysis that could improve the understanding, significance and interpretation of this new section.

Main comment:

1. I have been asked by the Editor to look over the RT-related comments from Reviewer 2. In response to Response 2.2, the authors describe that SPRINTER is robust up to 40% of errors or alterations in replication timing. What the authors describe here is convincing. However, for Supplementary Fig. 46, where authors examined per-clone S-fraction before and after removing tumour-specific ART, visually it seems like there is a small but consistent improvement of S-fraction prediction after removal of tumour-specific ART.
 - a. Is this difference significant? Is this within the predictive +/- confidence range of SPRINTER?
 - b. Is this because removing tumour-specific ART goes beyond the % of RT 'error' that SPRINTER is benchmarked to be robust against (i.e. Fig. R6e, it looks like EarlyS and LateS are robust up to 0.26 fraction of RT error, and EarlyS and LateS cells are the easiest to be erroneously called as G1 or G2 cells). I imagine that adding up all the clone-specific ART could result in a large % of tumour-specific ART.
 - c. I do not view this as a negative for SPRINTER. It would make sense that removing additional 'noise' in the form of ART will improve S-fraction prediction even more than what SPRINTER can already do. I am moreso asking for clarification and perhaps a readjustment of the statement that 'SPRINTER is robust to up to 40% of errors...', if that is the case.

Minor comments:

1. The new ART analysis.
 - a. To my knowledge, this is the first time patient solid tumour ART has been able to be elucidated to this degree and correlated with gene expression. In line with the significance of these findings, the authors could consider giving the ART analysis it's own results section or at least altering the title of this results section to reflect this.
 - b. It would be great to see some examples of the actual ART change between clones, in the same manner as Fig. 2g, but obviously more zoomed in to the locus rather than a whole chromosome.
 - c. Figure 4d – could the authors include some form of separation or distinction between clones of clade 2 and 3?

Author Rebuttal, first revision:

Reviewer responses for manuscript NG-A63436R

Characterising the evolutionary dynamics of cancer proliferation in single-cell clones

We thank the reviewers for their additional thoughtful comments. In response to their final comments, we provide a point-by-point response (black text) to each of the reviewer comments (blue text). We also quote related changes to the manuscript in red text. All references to sections and figures refer to the newly revised version of the manuscript, including the revision of the main text and the revision of the Supplementary Information.

Reviewer 1

Response 1.1 In this revised manuscript, the authors have done extensive work to address the reviewers' concerns. This additional work has significantly strengthened the study— both from a technical standpoint and the biological insights enabled by SPRINTER. I believe that these improvements now make the manuscript suitable for publication in Nature Genetics.

We thank the reviewer for their appreciation and positive evaluation of our manuscript.

Response 1.2 One continued suggestion is that the authors make it clear that this is not a single-cell study but a single-clone study in which clones are defined by CNV. For example, if an SNV/SV is present in 10% of cells of a 30-cell CNV clone, it is unlikely to be captured with this approach. At the extreme, based on studies of normal tissue, each cell likely harbors unique SNV/SV/Indels, and none of those will be captured at the clone level. Thus, this approach is not missing a small fraction but the vast majority of the small variant genetic diversity at the single-cell level. I believe that is an important limitation of the study that should be further highlighted.

We have revised the related sentence in the Discussion to further highlight that most mutations unique to individual cells or present at particularly low frequency are missed by the analyses in this study:

“Lastly, we note that low scDNA-seq coverage prevents the comprehensive characterisation of SNVs only present in individual cells, which would require deeper sequencing experiments.”

Reviewer 3

Response 3.1 The authors should be commended for an impressive and thorough rebuttal. The authors have addressed all the concerns I have raised. The new ART analysis is particularly interesting and impactful for the field. Below I have a few comments pertaining to the new ART analysis that could improve the understanding, significance and interpretation of this new section.

We thank the reviewer for their appreciation and positive evaluation of our manuscript and rebuttal.

Response 3.2 Main comment: 1. I have been asked by the Editor to look over the RT-related comments from Reviewer 2. In response to Response 2.2, the authors describe that SPRINTER is robust up to 40% of errors or alterations in replication timing. What the authors describe here is convincing. However, for Supplementary Fig. 46, where authors examined per-clone S-fraction before and after removing tumour-specific ART, visually it seems like there is a small but consistent improvement of S-fraction prediction after removal of tumour-specific ART.

A minor improvement in the estimated S fractions is expected when excluding tumour-specific ART events because it can improve the sensitivity of the hypothesis test introduced by SPRINTER to identify S phase cells. In fact, excluding tumour-specific ART events (i.e., corresponding to early or late bins with read counts lower or higher than expected, respectively) can generally lead to a slight increase in the observed values of the test statistics used by SPRINTER, which capture the relative fraction of early and late bins with higher and lower values of the replication timing profile than expected, respectively, consequently leading to a slight increase in the sensitivity (further related details are in the subsection "*Identifying S phase cells*" in Methods). In this new version of the manuscript, we have introduced a new analysis reported in the revised Supplementary Fig. 42 (reproduced in Fig. F1 below with further details described in Response 3.3 below) to confirm that these increases in the estimated S fractions are within the predictive confidence range of SPRINTER (Fig. F1b below) and are not significantly different to the default results (Fig. F1c below). Moreover, we calculated that >90% of the clones only have up to 12 additional S phase cells, further confirming that the differences compared to the default results are minor and are consistent with the expected slight increase in sensitivity. Finally, we highlight that the increase in the estimated S fractions mostly occurred in the samples containing the highest proliferation clones belonging to the third metastatic clade (i.e., clones in samples AD01 and AD05 from the adrenal metastases, and sample BR03 from the left frontal lobe metastasis), in which SPRINTER is more likely to have missed some S phase cells as also seen from the Ki-67 analysis (as described in Response 3.12 from the previous point-by-point reviewer responses). Overall, this observation further confirms that the minor observed differences compared to the default results are not significantly different and correspond to minor improvements in the sensitivity for the most proliferative clones.

Fig. F1: SPINTER's estimates of S fractions are not affected by clone-specific altered replication timing (ART). **a**, The bootstrapped estimate of the S fraction of each clone (y-axis) was calculated by sampling cells with replacement in each SPINTER-inferred clone from each sample of the NSCLC dataset (x-axis), using either using the default replication timing classification (green) or excluding genomic regions in which high-confidence ART events (found in most clones in >2 samples) have been identified (orange). **b**, The fraction of S phase cells assigned to each clone (each dot was sized proportionally to the corresponding number of cells and with colours matching those defined in (c)) has been directly estimated by SPINTER either using the default replication timing classification (x-axis) or excluding high-confidence ART events (y-axis). The expected range of uncertainty (grey shadow with diagonal represented as a dashed line) is calculated using the average size of the 99% confidence interval estimated using bootstrapping per clone from SPINTER's results. **c**, The 99% confidence interval for the odds ratio (vertical lines on y-axis) was calculated for each clone (x-axis) comparing the S fractions estimated using the default replication timing classification to those excluding high-confidence ART events (dashed line, with the consistency between estimates indicated by the vertical lines crossing the dashed horizontal line).

Response 3.3 a. Is this difference significant? Is this within the predictive +/- confidence range of SPINTER?

We have performed a new analysis showing that the differences between the S fractions estimated with default replication timing and excluding ART events are small and within the confidence range of SPINTER (Fig. F1b). Specifically, we estimated the confidence range of SPINTER using the same bootstrapping approach applied in other analysis in this study (e.g., Fig. 3a). Moreover, in this new analysis we also showed that the S fractions estimated excluding ART events are not significantly different than the S fractions estimated with default replication timing for every clone across all samples (Fig. F1c).

Response 3.4 b. Is this because removing tumour-specific ART goes beyond the % of RT ‘error’ that SPRINTER is benchmarked to be robust against (i.e. Fig. R6e, it looks like EarlyS and LateS are robust up to 0.26 fraction of RT error, and EarlyS and LateS cells are the easiest to be erroneously called as G1 or G2 cells). I imagine that adding up all the clone-specific ART could result in a large % of tumour-specific ART.

The fraction of tumour-specific ART in every clone is not beyond the fraction of replication timing errors that SPRINTER is robust against, but rather it is well within the benchmarked levels of robustness. In fact, the fraction of tumour-specific ART is below 14% for every clone and <10% on average across clones, well within the range at which SPRINTER remains robust (Supplementary Fig. 16).

Response 3.5 c. I do not view this as a negative for SPRINTER. It would make sense that removing additional ‘noise’ in the form of ART will improve S-fraction prediction even more than what SPRINTER can already do. I am moreso asking for clarification and perhaps a readjustment of the statement that ‘SPRINTER is robust to up to 40% of errors...’, if that is the case.

The reviewer is correct that removing additional noise in the form of ART can lead to minor improvements in the estimation of S fractions, as explained in detail in Response 3.2 above. We also agree with the reviewer’s request and we have reworded the related statement regarding robustness:

“Importantly, SPRINTER’s accuracy remained robust for a fraction of replication-timing errors higher than the maximum expected in both normal and cancer cells (Supplementary Fig. 16) [...]”

Response 3.6 Minor comments: 1. The new ART analysis. a. To my knowledge, this is the first time patient solid tumour ART has been able to be elucidated to this degree and correlated with gene expression. In line with the significance of these findings, the authors could consider giving the ART analysis it’s own results section or at least altering the title of this results section to reflect this.

We thank the reviewer for the useful suggestion which we have implemented in the new revision of this manuscript by separating the new ART analysis into its own new section in Results titled *“The evolution and ART of clones with different proliferation”*.

Response 3.7 b. It would be great to see some examples of the actual ART change between clones, in the same manner as Fig. 2g, but obviously more zoomed in to the locus rather than a whole chromosome.

We have introduced a new Extended Data Fig. 8 (reproduced in Fig. F2 below) to display detailed and zoomed in examples per clone of the data underlying some of the key ART events that we have identified and described in Fig. 4d.

Fig. F2: SPRINTER enables the identification of clone-specific ART supported by underlying read counts. SPRINTER identifies different ART events affecting different genes (annotated text) and present in distinct clones (**a-e**) that belong to different phylogenetic branches (left and right indicated by lilac and light blue rectangles) or different metastatic clades (coloured triangles). SPRINTER identifies clone-specific late-to-early (dark magenta) and early-to-late (dark green) ART events in genomic regions across chromosomes (x-axis) if they have calculated values of the replication timing profile per clone (clone-specific RTP, y-axis) that are higher or lower, respectively, than expected.

Response 3.8 c. Figure 4d – could the authors include some form of separation or distinction between clones of clade 2 and 3?

We have introduced some separators in Fig. 4d to clearly distinguish the second and third metastatic clades.

Final Decision Letter:

15th Oct 2024

Dear Dr Zaccaria,

I am delighted to say that your manuscript "Characterising the evolutionary dynamics of cancer proliferation in single-cell clones with SPRINTER" has been accepted for publication in an upcoming issue of *Nature Genetics*.

Over the next few weeks, your paper will be copyedited to ensure that it conforms to *Nature Genetics* style. Once your paper is typeset, you will receive an email with a link to choose the appropriate publishing options for your paper and our Author Services team will be in touch regarding any additional information that may be required.

Your paper will be published online after we receive your corrections and will appear in print in the next available issue. You can find out your date of online publication by contacting the Nature Press Office (press@nature.com) after sending your e-proof corrections.

Before your paper is published online, we shall be distributing a press release to news organizations worldwide, which may very well include details of your work. We are happy for your institution or funding agency to prepare its own press release, but it must mention the embargo date and *Nature Genetics*. Our Press Office may contact you closer to the time of publication, but if you or your Press Office have any enquiries in the meantime, please contact press@nature.com.

Please note that *Nature Genetics* is a Transformative Journal (TJ). Authors may publish their research with us through the traditional subscription access route or make their paper immediately open access through payment of an article-processing charge (APC). Authors will not be required to make a final

decision about access to their article until it has been accepted. Find out more about Transformative Journals

Authors may need to take specific actions to achieve compliance with funder and institutional open access mandates. If your research is supported by a funder that requires immediate open access (e.g. according to Plan S principles) then you should select the gold OA route, and we will direct you to the compliant route where possible. For authors selecting the subscription publication route, the journal's standard licensing terms will need to be accepted, including <https://www.nature.com/nature-portfolio/editorial-policies/self-archiving-and-license-to-publish>. Those licensing terms will supersede any other terms that the author or any third party may assert apply to any version of the manuscript.

If you have not already done so, we strongly recommend that you upload the step-by-step protocols used in this manuscript to protocols.io. protocols.io is an open online resource that allows researchers to share their detailed experimental know-how. All uploaded protocols are made freely available and are assigned DOIs for ease of citation. Protocols can be linked to any publications in which they are used and will be linked to from your article. You can also establish a dedicated workspace to collect all your lab Protocols. By uploading your Protocols to protocols.io, you are enabling researchers to more readily reproduce or adapt the methodology you use, as well as increasing the visibility of your protocols and papers. Upload your Protocols at <https://protocols.io>. Further information can be found at <https://www.protocols.io/help/publish-articles>.

Sincerely,

Safia Danovi, PhD
Senior Editor, Nature Genetics
ORCID: 0009-0007-7822-5479